# Benign Overfitting and Grokking in ReLU Networks for XOR Cluster Data

**Zhiwei Xu**[†]**, Yutong Wang**[†]**, Spencer Frei**[‡]**, Gal Vardi**[◇]**, Wei Hu**[†]
[†]University of Michigan, [‡]University of California, Davis, [◇]TTI-Chicago and Hebrew University
{zhiweixu,yutongw,vvh}@umich.edu,sfrei@ucdavis.edu,galvardi@ttic.edu

## Abstract

Neural networks trained by gradient descent (GD) have exhibited a number of surprising generalization behaviors. First, they can achieve a perfect fit to noisy training data and still generalize near-optimally, showing that overfitting can sometimes be benign. Second, they can undergo a period of classical, harmful overfitting—achieving a perfect fit to training data with near-random performance on test data—before transitioning ("grokking") to near-optimal generalization later in training. In this work, we show that both of these phenomena provably occur in two-layer ReLU networks trained by GD on XOR cluster data where a constant fraction of the training labels are flipped. In this setting, we show that after the first step of GD, the network achieves 100% training accuracy, perfectly fitting the noisy labels in the training data, but achieves near-random test accuracy. At a later training step, the network achieves near-optimal test accuracy while still fitting the random labels in the training data, exhibiting a "grokking" phenomenon. This provides the first theoretical result of benign overfitting in neural network classification when the data distribution is not linearly separable. Our proofs rely on analyzing the feature learning process under GD, which reveals that the network implements a non-generalizable linear classifier after one step and gradually learns generalizable features in later steps.

## 1 Introduction

Classical wisdom in machine learning regards overfitting to noisy training data as harmful for generalization, and regularization techniques such as early stopping have been developed to prevent overfitting. However, modern neural networks can exhibit a number of counterintuitive phenomena that contravene this classical wisdom. Two intriguing phenomena that have attracted significant attention in recent years are *benign overfitting* (Bartlett et al., 2020) and *grokking* (Power et al., 2022):

- **Benign overfitting:** A model perfectly fits noisily labeled training data, but still achieves near-optimal test error.
- **Grokking:** A model initially achieves perfect training accuracy but no generalization (i.e. no better than a random predictor), and upon further training, transitions to almost perfect generalization.

Recent theoretical work has established benign overfitting in a variety of settings, including linear regression (Hastie et al., 2019; Bartlett et al., 2020), linear classification (Chatterji & Long, 2021a; Wang & Thrampoulidis, 2021), kernel methods (Belkin et al., 2019; Liang & Rakhlin, 2020), and neural network classification (Frei et al., 2022b; Kou et al., 2023). However, existing results of benign overfitting in neural network classification settings are restricted to linearly separable data distributions, leaving open the question of how benign overfitting can occur in fully non-linear settings. For grokking, several recent papers (Nanda et al., 2023; Gromov, 2023; Varma et al., 2023) have proposed explanations, but to the best of our knowledge, no prior work has established a rigorous proof of grokking in a neural network setting.

In this work, we characterize a setting in which both benign overfitting and grokking provably occur. We consider a two-layer ReLU network trained by gradient descent on a binary classification task

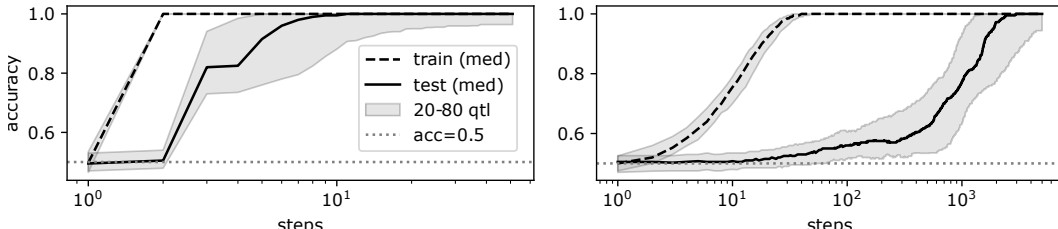

Figure 1: Comparing train and test accuracies of a two-layer neural network (2.1) trained on noisily labeled XOR data over 100 independent runs. *Left/right panel* shows benign overfitting and grokking when the step size is larger/smaller compared to the weight initialization scale. For plotting the x-axis, we add 1 to time so that the initialization $t = 0$ can be shown in log scale. See Appendix A.7 for details of the experimental setup.

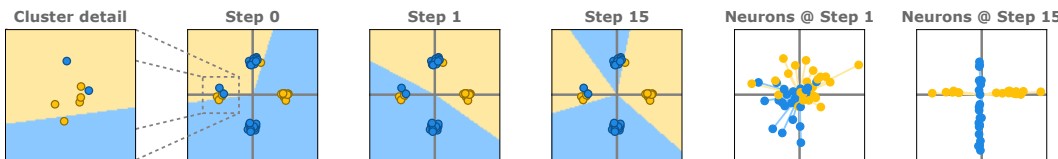

Figure 2: *Left four panels*: 2-dimensional projection of the noisily labeled XOR cluster data (Definition 2.1) and the decision boundary of the neural network (2.1) classifier restricted to the subspace spanned by the cluster means at times $t = 0, 1$ and 15. *Right two panels*: 2-dimensional projection of the neuron weights plotted at times $t = 1$ and 15.

defined by an XOR cluster data distribution (Figure 2). Specifically, datapoints from the positive class are drawn from a mixture of two high-dimensional Gaussian distributions $\frac{1}{2}N(\mu_1, I) + \frac{1}{2}N(-\mu_1, I)$, and datapoints from the negative class are drawn from $\frac{1}{2}N(\mu_2, I) + \frac{1}{2}N(-\mu_2, I)$, where $\mu_1$ and $\mu_2$ are orthogonal vectors. We then allow a constant fraction of the labels to be flipped. In this setting, we rigorously prove the following results: (i) **One-step catastrophic overfitting:** After one gradient descent step, the network perfectly fits every single training datapoint (no matter if it has a clean or flipped label), but has test accuracy close to $50\%$, performing no better than random guessing. (ii) **Grokking and benign overfitting:** After training for more steps, the network undergoes a "grokking" period from catastrophic to benign overfitting—it eventually reaches near $100\%$ test accuracy, while maintaining $100\%$ training accuracy the whole time. This behavior can be seen in Figure 1, where we also see that with a smaller step size the same grokking phenomenon occurs but with a delayed time for both overfitting and generalization.

Our results provide the first theoretical characterization of benign overfitting in a truly non-linear setting involving training a neural network on a non-linearly separable distribution. Interestingly, prior work on benign overfitting in neural networks for linearly separable distributions (Frei et al., 2022b; Cao et al., 2022; Xu & Gu, 2023; Kou et al., 2023) have not shown a time separation between catastrophic overfitting and generalization, which suggests that the XOR cluster data setting is fundamentally different.

Our proofs rely on analyzing the feature learning behavior of individual neurons over the gradient descent trajectory. After one training step, we prove that the network approximately implements a linear classifier over the underlying data distribution, which is able to overfit all the training datapoints but unable to generalize. Upon further training, the neurons gradually align with the core features $\pm\mu_1$ and $\pm\mu_2$, which is sufficient for generalization. See Figure 2 for visualizations of the network's decision boundary and neuron weights at different time steps, which confirm our theory.

### 1.1 ADDITONAL RELATED WORK

**Benign overfitting.** The literature on benign overfitting (also known as harmless interpolation) is now immense; for a general overview, we refer the readers to the surveys Bartlett et al. (2021); Belkin (2021); Dar et al. (2021). We focus here on those works on benign overfitting in neural networks. Frei et al. (2022b) showed that two-layer networks with smooth leaky ReLU activations

trained by gradient descent (GD) exhibit benign overfitting when trained on a high-dimensional binary cluster distribution. Xu & Gu (2023) extended their results to more general activations like ReLU. Cao et al. (2022) showed that two-layer convolutional networks with polynomial-ReLU activations trained by GD exhibit benign overfitting for image-patch data; Kou et al. (2023) extended their results to allow for label-flipping noise and standard ReLU activations. Each of these works used a trajectory-based analysis and none of them identified a grokking phenomenon. Frei et al. (2023a); Kornowski et al. (2023) showed how stationary points of margin-maximization problems associated with homogeneous neural network training problems can exhibit benign overfitting. Finally, Mallinar et al. (2022) proposed a taxonomy of overfitting behaviors in neural networks, whereby overfitting is "catastrophic" if test-time performance is comparable to a random guess, "benign" if it is near-optimal, and "tempered" if it lies between catastrophic and benign.

**Grokking.**   The phenomenon of grokking was first identified by Power et al. (2022) in decoder-only transformers trained on algorithmic datasets. Liu et al. (2022) provided an effective theory of representation learning to understand grokking. Thilak et al. (2022) attributed grokking to the slingshot mechanism, which can be measured by the cyclic phase transitions between stable and unstable training regimes. Žunkovič & Ilievski (2022) showed a time separation between achieving zero training error and zero test error in a binary classification task on a linearly separable distribution. Liu et al. (2023) identified a large initialization scale together with weight decay as a mechanism for grokking. Barak et al. (2022); Nanda et al. (2023) proposed progress metrics to measure the progress towards generalization during training. Davies et al. (2023) hypothesized a pattern-learning model for grokking and first reported a model-wise grokking phenomenon. Merrill et al. (2023) studied the learning dynamics in a two-layer neural network on a sparse parity task, attributing grokking to the competition between dense and sparse subnetworks. Varma et al. (2023) utilized circuit efficiency to interpret grokking and discovered two novel phenomena called ungrokking and semi-grokking.

**Feature learning for XOR distributions.**   The behavior of neural networks trained on the XOR cluster distribution we consider here, or its variants like the sparse parity problem, have been extensively studied in recent years. Wei et al. (2019) showed that neural networks in the mean-field regime, where neural networks can learn features, have better sample complexity guarantees than neural networks in the neural tangent kernel (NTK) regime in this setting. Barak et al. (2022); Telgarsky (2023) examined the sample complexity of learning sparse parities on the hypercube for neural networks trained by SGD. Most related to this work, Frei et al. (2022a) characterized the dynamics of GD in ReLU networks in the same distributional setting we consider here, namely the XOR cluster with label-flipping noise. They showed that by early-stopping, the neural network achieves perfect (clean) test accuracy although the training error is close to the label noise rate; in particular, their network achieved optimal generalization *without* overfitting, which is fundamentally different from our result. By contrast, we show that the network first exhibits catastrophic overfitting before transitioning to benign overfitting later in training.[1]

## 2   PRELIMINARIES

### 2.1   NOTATION

For a vector $x$, denote its Euclidean norm by $\|x\|$. For a matrix $X$, denote its Frobenius norm by $\|X\|_F$ and its spectral norm by $\|X\|$. Denote the indicator function by $\mathbb{I}(\cdot)$. Denote the sign of a scalar $x$ by $\mathrm{sgn}(x)$. Denote the cosine similarity of two vectors $u, v$ by $\mathrm{cossim}(u, v) := \frac{\langle u,v \rangle}{\|u\|\|v\|}$. Denote a multivariate Gaussian distribution with mean vector $\mu$ and covariance matrix $\Sigma$ by $N(\mu, \Sigma)$. Denote by $\sum_j q_j N(\mu_j, \Sigma_j)$ a mixture of Gaussian distributions, namely, with probability $q_j$, the sample is generated from $N(\mu_j, \Sigma_j)$. Let $I_p$ be the $p \times p$ identity matrix. For a finite set $\mathcal{A} = \{a_i\}_{i=1}^n$, denote the uniform distribution on $\mathcal{A}$ by $\mathrm{Unif}.\mathcal{A}$. For a random variable $X$, denote its expectation by $\mathbb{E}[X]$. For an integer $d \geq 1$, denote the set $\{1, \cdots, d\}$ by $[d]$. For a finite set $\mathcal{A}$, let $|\mathcal{A}|$ be its cardinality. We use $\{\pm\mu\}$ to represent the set $\{+\mu, -\mu\}$. For two positive sequences $\{x_n\}, \{y_n\}$, we say $x_n = O(y_n)$ (respectively $x_n = \Omega(y_n)$), if there exists a universal constant $C > 0$ such

---

[1]The reason for the different behaviors between our work and Frei et al. (2022a) is because they work in a setting with a larger signal-to-noise ratio (i.e., the norm of the cluster means is larger than the one we consider).

that $x_n \leq C y_n$ (respectively $x_n \geq C y_n$) for all $n$, and say $x_n = o(y_n)$ if $\lim_{n \to \infty} \frac{x_n}{y_n} = 0$. We say $x_n = \Theta(y_n)$ if $x_n = O(y_n)$ and $y_n = O(x_n)$.

## 2.2 DATA GENERATION SETTING

Let $\mu_1, \mu_2 \in \mathbb{R}^p$ be two orthogonal vectors, i.e. $\mu_1^\top \mu_2 = 0$.[2] Let $\eta \in [0, 1/2)$ be the label flipping probability.

**Definition 2.1** (XOR cluster data). Define $P_{\text{clean}}$ as the distribution over the space $\mathbb{R}^p \times \{\pm 1\}$ of labelled data such that a datapoint $(x, \widetilde{y}) \sim P_{\text{clean}}$ is generated according to the following procedure: First, sample the label $\widetilde{y} \sim \text{Unif}\{\pm 1\}$. Second, generate $x$ as follows:

(1) If $\widetilde{y} = 1$, then $x \sim \frac{1}{2} N(+\mu_1, I_p) + \frac{1}{2} N(-\mu_1, I_p)$;

(2) If $\widetilde{y} = -1$, then $x \sim \frac{1}{2} N(+\mu_2, I_p) + \frac{1}{2} N(-\mu_2, I_p)$.

Define $P$ to be the distribution over $\mathbb{R}^p \times \{\pm 1\}$ which is the $\eta$-noise-corrupted version of $P_{\text{clean}}$, namely: to generate a sample $(x, y) \sim P$, first generate $(x, \widetilde{y}) \sim P_{\text{clean}}$, and then let $y = \widetilde{y}$ with probability $1 - \eta$, and $y = -\widetilde{y}$ with probability $\eta$.

We consider $n$ training datapoints $\{(x_i, y_i)\}_{i=1}^n$ generated i.i.d from the distribution $P$. We assume the sample size $n$ to be sufficiently large (i.e., larger than any universal constant appearing in this paper). Note the $x_i$'s are from a mixture of four Gaussians centered at $\pm \mu_1$ and $\pm \mu_2$. We denote centers $:= \{\pm \mu_1, \pm \mu_2\}$ for convenience. For simplicity, we assume $\|\mu_1\| = \|\mu_2\|$, omit the subscripts and denote them by $\|\mu\|$.

## 2.3 NEURAL NETWORK, LOSS FUNCTION, AND TRAINING PROCEDURE

We consider a two-layer neural network of width $m$ of the form

$$f(x; W) := \sum_{j=1}^m a_j \phi(\langle w_j, x \rangle), \tag{2.1}$$

where $w_1, \ldots, w_m \in \mathbb{R}^p$ are the first-layer weights, $a_1, \ldots, a_m \in \mathbb{R}$ are the second-layer weights, and the activation $\phi(z) := \max\{0, z\}$ is the ReLU function. We denote $W = [w_1, \ldots, w_m] \in \mathbb{R}^{p \times m}$ and $a = [a_1, \ldots, a_m]^\top \in \mathbb{R}^m$. We assume the second-layer weights are sampled according to $a_j \overset{\text{i.i.d.}}{\sim} \text{Unif}\{\pm \frac{1}{\sqrt{m}}\}$ and are fixed during the training process.

We define the empirical risk using the logistic loss function $\ell(z) = \log(1 + \exp(-z))$: $\widehat{L}(W) := \frac{1}{n} \sum_{i=1}^n \ell(y_i f(x_i; W))$. We use gradient descent (GD) $W^{(t+1)} = W^{(t)} - \alpha \nabla \widehat{L}(W^{(t)})$ to update the first-layer weight matrix $W$, where $\alpha$ is the step size. Specifically, at time $t = 0$ we randomly initialize the weights by

$$w_j^{(0)} \overset{\text{i.i.d.}}{\sim} N(0, \omega_{\text{init}}^2 I_p), \quad j \in [m],$$

where $\omega_{\text{init}}^2$ is the initialization variance; at each time step $t = 0, 1, 2, \ldots$, the GD update can be calculated as

$$w_j^{(t+1)} - w_j^{(t)} = -\alpha \frac{\partial \widehat{L}(W^{(t)})}{\partial w_j} = \frac{\alpha a_j}{n} \sum_{i=1}^n g_i^{(t)} \phi'(\langle w_j^{(t)}, x_i \rangle) y_i x_i, \quad j \in [m], \tag{2.2}$$

where $g_i^{(t)} := -\ell'(y_i f(x_i; W^{(t)}))$.

## 3 MAIN RESULTS

Given a large enough universal constant $C$, we make the following assumptions:

---

[2] Our results hold when $\mu_1$ and $\mu_2$ are near-orthogonal. We assume exact orthogonality for ease of presentation.

(A1) The norm of the mean satisfies $\|\mu\|^2 \geq Cn^{0.51}\sqrt{p}$.

(A2) The dimension of the feature space satisfies $p \geq Cn^2\|\mu\|^2$.

(A3) The noise rate satisfies $\eta \leq 1/C$.

(A4) The step size satisfies $\alpha \leq 1/(Cnp)$.

(A5) The initialization variance satisfies $\omega_{\text{init}}\, nm^{3/2}p \leq \alpha\|\mu\|^2$.

(A6) The number of neurons satisfies $m \geq Cn^{0.02}$.

Assumption (A1) concerns the signal-to-noise ratio (SNR) in the distribution, where the order $0.51$ can be extended to any constant strictly larger than $\frac{1}{2}$. The assumption of high-dimensionality (A2) is important for enabling benign overfitting, and implies that the training datapoints are near-orthogonal. For a given $n$, these two assumptions are simultaneously satisfied if $\|\mu\| = \Theta(p^\beta)$ where $\beta \in \left(\frac{1}{4}, \frac{1}{2}\right)$ and $p$ is a sufficiently large polynomial in $n$. Assumption (A3) ensures that the label noise rate is at most a constant. While Assumption (A4) ensures the step size is small enough to allow for a variant of smoothness between different steps, Assumption (A5) ensures that the step size is large relative to the initialization scale so that the behavior of the network after a single step of GD is significantly different from that at random initialization. Assumption (A6) ensures the number of neurons is large enough to allow for concentration arguments at random initialization.

With these assumptions in place, we can state our main theorem which characterizes the training error and test error of the neural network at different times during the training trajectory.

**Theorem 3.1.** *Suppose that Assumptions (A1)-(A6) hold. With probability at least $1 - n^{-\Omega(1)} - O(1/\sqrt{m})$ over the random data generation and initialization of the weights, we have:*

- *The classifier $\operatorname{sgn}(f(x; W^{(t)}))$ can correctly classify all training datapoints for $1 \leq t \leq \sqrt{n}$:*

$$y_i = \operatorname{sgn}(f(x_i; W^{(t)})), \quad \forall i \in [n].$$

- *The classifier $\operatorname{sgn}(f(x; W^{(t)}))$ has near-random test error at $t = 1$:*

$$\tfrac{1}{2}(1 - n^{-\Omega(1)}) \leq \mathbb{P}_{(x,y) \sim P_{clean}}(y \neq \operatorname{sgn}(f(x; W^{(1)}))) \leq \tfrac{1}{2}(1 + n^{-\Omega(1)}).$$

- *The classifier $\operatorname{sgn}(f(x; W^{(t)}))$ generalizes when $Cn^{0.01} \leq t \leq \sqrt{n}$:*

$$\mathbb{P}_{(x,y) \sim P_{clean}}(y \neq \operatorname{sgn}(f(x; W^{(t)}))) \leq \exp(-\Omega(n^{0.99}\|\mu\|^4/p)) = \exp(-\Omega(n^{2.01})).$$

Theorem 3.1 shows that at time $t = 1$, the network achieves 100% training accuracy despite the constant fraction of flipped labels in the training data. The second part of the theorem shows that this overfitting is catastrophic as the test error is close to that of a random guess. On the other hand, by the first and third parts of the theorem, as long as the time step $t$ satisfies $Cn^{0.01} \leq t \leq \sqrt{n}$, the network continues to overfit to the training data while simultaneously achieving test error $\exp(-\Omega(n^{2.01}))$, which guarantees a near-zero test error for large $n$. In particular, the network exhibits benign overfitting, and it achieves this by grokking. Notably, Theorem 3.1 is the first guarantee for benign overfitting in neural network classification for a nonlinear data distribution, in contrast to prior works which required linearly separable distributions (Frei et al., 2022b; 2023a; Cao et al., 2022; Xu & Gu, 2023; Kou et al., 2023; Kornowski et al., 2023).

We note that Theorem 3.1 requires an upper bound on the number of iterations of gradient descent, i.e. it does not provide a guarantee as $t \to \infty$. At a technical level, this is needed so that we can guarantee that the ratio of the sigmoid losses between all samples $r(t) := \max_{i,j \in [n]} \dfrac{g_i^{(t)}}{g_j^{(t)}}$ is close to 1, and we show that this holds if $t \leq \sqrt{n}$. This property prevents the training data with flipped labels from having an out-sized influence on the feature learning dynamics. Prior works in other settings have shown that $r(t)$ is at most a large constant for any step $t$ for a similar purpose (Frei et al., 2022b; Xu & Gu, 2023), however the dynamics of learning in the XOR setting are more intricate and require a tighter bound on $r(t)$. We leave the question of generalizing our results to longer training times for future work. In Section 4, we provide an overview of the key ingredients to the proof of Theorem 3.1.

# 4 PROOF SKETCH

We first introduce some additional notation. For $i \in [n]$, let $\bar{x}_i \in \text{centers} = \{\pm\mu_1, \pm\mu_2\}$ be the mean of the Gaussian from which the sample $(x_i, y_i)$ is drawn. For each $\nu \in \text{centers}$, define $\mathcal{I}_\nu = \{i \in [n] : \bar{x}_i = \nu\}$, i.e., the set of indices $i$ such that $x_i$ belongs to the cluster centered at $\nu$. Thus, $\{\mathcal{I}_\nu\}_{\nu \in \text{centers}}$ is a partition of $[n]$. Moreover, define $\mathcal{C} = \{i \in [n] : y_i = \widetilde{y}_i\}$ and $\mathcal{N} = \{i \in [n] : y_i \neq \widetilde{y}_i\}$ to be the set of clean and noisy samples, respectively. Further we define for each $\nu \in \text{centers}$ the following sets:

$$\mathcal{C}_\nu := \mathcal{C} \cap \mathcal{I}_\nu \quad \text{and} \quad \mathcal{N}_\nu := \mathcal{N} \cap \mathcal{I}_\nu.$$

Let $c_\nu = |\mathcal{C}_\nu|$ and $n_\nu = |\mathcal{N}_\nu|$. Define the training input data matrix $X = [x_1, \ldots, x_n]^\top$. Let $\varepsilon \in (0, 10^{-3}/4)$ be a universal constant.

In Section 4.1, we present several properties satisfied with high probability by the training data and random initialization, which are crucial in our proof. In Section 4.2, we outline the major steps in the proof of Theorem 3.1.

## 4.1 PROPERTIES OF THE TRAINING DATA AND RANDOM INITIALIZATION

**Lemma 4.1** (Properties of training data). *Suppose Assumptions (A1) and (A2) hold. Let the training data $\{(x_i, y_i)\}_{i=1}^n$ be sampled i.i.d from $P$ as in Definition 2.1. With probability at least $1 - O(n^{-\varepsilon})$ the training data satisfy properties (B1)-(B4) defined below.*

*(B1) For all $k \in [n]$, $\max_{\nu \in \text{centers}} \langle x_k - \bar{x}_k, \nu \rangle \leq 10\sqrt{\log n}\|\mu\|$ and $|\|x_k\|^2 - p - \|\mu\|^2| \leq 10\sqrt{p \log n}$.*

*(B2) For each $i, k \in [n]$ such that $i \neq k$, we have $|\langle x_i, x_k \rangle - \langle \bar{x}_i, \bar{x}_k \rangle| \leq 10\sqrt{p \log n}$.*

*(B3) For $\nu \in \text{centers}$, we have $|c_\nu + n_\nu - n/4| \leq \sqrt{\varepsilon n \log n}$ and $|n_\nu - \eta(c_\nu + n_\nu)| \leq \sqrt{\varepsilon \eta n \log n}$.*

*(B4) For $\nu \in \text{centers}$, we have $|c_\nu + n_\nu - c_{-\nu} - n_{-\nu}| \geq n^{1/2-\varepsilon}$ and $|n_\nu - n_{-\nu}| \geq \eta n^{1/2-\varepsilon}$.*

*Denote by $\mathcal{G}_{data}$ the set of training data satisfying conditions (B1)-(B4). Thus, the result can be stated succinctly as $\mathbb{P}(X \in \mathcal{G}_{data}) \geq 1 - O(n^{-\varepsilon})$.*

The proof of Lemma 4.1 can be found in Appendix A.2.1. Conditions (B1) and (B2) are essentially the same as Frei et al. (2022b, Lemma 4.3) or Chatterji & Long (2021b, Lemma 10). Conditions (B3) and (B4) concern the number of clean and noisy examples in each cluster, and can be proved by concentration and anti-concentration arguments, respectively.

Lemma 4.1 has an important corollary.

**Corollary 4.2** (Near-orthogonality of training data). *Suppose Assumptions (A1), (A2), and Conditions (B1), (B2) from Lemma 4.1 all hold. Then for all $1 \leq i \neq k \leq n$,*

$$|\text{cossim}(x_i, x_k)| \leq \frac{2}{Cn^2}.$$

This near-orthogonality comes from the high dimensionality of the feature space (i.e., Assumption (A2)) and will be crucially used throughout the proofs on optimization and generalization of the network. The proof of Corollary 4.2 can be found in Appendix A.2.1.

Next, we divide the neuron indices into two sets according to the sign of the corresponding second-layer weight:

$$\mathcal{J}_{\text{Pos}} := \{j \in [m] : a_j > 0\}; \quad \mathcal{J}_{\text{Neg}} := \{j \in [m] : a_j < 0\}.$$

We will conveniently call them positive and negative neurons. Our next lemma shows that some properties of the random initialization hold with a large probability. The proof details can be found in Appendix A.3.1.

**Lemma 4.3** (Properties of the random weight initialization). *Suppose Assumptions (A1), (A2) and (A6) hold. All conditions below simultaneously hold with probability at least $1 - O(n^{-\varepsilon})$ over the random initialization:*

*(C1)* $\left\| W^{(0)} \right\|_F^2 \leq \frac{3}{2} \omega_{init}^2 \, mp.$

*(C2)* $|\mathcal{J}_{\mathtt{Pos}}| \geq m/3$ *and* $|\mathcal{J}_{\mathtt{Neg}}| \geq m/3.$

*The result can be stated equivalently as follows: Denote the set of $W^{(0)}$ satisfying condition (C1) by $\mathcal{G}_W$. Denote the set of $a = (a_j)_{j=1}^m$ satisfying condition (C2) by $\mathcal{G}_A$. Then $\mathbb{P}(a \in \mathcal{G}_A, W^{(0)} \in \mathcal{G}_W) \geq 1 - O(n^{-\varepsilon}).$*

We say that the sample $i$ *activates* neuron $j$ at time $t$ if $\langle w_j^{(t)}, x_i \rangle > 0$. Now, for each neuron $j \in [m]$, time $t \geq 0$ and $\nu \in$ centers, define the set of indices $i$ of samples $x_i$ with clean (resp. noisy) labels from the cluster centered at $\nu$ that activates neuron $j$ at time $t$:

$$\mathcal{C}_{\nu,j}^{(t)} := \{i \in \mathcal{C}_\nu : \langle w_j^{(t)}, x_i \rangle > 0\} \quad (\text{resp. } \mathcal{N}_{\nu,j}^{(t)} := \{i \in \mathcal{N}_\nu : \langle w_j^{(t)}, x_i \rangle > 0\}). \qquad (4.1)$$

Moreover, we define

$$d_{\nu,j}^{(t)} := |\mathcal{C}_{\nu,j}^{(t)}| - |\mathcal{N}_{\nu,j}^{(t)}|, \quad \text{and} \quad D_{\nu,j}^{(t)} := d_{\nu,j}^{(t)} - d_{-\nu,j}^{(t)}.$$

For $\kappa \in [0, 1/2)$ and $\nu \in$ centers, a neuron $j$ is said to be $(\nu, \kappa)$-*aligned* if

$$D_{\nu,j}^{(0)} > n^{1/2-\kappa}, \quad \text{and} \quad \max\{d_{-\nu,j}^{(0)}, d_{\nu,j}^{(0)}\} < \min\{c_\nu, c_{-\nu}\} - 2(n_{+\nu} + n_{-\nu}) - \sqrt{n} \qquad (4.2)$$

The first condition ensures that at initialization, there are at least $n^{1/2-\kappa}$ many more samples from cluster $\nu$ activating the $j$-th neuron than from cluster $-\nu$ after accounting for cancellations from the noisy labels. The second is a technical condition necessary for trajectory analysis. A neuron $j$ is said to be $(\pm\nu, \kappa)$-*aligned* if it is either $(\nu, \kappa)$-aligned or $(-\nu, \kappa)$-aligned.

**Lemma 4.4** (Properties of the interaction between training data and initial weights)**.** *Suppose Assumptions (A1)-(A3) and (A6) hold. Given $a \in \mathcal{G}_A$ (defined in Lemma 4.3) and $X \in \mathcal{G}_{data}$ (defined in Lemma 4.1), the followings hold with probability at least $1 - O(n^{-\varepsilon})$ over the random initialization $W^{(0)}$:*

*(D1) For all $i \in [n]$, the sample $x_i$ activates a large proportion of positive and negative neurons, i.e., $|\{j \in \mathcal{J}_{\mathtt{Pos}} : \langle w_j^{(0)}, x_i \rangle > 0\}| \geq m/7$ and $|\{j \in \mathcal{J}_{\mathtt{Neg}} : \langle w_j^{(0)}, x_i \rangle > 0\}| \geq m/7$ both hold.*

*(D2) For all $\nu \in$ centers and $\kappa \in [0, \frac{1}{2})$, both $|\{j \in \mathcal{J}_{\mathtt{Pos}} : j \text{ is } (\nu, \kappa)\text{-aligned}\}| \geq mn^{-10\varepsilon}$, and $|\{j \in \mathcal{J}_{\mathtt{Neg}} : j \text{ is } (\nu, \kappa)\text{-aligned}\}| \geq mn^{-10\varepsilon}$.*

*(D3) For all $\nu \in$ centers, we have $\left|\{j \in \mathcal{J}_{\mathtt{Pos}} : j \text{ is } (\pm\nu, 20\varepsilon)\text{-aligned}\}\right| \geq (1 - 10n^{-20\varepsilon})|\mathcal{J}_{\mathtt{Pos}}|.$ Moreover, the same statement holds if "$\mathcal{J}_{\mathtt{Pos}}$" is replaced with "$\mathcal{J}_{\mathtt{Neg}}$" everywhere.*

*(D4) For all $\nu \in$ centers and $\kappa \in [0, \frac{1}{2})$, let $\mathcal{J}_{\nu,\mathtt{Pos}}^\kappa := \{j \in \mathcal{J}_{\mathtt{Pos}} : j \text{ is } (\nu, \kappa)\text{-aligned}\}$. Then $\sum_{j \in \mathcal{J}_{\nu,\mathtt{Pos}}^\kappa} (c_\nu - n_\nu - d_{-\nu,j}^{(0)}) \geq \frac{n}{10}|\mathcal{J}_{\nu,\mathtt{Pos}}^\kappa|$. Moreover, the same statement holds if "$\mathcal{J}_{\mathtt{Pos}}$" is replaced with "$\mathcal{J}_{\mathtt{Neg}}$" everywhere.*

Condition (D1) makes sure that the neurons spread uniformly at initialization so that each datapoint activates at least a constant fraction of positive and negative neurons. Condition (D2) guarantees that for each $\nu \in$ centers, there are a fraction of neurons aligning with $\nu$ more than $-\nu$. Condition (D3) shows that most neurons will somewhat align with either $\nu$ or $-\nu$. Condition (D4) is a technical concentration result. For proof details, see Appendix A.3.2.

Define the set $\mathcal{G}_{\text{good}}$ as

$$\mathcal{G}_{\text{good}} := \{(a, W^{(0)}, X) : a \in \mathcal{G}_A, X \in \mathcal{G}_{\text{data}}, W^{(0)} \in \mathcal{G}_W \text{ and conditions (D1)-(D4) hold}\},$$

whose probability is lower bounded by $\mathbb{P}((a, W^{(0)}, X) \in \mathcal{G}_{\text{good}}) \geq 1 - O(n^{-\varepsilon})$. This is a consequence of Lemmas 4.1, 4.3 and 4.4 (see Appendix A.3.3).

**Definition 4.5.** If the training data $X$ and the initialization $a, W^{(0)}$ belong to $\mathcal{G}_{\text{good}}$, we define this circumstance as a "good run."

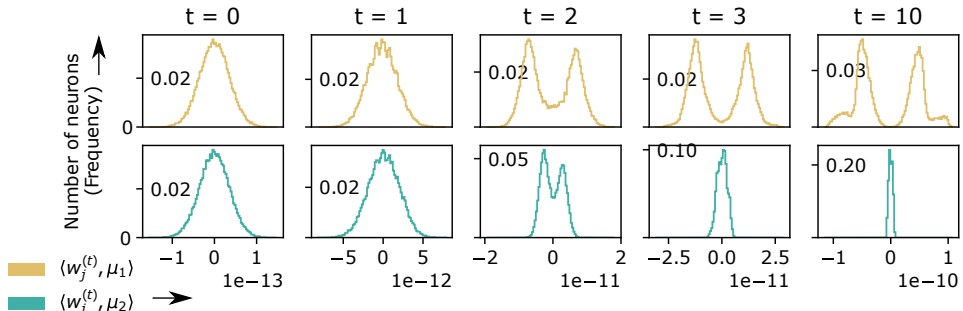

Figure 3: Histograms of inner products between positive neurons and $\mu_1$ or $\mu_2$ pooled over 100 independent runs under the same setting as in Figure 1. *Top (resp. bottom) row:* Inner products between positive neurons and $\mu_1$ (resp. $\mu_2$). While the distributions of the projections of positive neurons $w_j^{(t)}$ onto the $\mu_1$ and $\mu_2$ directions are nearly the same at times $t = 0, 1$, they become significantly more aligned with $\pm\mu_1$ over time. See Appendix A.7 for details of the experimental setup.

## 4.2 PROOF SKETCH FOR THEOREM 3.1

In order for the network to learn a generalizable solution for the XOR cluster distribution, we would like positive neurons' (i.e., those with $a_j > 0$) weights $w_j$ to align with $\pm\mu_1$, and negative neurons' weights to align with $\pm\mu_2$; we prove that this is satisfied for $t \in [Cn^{0.01}, \sqrt{n}]$. However, for $t = 1$, we show that the network only approximates a linear classifier, which can fit the training data in high dimension but has trivial test error. Figure 3 plots the evolution of the distribution of positive neurons' projections onto both $\mu_1$ and $\mu_2$, confirming that these neurons are much more aligned with $\pm\mu_1$ at a later training time, while they cannot distinguish $\pm\mu_1$ and $\pm\mu_2$ at $t = 1$.

Below we give a sketch of the proofs, and details are in Appendix A.5.

### 4.2.1 ONE-STEP CATASTROPHIC OVERFITTING

Under a good run, we have the following approximation for each neuron after the first iteration:

$$w_j^{(1)} \approx \frac{\alpha a_j}{2n} \sum_{i=1}^{n} \mathbb{I}(\langle w_j^{(0)}, x_i \rangle > 0) y_i x_i, \quad j \in [m].$$

For details of this approximation, see Appendix A.4.

Let $s_{ij} := \mathbb{I}(\langle w_j^{(0)}, x_i \rangle > 0)$. Then, for sufficiently large $m$, we can approximate the neural network output at $t = 1$ as

$$\sum_{j=1}^{m} a_j \phi(\langle w_j^{(1)}, x \rangle) \approx \frac{\alpha}{2n} \sum_{j=1}^{m} a_j \phi(a_j \langle \sum_{i=1}^{n} s_{ij} y_i x_i, x \rangle)$$

$$\overset{a.s.}{\to} \frac{\alpha}{4n} \langle \sum_{i=1}^{n} \mathbb{E}[s_{ij}] y_i x_i, x \rangle = \frac{\alpha}{8n} \langle \sum_{i=1}^{n} y_i x_i, x \rangle. \tag{4.3}$$

This convergence is ensured by the strong law of large numbers, given the independence of the first-layer and second-layer weights at initialization. This implies that the neural network classifier $\text{sgn}(f(\cdot; W^{(1)}))$ behaves similarly to the linear classifier $\text{sgn}(\langle \sum_{i=1}^{n} y_i x_i, \cdot \rangle)$. It can be shown that this linear classifier achieves 100% training accuracy whenever the training data are near orthogonal (Frei et al., 2023b, Appendix D), but because each class has two clusters with opposing means, linear classifiers only achieve 50% test error for the XOR cluster distribution. Thus at time $t = 1$, the network is able to fit the training data but is not capable of generalizing.

### 4.2.2 Multi-Step Generalization

Next, we show that positive (resp. negative) neurons gradually align with one of $\pm\mu_1$ (resp. $\pm\mu_2$), and forget both of $\pm\mu_2$ (resp. $\pm\mu_1$), making the network generalizable. Taking the direction $+\mu_1$ as an example, we define sets of neurons

$$\mathcal{J}_1 = \{j \in \mathcal{J}_{\text{Pos}} : j \text{ is } (+\mu_1, 20\varepsilon)\text{-aligned}\}; \quad \mathcal{J}_2 = \{j \in \mathcal{J}_{\text{Neg}} : j \text{ is } (\pm\mu_1, 20\varepsilon)\text{-aligned}\}.$$

We have by conditions (D2)-(D3) of Lemma 4.4 that under a good run,

$$|\mathcal{J}_1| \geq mn^{-10\varepsilon}, \quad |\mathcal{J}_2| \geq (1 - 10n^{-20\varepsilon})|\mathcal{J}_{\text{Neg}}|,$$

which implies that $\mathcal{J}_1$ contains a certain proportion of $\mathcal{J}_{\text{Pos}}$ and $\mathcal{J}_2$ covers most of $\mathcal{J}_{\text{Neg}}$. The next lemma shows that neurons in $\mathcal{J}_1$ will keep aligning with $+\mu_1$, but neurons in $\mathcal{J}_2$ will gradually forget $+\mu_1$.

**Lemma 4.6.** *Suppose that Assumptions (A1)-(A6) hold. Under a good run, we have that for $1 \leq t \leq \sqrt{n}$,*

$$\frac{1}{|\mathcal{J}_1|} \sum_{j \in \mathcal{J}_1} \langle w_j^{(t)}, +\mu_1 \rangle = \Omega\left(\frac{\alpha\|\mu\|^2}{\sqrt{m}}t\right);$$

$$\frac{1}{|\mathcal{J}_2|} \sum_{j \in \mathcal{J}_2} |\langle w_j^{(t)}, \mu_1 \rangle| = O\left(\frac{\alpha\|\mu\|^2}{\sqrt{m}} + \frac{\alpha\|\mu\|^2\sqrt{\log(n)}}{\sqrt{mn}}t\right).$$

We can see that when $t$ is large, $\sum_{j \in \mathcal{J}_2} |\langle w_j^{(t)}, \mu_1 \rangle|/|\mathcal{J}_2| = o(\sum_{j \in \mathcal{J}_1} \langle w_j^{(t)}, +\mu_1 \rangle/|\mathcal{J}_1|)$, thus for $x \sim N(+\mu_1, I_p)$, neurons with $j \in \mathcal{J}_1$ will dominate the output of $f(x; W^{(t)})$. For the other three clusters centered at $-\mu_1, +\mu_2, -\mu_2$ we have similar results, which then lead the model to generalization. Formally, we have the following theorem on generalization.

**Theorem 4.7.** *Suppose that Assumptions (A1)-(A6) hold. Under a good run, for $Cn^{10\varepsilon} \leq t \leq \sqrt{n}$, the generalization error of classifier $\text{sgn}(f(x, W^{(t)}))$ has an upper bound*

$$\mathbb{P}_{(x,y)\sim P_{clean}}(y \neq \text{sgn}(f(x; W^{(t)}))) \leq \exp\left(-\Omega\left(\frac{n^{1-20\varepsilon}\|\mu\|^4}{p}\right)\right).$$

## 5 Discussion

We have shown that two-layer neural networks trained on XOR cluster data with random label noise by GD reveal a number of interesting phenomena. First, early in training, the network interpolates all of the training data but fails to generalize to test data better than random chance, displaying a familiar form of (catastrophic) overfitting. Later in training, the network continues to achieve a perfect fit to the noisy training data but groks useful features so that it can achieve near-zero error on test data, thus exhibiting both grokking and benign overfitting simultaneously. Notably, this provides an example of benign overfitting in neural network classification for a distribution which is not linearly separable.

In contrast to prior works on grokking which found the usage of weight decay to be crucial for grokking (Liu et al., 2022; 2023), we observe grokking without any explicit forms of regularization, revealing the significance of the implicit regularization of GD. In our setting, the catastrophic overfitting stage of grokking occurs because early in training, the network behaves similarly to a linear classifier. This linear classifier is capable of fitting the training data due to the high-dimensionality of the feature space but fails to generalize as linear classifiers are not complex enough to achieve test performance above random chance for the XOR cluster. Later in training, the network groks useful features, corresponding to the cluster means, which allow for good generalization.

There are a few natural questions for future research. First, our analysis requires an upper bound on the number of training steps due to technical reasons; it is intriguing to understand the generalization behavior as time grows to infinity. Second, our proof crucially relies upon the assumption that the training data are nearly-orthogonal which requires that the ambient dimension is large relative to the number of samples. Prior work has shown with experiments that overfitting is less benign in this setting when the dimension is small relative to the number of samples (Frei et al., 2022a, Fig. 2); a precise characterization of the effect of high-dimensional data on generalization remains open.

ACKNOWLEDGMENTS

YW acknowledges support from the Eric and Wendy Schmidt AI in Science Postdoctoral Fellowship, a Schmidt Futures program. GV acknowledges support from the NSF and the Simons Foundation via the Collaboration on the Theoretical Foundations of Deep Learning. WH acknowledges support from the Google Research Scholar program.

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

# A Appendix organization

## A.1 Additional Notation

Denote the c.d.f of standard normal distribution by $\Phi(\cdot)$ and the p.d.f. of standard normal distribution by $\Phi'(\cdot)$. Denote $\bar{\Phi}(\cdot) = 1 - \Phi(\cdot)$. Denote the Bernoulli distribution which takes 1 with probability $p \in (0, 1)$ by $\text{Bern}(p)$. Denote the Binomial distribution with size $n$ and probability $p$ by $\text{B}(n, p)$. For a random variable $X$, denote its variance by $\text{Var}(X)$; and its absolute third central moment by $\rho(X)$.

## A.2 Properties of the training data

### A.2.1 Proof of Lemma 4.1

**Lemma 4.1** (Properties of training data). *Suppose Assumptions (A1) and (A2) hold. Let the training data $\{(x_i, y_i)\}_{i=1}^n$ be sampled i.i.d from $P$ as in Definition 2.1. With probability at least $1 - O(n^{-\varepsilon})$ the training data satisfy properties (B1)-(B4) defined below.*

*(B1) For all $k \in [n]$, $\max_{\nu \in \text{centers}} \langle x_k - \bar{x}_k, \nu \rangle \le 10\sqrt{\log n}\|\mu\|$ and $|\|x_k\|^2 - p - \|\mu\|^2| \le 10\sqrt{p \log n}$.*

*(B2) For each $i, k \in [n]$ such that $i \ne k$, we have $|\langle x_i, x_k \rangle - \langle \bar{x}_i, \bar{x}_k \rangle| \le 10\sqrt{p \log n}$.*

*(B3) For $\nu \in \text{centers}$, we have $|c_\nu + n_\nu - n/4| \le \sqrt{\varepsilon n \log n}$ and $|n_\nu - \eta(c_\nu + n_\nu)| \le \sqrt{\varepsilon \eta n \log n}$.*

*(B4)* For $\nu \in$ centers, we have $|c_\nu + n_\nu - c_{-\nu} - n_{-\nu}| \geq n^{1/2-\varepsilon}$ and $|n_\nu - n_{-\nu}| \geq \eta n^{1/2-\varepsilon}$.

Denote by $\mathcal{G}_{data}$ the set of training data satisfying conditions (B1)-(B4). Thus, the result can be stated succinctly as $\mathbb{P}(X \in \mathcal{G}_{data}) \geq 1 - O(n^{-\varepsilon})$.

*Proof.* Before proceeding with the proof, we recall that centers $= \{\pm\mu_1, \pm\mu_2\}$. We first show that (B1) holds with large probability. To this end, fix $k \in [n]$. We have by the construction of $x_k$ in Section 2.2 that $x_k \sim N(\bar{x}_k, I_p)$ for some $\bar{x}_k \in \{\pm\mu_1, \pm\mu_2\}$. Let $\xi_k = x_k - \bar{x}_k$. By Lemma A.17, we have

$$\mathbb{P}\left(\|\xi_k\| > \sqrt{p(t+1)}\right) \leq \mathbb{P}\left(\left|\|\xi_k\|^2 - p\right| > pt\right) \leq 2\exp(-pt^2/8), \quad \forall t \in (0,1). \tag{A.1}$$

Note that for any fixed non-zero vector $\nu \in \mathbb{R}^p$, we have $\langle\nu, \xi_k\rangle \sim N(0, \|\nu\|^2)$. Therefore, again by Lemma A.17, we have

$$\mathbb{P}(|\langle\nu, \xi_k\rangle| > t\|\nu\|) \leq \exp(-t^2/2), \quad \forall t \geq 1 \tag{A.2}$$

where the parameter $t$ in both inequality will be chosen later. To show that the first inequality of (B1) holds w.h.p, we show the complement event $\mathcal{F}_k := \{\max_{\nu\in\text{centers}}\langle\xi_k, \nu\rangle > t\|\mu\|\}$ has low probability. Applying the union bound,

$$\mathbb{P}(\mathcal{F}_k) \leq \sum_{\nu\in\{\pm\mu_1,\pm\mu_2\}} \mathbb{P}(|\langle\xi_k, \nu\rangle| > t\|\mu\|) \quad \because \text{Union bound}$$

$$\leq 4\exp(-t^2/2) \quad \because \text{Inequality (A.2)}.$$

Let $\delta := n^{-\varepsilon}$. Picking $t = \sqrt{2\log(16n/\delta)}$ in inequality (A.2) and applying the union bound again, we have

$$\mathbb{P}\left(\bigcup_{k=1}^{n} \mathcal{F}_k\right) \leq 4n\exp(-t^2/2) \leq \delta/4. \tag{A.3}$$

Next, fix $t_1 \in (0,1)$ and $t_2 \geq 1$ arbitrary. To show that the second inequality of (B1) holds w.h.p, we first prove an intermediate step: the complement event $\mathcal{E}_k := \{\left|\|x_k\|^2 - p - \|\mu\|^2\right| > pt_1 + 2\|\mu\|t_2\}$ has low probability. Towards this, first note that since

$$\|x_k\|^2 = \|\bar{x}_k\|^2 + \|\xi_k\|^2 + 2\langle\bar{x}_k, \xi_k\rangle = \|\mu\|^2 + \|\xi_k\|^2 + 2\langle\bar{x}_k, \xi_k\rangle$$

we have the alternative characterization of $\mathcal{E}_k$ as

$$\mathcal{E}_k = \{\left|\|\xi_k\|^2 - p + 2\langle\bar{x}_k, \xi_k\rangle\right| > pt_1 + 2\|\mu\|t_2\}.$$

Next, recall the fact: if $X, Y \in \mathbb{R}$ are random variables and $a, b \in \mathbb{R}$ are constants, then

$$\mathbb{P}(|X + Y| > a + b) \leq \mathbb{P}(|X| > a) + \mathbb{P}(|Y| > b). \tag{A.4}$$

To see this, first note that $|X + Y| \leq |X| + |Y|$ by the triangle inequality. From this we deduce that $\mathbb{P}(|X + Y| > a + b) \leq \mathbb{P}(|X| + |Y| > a + b)$. Now, by the union bound, we have

$$\mathbb{P}(|X| + |Y| > a + b) \leq \mathbb{P}(\{|X| > a\} \cup \{|Y| > b\}) \leq \mathbb{P}(|X| > a) + \mathbb{P}(|Y| > b)$$

which proves (A.4). Now, to upper bound $\mathbb{P}(\mathcal{E}_k)$, note that

$$\begin{aligned}
\mathbb{P}(\mathcal{E}_k) &= \mathbb{P}(\left|\|\xi_k\|^2 - p + 2\langle\bar{x}_k, \xi_k\rangle\right| > pt_1 + 2\|\mu\|t_2) \\
&\leq \mathbb{P}\left(\left|\|\xi_k\|^2 - p\right| > pt_1\right) + \mathbb{P}(|\langle\bar{x}_k, \xi_k\rangle| > t_2\|\mu\|) \quad \because \text{Inequality (A.4)} \\
&\leq 2\exp(-pt_1^2/8) + \exp(-t_2^2/2). \quad \because \text{Inequalities (A.1) and (A.2)} \tag{A.5}
\end{aligned}$$

Inequality (A.5) is the crucial intermediate step to proving the second inequality of (B1). It will be convenient to complete the proof of the second inequality of (B1) simultaneously with that of (B2). To this end, we next prove an analogous intermediate step to (B2).

Fix $s_1, s_2 \geq 1$ to be chosen later. Define the event $\mathcal{E}_{ij} := \{|\langle x_i, x_j\rangle - \langle\bar{x}_i, \bar{x}_j\rangle| > s_1\sqrt{p} + 2t_2\|\mu\|\}$ for each pair $i, j \in [n]$ such that $1 \leq i \neq j \leq n$. We upper bound $\mathbb{P}(\mathcal{E}_{ij})$ in similar fashion as in (A.5). To this end, fix $i, j \in [n]$ such that $i \neq j$. Note that the identity $\langle x_i, x_j\rangle = \xi_i^\top\xi_j + \bar{x}_i^\top\bar{x}_j + \xi_i^\top\bar{x}_j + \xi_j^\top\bar{x}_i$

implies that $|\langle x_i, x_j \rangle - \langle \bar{x}_i, \bar{x}_j \rangle| = |\xi_i^\top \xi_j + \xi_i^\top \bar{x}_j + \xi_j^\top \bar{x}_i|$. Now, we claim that

$$
\begin{aligned}
\mathbb{P}(\mathcal{E}_{ij}) &= \mathbb{P}(|\xi_i^\top \xi_j + \xi_i^\top \bar{x}_j + \xi_j^\top \bar{x}_i| \geq s_1 \sqrt{p} + 2t_2 \|\mu\|) \\
&\leq \mathbb{P}(|\xi_i^\top \xi_j| > s_1 \sqrt{p}) + \mathbb{P}(|\xi_i^\top \bar{x}_j| > t_2 \|\mu\|) + \mathbb{P}(|\xi_j^\top \bar{x}_i| > t_2 \|\mu\|) \\
&\leq \exp(-s_1^2/2s_2) + 2\exp(-p(s_2-1)^2/8) + 2\exp(-t_2^2/2), \quad\quad\quad (\text{A.6})
\end{aligned}
$$

The first inequality simply follows from applying (A.4) twice. Moreover, $\mathbb{P}(|\xi_i^\top \bar{x}_j| > t_2 \|\mu\|)$ and $\mathbb{P}(|\xi_j^\top \bar{x}_i| > t_2 \|\mu\|) \leq \exp(-t_2^2/2)$ follows from (A.2). To prove the claim, it remains to prove

$$
\begin{aligned}
&\mathbb{P}(|\langle \xi_i, \xi_j \rangle| > s_1 \sqrt{p}) \\
&\leq \mathbb{P}\big(|\langle \xi_i, \xi_j \rangle| > s_1 \sqrt{p} \mid \|\xi_j\| \leq \sqrt{s_2 p}\big) + \mathbb{P}(\|\xi_j\| > \sqrt{s_2 p}) \quad \because \text{law of total expectation} \\
&\leq \exp(-s_1^2/2s_2) + 2\exp(-p(s_2-1)^2/8). \quad\quad\quad\quad\quad\quad\quad\quad\quad\quad\quad\quad (\text{A.7})
\end{aligned}
$$

To prove the inequality at (A.7), first we get $\mathbb{P}(\|\xi_j\| > \sqrt{s_2 p}) \leq 2\exp(-p(s_2-1)^2/8)$ by applying (A.1) to upper bounds the second summand of the left-hand side of (A.7). For upper bounding the first summand, first let $\mathbb{P}\big(|\langle \xi_i, \xi_j \rangle| > s_1 \sqrt{p} \mid \xi_j\big)$ be the conditional probability conditioned on a realization of $\xi_j$ (while $\xi_i$ remains random). Then by definition

$$
\mathbb{P}\big(|\langle \xi_i, \xi_j \rangle| > s_1 \sqrt{p} \mid \|\xi_j\| \leq \sqrt{s_2 p}\big) = \mathbb{E}_{\xi_j}[\mathbb{P}\big(|\langle \xi_i, \xi_j \rangle| > s_1 \sqrt{p} \mid \xi_j\big) \mid \|\xi_j\| \leq \sqrt{s_2 p}]. \quad (\text{A.8})
$$

For fixed $\xi_j$ such that $\|\xi_j\| \leq \sqrt{s_2 p}$, we have by (A.2) that

$$
\mathbb{P}\big(|\langle \xi_i, \xi_j \rangle| > s_1 \sqrt{p} \mid \xi_j\big) = \mathbb{P}\big(|\langle \xi_i, \xi_j \rangle| > \|\xi_j\|(s_1 \sqrt{p}/\|\xi_j\|) \mid \xi_j\big) \leq \exp(-(s_1 \sqrt{p}/\|\xi_j\|)^2/2).
$$

Continue to assume fixed $\xi_j$ such that $\|\xi_j\| \leq \sqrt{s_2 p}$, note that $s_1 \sqrt{p}/\|\xi_j\| \geq s_1 \sqrt{p}/\sqrt{s_2 p} = s_1/\sqrt{s_2}$ implies

$$
\exp(-(s_1 \sqrt{p}/\|\xi_j\|)^2/2) \leq \exp(-(s_1/\sqrt{s_2})^2/2).
$$

Hence, $\mathbb{P}\big(|\langle \xi_i, \xi_j \rangle| > s_1 \sqrt{p} \mid \xi_j\big) \leq \exp(-s_1^2/2s_2)$. Applying $\mathbb{E}_{\xi_j}[\ \cdot\ \mid \|\xi_j\| \leq \sqrt{s_2 p}]$ to both side of the preceding inequality, we get $\mathbb{P}\big(|\langle \xi_i, \xi_j \rangle| > s_1 \sqrt{p} \mid \|\xi_j\| \leq \sqrt{s_2 p}\big) \leq \exp(-s_1^2/2s_2)$ which upper bounds the first summand of the left-hand side of (A.7). We now choose the values for $t_1 = \sqrt{8 \log(16n/\delta)/p}$, $t_2 = \sqrt{2 \log(16n^2/\delta)}$, $s_1 = 2\sqrt{\log(8n^2/\delta)}$, and $s_2 = 1 + \sqrt{8 \log(16n^2/\delta)/p}$. Recall that $\delta = n^{-\varepsilon}$ and $n$ is sufficiently large, then we have

$$
\sqrt{\log(16n^2/\delta)/p} = \sqrt{\log(16n^{2+\varepsilon})/p} \leq \sqrt{3 \log(16n)/p} \leq 1
$$

by Assumptions (A1) and (A2). Combining (A.5) and (A.6) then applying the union bound, we have

$$
\begin{aligned}
&\mathbb{P}((\cup_{k=1}^n \mathcal{E}_k) \cup (\cup_{i,j\in[n]:i\neq j}\mathcal{E}_{ij})) \leq \sum_{k=1}^n \mathbb{P}(\mathcal{E}_k) + \sum_{i,j\in[n]:i\neq j} \mathbb{P}(\mathcal{E}_{ij}) \\
&\leq 2n\exp(-\tfrac{pt_1^2}{8}) + n^2[2\exp(-\tfrac{t_2^2}{2}) + \exp(-\tfrac{s_1^2}{2s_2}) + 2\exp(-\tfrac{p(s_2-1)^2}{8})] \leq \delta.
\end{aligned} \quad (\text{A.9})
$$

Moreover, plugging the above values of $t_1$, $t_2$ and $s_1$ into the definition of $\mathcal{E}_k$ and $\mathcal{E}_{ij}$, we see that (B1) and (B2) are satisfied since they contain the complement of the event in (A.9).

Next, show that (B3) holds with large probability. We prove the inequality involving $|c_\nu + n_\nu - n/4|$ portion of (B3). Proofs for the rest of the inequalities in (B3) follow analogously using the same technique below. Recall from the data generation model, for each $k \in [n]$, $\bar{x}_k$ is sampled i.i.d $\sim$ Unif$\{\pm\mu_1, \pm\mu_2\}$. Define the following indicator random variable:

$$
\mathbb{I}_\nu(k) = \begin{cases} 1 & \text{if } \bar{x}_k = \nu \\ 0 & \text{otherwise,} \end{cases} \quad \text{for each } k \in [n], \text{ and } \nu \in \{\pm\mu_1, \pm\mu_2\}
$$

Then we have $\sum_\nu \mathbb{I}_\mu(k) = 1$ for each $k$, and $\mathbb{E}[\mathbb{I}_\nu(k)] = n/4$ for each $\nu$. Applying Hoeffding's inequality (Lemma A.18), we obtain

$$
\mathbb{P}(|\sum_{k=1}^n \mathbb{I}_\nu(k) - n/4| > t\sqrt{n}) \leq 2\exp(-2t^2).
$$

Applying the union bound, we have

$$
\mathbb{P}(\max_\nu |\sum_{k=1}^n \mathbb{I}_\nu(k) - n/4| > t\sqrt{n}) \leq 8\exp(-2t^2). \quad\quad\quad (\text{A.10})
$$

Thus we can bound the above tail probability by $O(\delta)$ by letting $t = \sqrt{\log(1/\delta)/2}$, and the upper bound $t\sqrt{n} \leq \sqrt{n \log(1/\delta)} = \sqrt{n\varepsilon \log(n)}$.

Next, show that (B4) holds with large probability. We prove the inequality involving $|c_\nu + n_\nu - c_{-\nu} - n_{-\nu}|$ portion of (B4). Proofs for the rest of the inequalities in (B4) follow analogously using the same technique below. Note that for each $k$,

$$\mathbb{E}[\mathbb{I}_\nu(k) - \mathbb{I}_{-\nu}(k)] = 0; \quad \mathbb{E}[|\mathbb{I}_\nu(k) - \mathbb{I}_{-\nu}(k)|^l] = \frac{1}{4} \text{ for any } l \geq 1.$$

It yields that

$$\rho(\mathbb{I}_\nu(k) - \mathbb{I}_{-\nu}(k))/\text{Var}(\mathbb{I}_\nu(k) - \mathbb{I}_{-\nu}(k))^{3/2} = 2.$$

Applying the Berry-Esseen theorem (Lemma A.19), we have

$$\mathbb{P}(|c_\nu + n_\nu - c_{-\nu} - n_{-\nu}| > t\sqrt{n}) = \mathbb{P}(|\sum_{k=1}^{n}(\mathbb{I}_\nu(k) - \mathbb{I}_{-\nu}(k))| > t\sqrt{n}) \geq 2\bar{\Phi}(2t) - \frac{12}{\sqrt{n}}.$$

Let $t = n^{-\varepsilon}$. By $\Phi(t) \leq 1/2 + \Phi'(0)t$, we have

$$\mathbb{P}(|\sum_{k=1}^{n}(\mathbb{I}_\nu(k) - \mathbb{I}_{-\nu}(k))| > t\sqrt{n}) \geq 1 - \frac{4}{\sqrt{2\pi}n^\varepsilon} - \frac{12}{\sqrt{n}} = 1 - O(\delta). \tag{A.11}$$

Combining (A.3), (A.9)-(A.11), we prove that conditions (B1)-(B4) hold with probability at least $1 - O(\delta)$ over the randomness of the training data. As a consequence of (B1), we have

$$p/2 \leq p + \|\mu\|^2 - 10\sqrt{p\log(n)} \leq \|x_k\|^2 \leq p + \|\mu\|^2 + 10\sqrt{p\log(n)} \leq 2p$$

by Assumption (A1) and (A2). $\qquad\square$

### A.2.2 Proof of Corollary 4.2

**Corollary 4.2** (Near-orthogonality of training data). *Suppose Assumptions (A1), (A2), and Conditions (B1), (B2) from Lemma 4.1 all hold. Then for all $1 \leq i \neq k \leq n$,*

$$|\text{cossim}(x_i, x_k)| \leq \frac{2}{Cn^2}.$$

*Proof.* By Lemma 4.1, we have that under (B1) and (B2), when $i \neq j$,

$$\frac{|\langle x_i, x_j \rangle|}{\|x_i\| \cdot \|x_j\|} \leq \frac{\|\mu\|^2 + 10\sqrt{p\log(n)}}{p + \|\mu\|^2 - 10\sqrt{p\log(n)}} \leq \frac{2\|\mu\|^2}{p} \leq \frac{2}{Cn^2},$$

for sufficiently large $p$. Here the second inequality comes from Assumption (A1); and the last inequality comes from Assumption (A2). $\qquad\square$

### A.3 Properties of the initial weights and activation patterns

We begin with additional notations that is used for the proofs of Lemmas 4.3 and 4.4. Following the notations in Xu & Gu (2023), we simplify the notation of $\mathcal{J}_{\text{Pos}}$ and $\mathcal{J}_{\text{Neg}}$ defined in Section 4 as

$$\mathcal{J}_{\text{P}} := \mathcal{J}_{\text{Pos}} = \{j \in [m] : a_j > 0\}; \quad \mathcal{J}_{\text{N}} := \mathcal{J}_{\text{Neg}} = \{j \in [m] : a_j < 0\}.$$

**Active neurons/samples.** We denote the set $\mathcal{A}^{(t)}$ of all sample-neuron pairs of indices $(i, j) \in [n] \times [m]$ such that the neuron $j$ is active with respect to the sample $x_i$ at time $t$. In other words, we define

$$\mathcal{A}^{(t)} := \{(i, j) \in [n] \times [m] : \langle w_j^{(t)}, x_i \rangle > 0\}.$$

Next, for a fixed time index $t \geq 0$, define for each sample index $i \in [n]$ the subset $\mathcal{A}^{i,(t)} \subseteq [m]$:

$$\mathcal{A}^{i,(t)} := \{j \in [m] : \langle w_j^{(t)}, x_i \rangle > 0\},$$

Likewise, for each neuron index $j \in [m]$, define the subset $\mathcal{A}_j^{(t)} \subseteq [n]$:

$$\mathcal{A}_j^{(t)} := \{i \in [n] : \langle w_j^{(t)}, x_i \rangle > 0\}.$$

Note that $\mathcal{A}^{i,(t)}$ (resp. $\mathcal{A}_j^{(t)}$) is the set of neurons (resp. samples) that is active w.r.t sample $i$ (resp. neuron $j$) at time $t$. Next, we define the set of positive (resp. negative) neurons that is active w.r.t sample $i$ at time $t$:

$$\mathcal{J}_{\mathtt{P}}^{i,(t)} := \mathcal{J}_{\mathtt{P}} \cap \mathcal{A}^{i,(t)}; \quad \mathcal{J}_{\mathtt{N}}^{i,(t)} := \mathcal{J}_{\mathtt{N}} \cap \mathcal{A}^{i,(t)}. \tag{A.12}$$

Likewise, we define the set of clean (resp. noisy) samples that is active w.r.t neuron $j$ at time $t$:

$$\mathcal{C}_{\nu,j}^{(t)} := \mathcal{C}_\nu \cap \mathcal{A}_j^{(t)}; \quad \mathcal{N}_{\nu,j}^{(t)} := \mathcal{N}_\nu \cap \mathcal{A}_j^{(t)}, \text{ for } j \in [m], \ \nu \in \text{centers}. \tag{A.13}$$

Note that the above definitions in (A.13) are equivalent to that of (4.1) from the main text. Finally, define the notational shorthand $n_{\pm\nu} := n_\nu + n_{-\nu}$.

$(\nu, \kappa)$**-aligned neurons**. For each $\nu \in$ centers and parameter $\kappa \in [0, \frac{1}{2})$, we denote the sets of neuron indices $j \in [n]$ of corresponding to $(\nu, \kappa)$-*aligned* neurons[3] by:

$$\mathcal{J}_\nu^\kappa := \{j \in [m] : D_{\nu,j}^{(0)} > n^{1/2-\kappa}, \text{ and } \max\{d_{+\nu,j}^{(0)}, d_{-\nu,j}^{(0)}\} < \min\{c_\nu, c_{-\nu}\} - 2n_{\pm\nu} - \sqrt{n}\}.$$

In other words, we have the following identity between sets of neuron indices:

$$\mathcal{J}_\nu^\kappa = \{j \in [m] : \text{neuron } j \text{ is } (\nu, \kappa)\text{-aligned}\}$$

Finally, we define the subsets of positive (resp. negative) $(\nu, \kappa)$-aligned neurons:

$$\mathcal{J}_{\nu,\mathtt{P}}^\kappa = \mathcal{J}_{\mathtt{P}} \cap \mathcal{J}_\nu^\kappa; \quad \mathcal{J}_{\nu,\mathtt{N}}^\kappa = \mathcal{J}_{\mathtt{N}} \cap \mathcal{J}_\nu^\kappa. \tag{A.14}$$

### A.3.1 PROOF OF LEMMA 4.3

**Lemma 4.3** (Properties of the random weight initialization)**.** *Suppose Assumptions (A1), (A2) and (A6) hold. All conditions below simultaneously hold with probability at least $1 - O(n^{-\varepsilon})$ over the random initialization:*

*(C1)* $\left\| W^{(0)} \right\|_F^2 \leq \frac{3}{2} \omega_{init}^2 \, mp.$

*(C2)* $|\mathcal{J}_{\mathtt{Pos}}| \geq m/3$ *and* $|\mathcal{J}_{\mathtt{Neg}}| \geq m/3.$

*The result can be stated equivalently as follows: Denote the set of $W^{(0)}$ satisfying condition (C1) by $\mathcal{G}_W$. Denote the set of $a = (a_j)_{j=1}^m$ satisfying condition (C2) by $\mathcal{G}_A$. Then $\mathbb{P}(a \in \mathcal{G}_A, W^{(0)} \in \mathcal{G}_W) \geq 1 - O(n^{-\varepsilon}).$*

*Proof.* Recall earlier for simplicity, we defined for simplicity $\mathcal{J}_{\mathtt{P}} = \mathcal{J}_{\mathtt{Pos}}$ and $\mathcal{J}_{\mathtt{N}} = \mathcal{J}_{\mathtt{Neg}}$. Let $\delta = n^{-\varepsilon}$. Then (C1) is proved to hold with probability $1 - O(\delta)$ in the Lemma 4.2 of Frei et al. (2022b). For (C2), since $|\mathcal{J}_{\mathtt{P}}|$ and $|\mathcal{J}_{\mathtt{N}}|$ both follow the binomial distribution[4] $\mathrm{B}(m, 1/2)$, it suffices to show that $\mathbb{P}(|\mathcal{J}_{\mathtt{P}}| \geq m/3) \geq 1 - \delta$. Applying Hoeffding's inequality (Lemma A.18), we have

$$\mathbb{P}(|\mathcal{J}_{\mathtt{P}}| \leq m/3) = \mathbb{P}(|\mathcal{J}_{\mathtt{P}}| - m/2 \leq -m/6) \leq \exp(-m/18) \leq \delta,$$

where the last inequality comes from Assumption (A6). $\qquad\square$

### A.3.2 PROOF OF LEMMA 4.4

**Lemma 4.4** (Properties of the interaction between training data and initial weights)**.** *Suppose Assumptions (A1)-(A3) and (A6) hold. Given $a \in \mathcal{G}_A$ (defined in Lemma 4.3) and $X \in \mathcal{G}_{data}$ (defined in Lemma 4.1), the followings hold with probability at least $1 - O(n^{-\varepsilon})$ over the random initialization $W^{(0)}$:*

---

[3]See (4.2) in the main text for the definition of $(\nu, \kappa)$-aligned neurons

[4]See Section A.1 for details on the notation

*(D1) For all $i \in [n]$, the sample $x_i$ activates a large proportion of positive and negative neurons, i.e., $|\{j \in \mathcal{J}_{\mathrm{Pos}} : \langle w_j^{(0)}, x_i \rangle > 0\}| \geq m/7$ and $|\{j \in \mathcal{J}_{\mathrm{Neg}} : \langle w_j^{(0)}, x_i \rangle > 0\}| \geq m/7$ both hold.*

*(D2) For all $\nu \in$ centers and $\kappa \in [0, \frac{1}{2})$, both $|\{j \in \mathcal{J}_{\mathrm{Pos}} : j \text{ is } (\nu, \kappa)\text{-aligned}\}| \geq mn^{-10\varepsilon}$, and $|\{j \in \mathcal{J}_{\mathrm{Neg}} : j \text{ is } (\nu, \kappa)\text{-aligned}\}| \geq mn^{-10\varepsilon}$.*

*(D3) For all $\nu \in$ centers, we have $|\{j \in \mathcal{J}_{\mathrm{Pos}} : j \text{ is } (\pm\nu, 20\varepsilon)\text{-aligned}\}| \geq (1 - 10n^{-20\varepsilon})|\mathcal{J}_{\mathrm{Pos}}|$. Moreover, the same statement holds if "$\mathcal{J}_{\mathrm{Pos}}$" is replaced with "$\mathcal{J}_{\mathrm{Neg}}$" everywhere.*

*(D4) For all $\nu \in$ centers and $\kappa \in [0, \frac{1}{2})$, let $\mathcal{J}_{\nu,\mathrm{Pos}}^{\kappa} := \{j \in \mathcal{J}_{\mathrm{Pos}} : j \text{ is } (\nu, \kappa)\text{-aligned}\}$. Then $\sum_{j \in \mathcal{J}_{\nu,\mathrm{Pos}}^{\kappa}} (c_\nu - n_\nu - d_{-\nu,j}^{(0)}) \geq \frac{n}{10}|\mathcal{J}_{\nu,\mathrm{Pos}}^{\kappa}|$. Moreover, the same statement holds if "$\mathcal{J}_{\mathrm{Pos}}$" is replaced with "$\mathcal{J}_{\mathrm{Neg}}$" everywhere.*

Before we proceed with the proof of Lemma 4.4, we consider the following restatements of (D1) through (D4):

(D'1) For each $i \in [n]$, $x_i$ activates a constant fraction of neurons initially, i.e. for each $i \in [n]$ the sets $\mathcal{J}_{\mathrm{P}}^{i,(0)}$ and $\mathcal{J}_{\mathrm{N}}^{i,(0)}$ defined at (A.12) satisfy

$$|\mathcal{J}_{\mathrm{P}}^{i,(0)}| \geq m/7 \quad \text{and} \quad |\mathcal{J}_{\mathrm{N}}^{i,(0)}| \geq m/7.$$

(D'2) For $\nu \in$ centers and $\kappa \in [0, 1/2)$, we have $\min\{|\mathcal{J}_{\nu,\mathrm{P}}^{\kappa}|, |\mathcal{J}_{\nu,\mathrm{N}}^{\kappa}|\} \geq mn^{-10\varepsilon}$.

(D'3) For $\nu \in$ centers, we have $|\mathcal{J}_{\nu,\mathrm{P}}^{20\varepsilon} \cup \mathcal{J}_{-\nu,\mathrm{P}}^{20\varepsilon}| \geq (1 - 10n^{-20\varepsilon})|\mathcal{J}_{\mathrm{P}}|$ and $|\mathcal{J}_{\nu,\mathrm{N}}^{20\varepsilon} \cup \mathcal{J}_{-\nu,\mathrm{N}}^{20\varepsilon}| \geq (1 - 10n^{-20\varepsilon})|\mathcal{J}_{\mathrm{N}}|$.

(D'4) For $\nu \in$ centers and $\kappa \in [0, \frac{1}{2})$, we have $\sum_{j \in \mathcal{J}} (c_\nu - d_{-\nu,j}^{(0)}) \geq \frac{n}{10}|\mathcal{J}|$, where $\mathcal{J} \in \{\mathcal{J}_{\nu,\mathrm{P}}^{\kappa}, \mathcal{J}_{\nu,\mathrm{N}}^{\kappa}\}$.

Unwinding the definitions, we note that the (D'1) through (D'4) are equivalent to the (D1) through (D4) of Lemma 4.4

*Proof.* Let $\delta = n^{-\varepsilon}$. Throughout this proof, we implicitly condition on the fixed $\{a_j\} \in \mathcal{G}_A$ and $\{x_i\} \in \mathcal{G}_{\mathrm{data}}$, i.e., when writing a probability and expectation we write $\mathbb{P}(\cdot)$ and $\mathbb{E}[\cdot]$ to denote $\mathbb{P}(\cdot|\{a_j\}, \{x_i\})$ and $\mathbb{E}[\cdot|\{a_j\}, \{x_i\}]$ respectively.

**Proof of condition (D1):** Define the following events for each $i \in [n]$:

$$\mathcal{P}_i := \{|\mathcal{J}_{\mathrm{P}}^{i,(0)}| \geq m/7\}; \quad \mathcal{N}_i := \{|\mathcal{J}_{\mathrm{N}}^{i,(0)}| \geq m/7\}.$$

We first show that $\cap_{i=1}^{n}(\mathcal{P}_i \cap \mathcal{N}_i)$ occurs with large probability. To this end, applying the union bound, we have

$$\mathbb{P}\left(\cap_{i=1}^{n}(\mathcal{P}_i \cap \mathcal{N}_i)\right) = 1 - \mathbb{P}\left(\cup_{i=1}^{n}(\mathcal{P}_i^c \cup \mathcal{N}_i^c)\right) \geq 1 - \sum_{i=1}^{n}\left(\mathbb{P}(\mathcal{P}_i^c) + \mathbb{P}(\mathcal{N}_i^c)\right).$$

Note that $\mathcal{P}_i$ and $\mathcal{N}_i$ are defined completely analogously corresponding to when $a_j > 0$ and $a_j < 0$, respectively. Thus, to prove (D1), it suffices to show that $\mathbb{P}(\mathcal{P}_i^c) \leq \delta/(4n)$ for each $i$, or equivalently,

$$\mathbb{P}\left(\sum_{j \in \mathcal{J}_{\mathrm{P}}} U_j \leq \frac{m}{7}\right) \leq \frac{\delta}{2n}$$

holds for each $i \in [n]$, where $U_j := \mathbb{I}(\langle w_j^{(0)}, x_i \rangle > 0)$. Note that given $x_i$ and $\mathcal{J}_{\mathrm{P}}$, $\{U_j\}_{j \in \mathcal{J}_{\mathrm{P}}}$ are i.i.d Bernoulli random variables with mean $1/2$, thus we have

$$\mathbb{P}\left(\sum_{j \in \mathcal{J}_{\mathrm{P}}} U_j \leq \frac{m}{7}\right) \leq \mathbb{P}\left(\sum_{j \in \mathcal{J}_{\mathrm{P}}} (U_j - \frac{1}{2}) \leq (\frac{1}{7} - \frac{1}{6})m\right) \leq \exp(-2m(\frac{1}{6} - \frac{1}{7})^2) \leq \frac{\delta}{2n},$$

where the first inequality uses $|\mathcal{J}_{\mathbb{P}}| \geq m/3$; the second inequality comes from Hoeffding's inequality (Lemma A.18); and the third inequality uses Assumption (A6). Now we have proved that (D1) holds with probability at least $1 - \delta$.

**Proof of condition (D2):** Without loss of generality, we only prove the results for $\mathcal{J}_{\nu,\mathbb{P}}^{\kappa}$. Note that $\mathcal{J}_{\nu,\mathbb{P}}^{\kappa_1} \subseteq \mathcal{J}_{\nu,\mathbb{P}}^{\kappa_2}$ for $\kappa_1 < \kappa_2$. Thus we only consider the case $\kappa = 0$. It suffices to show that for each $j \in [m]$,

$$\mathbb{P}(D_{\nu,j}^{(0)} > \sqrt{n}) \geq 8n^{-10\varepsilon} \quad \text{and} \quad \mathbb{P}(d_{\mu,j}^{(0)} \geq \min\{c_\nu, c_{-\nu}\} - 2n_{\pm\nu} - \sqrt{n}) \leq n^{-10\varepsilon}, \mu \in \{\pm\nu\}. \tag{A.15}$$

Suppose (A.15) holds for any $\nu \in \{\pm\mu_1, \pm\mu_2\}$. Applying the inequality $P(A \cap B) \geq 1 - P(A^c) - P(B^c)$, we have

$$\mathbb{P}(D_{\nu,j}^{(0)} > \sqrt{n}, d_{\mu,j}^{(0)} < \min\{c_\nu, c_{-\nu}\} - 2n_{\pm\nu} - \sqrt{n}, \mu \in \{\pm\nu\}) \geq 8n^{-10\varepsilon} - 2n^{-10\varepsilon} = 6n^{-10\varepsilon}.$$

Then we have

$$\mathbb{E}[|\mathcal{J}_{\nu,\mathbb{P}}|] \geq 6n^{-10\varepsilon}|\mathcal{J}_{\mathbb{P}}| \geq \frac{2m}{n^{10\varepsilon}},$$

where the last inequality uses $\min\{|\mathcal{J}_{\mathbb{P}}|, |\mathcal{J}_{\mathbb{N}}|\} \geq m/3$, which comes from the definition of $\mathcal{G}_A$. Note that given $\{a_j\}$ and $\{x_i\}$, $|\mathcal{J}_{\nu,\mathbb{P}}|$ is the summation of i.i.d Bernoulli random variables. Applying Hoeffding's inequality (Lemma A.18), we obtain

$$\mathbb{P}(|\mathcal{J}_{\nu,\mathbb{P}}| \leq \frac{m}{n^{10\varepsilon}}) \leq \mathbb{P}(|\mathcal{J}_{\nu,\mathbb{P}}| - \mathbb{E}[|\mathcal{J}_{\nu,\mathbb{P}}|] \leq -\frac{m}{n^{10\varepsilon}}) \leq \exp(-\frac{2m^2}{n^{20\varepsilon}|\mathcal{J}_{\mathbb{P}}|}) \leq n^{-\varepsilon},$$

where the last inequality uses $|\mathcal{J}_{\mathbb{P}}| = m - |\mathcal{J}_{\mathbb{N}}| \leq 2m/3$, $20\varepsilon \leq 0.01$, and Assumption (A6). Applying the union bound, we have

$$\mathbb{P}(\cap_{\nu \in \{\pm\mu_1, \pm\mu_2\}} \{|\mathcal{J}_{\nu,\mathbb{P}}| > m/n^{10\varepsilon}\}) \geq 1 - 4n^{-\varepsilon}.$$

Thus it remains to show (A.15). Without loss of generality, we will only prove (A.15) for $\nu = +\mu_1$, which can be easily extended to other $\nu$'s. Recall that $X = [x_1, \ldots, x_n]^\top$ is the given training data. Let $V = Xw_j^{(0)}$, then $V \sim N(0, XX^\top)$. Let $Z = [z_1, \cdots, z_n]^\top, z_i = v_i/\|x_i\|, i \in [n]$. Denote $\Sigma = \text{Cov}(Z)$. Then $Z \sim N(0, \Sigma)$. By Corollary 4.2, we have

$$\Sigma_{ii} = 1; \quad |\Sigma_{ij}| \leq \frac{2}{Cn^2}$$

for $1 \leq i \neq j \leq n$. Denote

$$\mathcal{A}_1 = \mathcal{C}_{+\mu_1} \cup \mathcal{N}_{-\mu_1}; \quad \mathcal{A}_2 = \mathcal{C}_{-\mu_1} \cup \mathcal{N}_{+\mu_1}.$$

By the definition of $\mathcal{G}_{\text{data}}$ and (B3) in Lemma 4.1, we have

$$||\mathcal{A}_1| - |\mathcal{A}_2|| \leq |c_{+\mu_1} - c_{-\mu_1}| + |n_{+\mu_1} - n_{-\mu_1}| \leq (1+\eta)\sqrt{n\varepsilon \log(n)}; \tag{A.16}$$

$$|\mathcal{A}_1| + |\mathcal{A}_2| = c_{+\mu_1} + n_{+\mu_1} + c_{-\mu_1} + n_{-\mu_1} \geq \frac{n}{2} - 2\sqrt{n\varepsilon \log(n)} = \frac{n}{2} - o(n) \tag{A.17}$$

for sufficiently large $n$. Note that equivalently, we can rewrite $D_{+\mu_1,j}^{(0)}$ as

$$\sum_{i \in \mathcal{A}_1} \mathbb{I}(z_i > 0) - \sum_{i \in \mathcal{A}_2} \mathbb{I}(z_i > 0). \tag{A.18}$$

Since we want to give a lower bound for $D_{+\mu_1,j}^{(0)}$, below we only consider the case when $|\mathcal{A}_1| < |\mathcal{A}_2|$. With the new expression of $D_{+\mu_1,j}^{(0)}$, we have

$$\mathbb{P}(D_{+\mu_1,j}^{(0)} > \sqrt{n}) = \sum_{k=0}^{\lfloor|\mathcal{A}_1|-\sqrt{n}\rfloor} \sum_{\substack{\mathcal{B}_2 \subseteq \mathcal{A}_2 \\ |\mathcal{B}_2|=k}} \sum_{\substack{\mathcal{B}_1 \subseteq \mathcal{A}_1 \\ |\mathcal{B}_1|>k+\sqrt{n}}} \mathbb{E}\Big[\prod_{i \in \mathcal{B}_1 \cup \mathcal{B}_2} \mathbb{I}(z_i > 0) \cdot \prod_{i \in (\mathcal{A}_1 \setminus \mathcal{B}_1) \cup (\mathcal{A}_2 \setminus \mathcal{B}_2)} \mathbb{I}(z_i \leq 0)\Big]. \tag{A.19}$$

By Lemma A.16, we have

$$\mathbb{E}\Big[\prod_{i\in\mathcal{B}_1\cup\mathcal{B}_2}\mathbb{I}(z_i>0)\cdot\prod_{i\in(\mathcal{A}_1\setminus\mathcal{B}_1)\cup(\mathcal{A}_2\setminus\mathcal{B}_2)}\mathbb{I}(z_i\le 0)\Big]\ge\gamma^{|\mathcal{A}_1|+|\mathcal{A}_2|},\qquad(\text{A}.20)$$

where $\gamma=1/2-4/(Cn)$. Let $Z'=[z_1',\cdots,z_n']^\top\sim N(0,I_n)$. Denote $\Delta_j:=\sum_{i\in\mathcal{A}_1}\mathbb{I}(z_i'>0)-\sum_{i\in\mathcal{A}_2}\mathbb{I}(z_i'>0)$, and $n_\Delta=|\mathcal{A}_1|+|\mathcal{A}_2|$. Then we have $\Delta_j\sim B(|\mathcal{A}_1|,1/2)-B(|\mathcal{A}_2|,1/2)$, $\mathbb{E}[\Delta_j]=(|\mathcal{A}_1|-|\mathcal{A}_2|)/2$, and

$$\frac{\mathbb{E}[\Delta_j]}{\sqrt{n_\Delta}}\ge\frac{-(1+\eta)\sqrt{n\varepsilon\log(n)}}{2\sqrt{n/2-o(n)}}\ge-\sqrt{n\varepsilon\log(n)}\qquad(\text{A}.21)$$

by (A.16) and (A.17). Here the last inequality comes from Assumption (A3). Combining (A.19) and (A.20), we have

$$\mathbb{P}(D_{+\mu_1,j}^{(0)}>\sqrt{n})\ge\sum_{k=0}^{\lfloor|\mathcal{A}_1|-\sqrt{n}\rfloor}\sum_{\substack{\mathcal{B}_2\subseteq\mathcal{A}_2\\|\mathcal{B}_2|=k}}\sum_{\substack{\mathcal{B}_1\subseteq\mathcal{A}_1\\|\mathcal{B}_1|>k+\sqrt{n}}}\gamma^{|\mathcal{A}_1|+|\mathcal{A}_2|}$$

$$=(2\gamma)^{|\mathcal{A}_1|+|\mathcal{A}_2|}\sum_{k=0}^{\lfloor|\mathcal{A}_1|-\sqrt{n}\rfloor}\sum_{\substack{\mathcal{B}_2\subseteq\mathcal{A}_2\\|\mathcal{B}_2|=k}}\sum_{\substack{\mathcal{B}_1\subseteq\mathcal{A}_1\\|\mathcal{B}_1|>k+\sqrt{n}}}(\frac{1}{2})^{|\mathcal{A}_1|+|\mathcal{A}_2|}\qquad(\text{A}.22)$$

$$=(2\gamma)^{|\mathcal{A}_1|+|\mathcal{A}_2|}\mathbb{P}(\Delta_j>\sqrt{n})$$

$$\ge(1-\frac{8}{Cn})^n\mathbb{P}(\Delta_j>\sqrt{n})\ge(1-\frac{8}{C})\mathbb{P}(\Delta_j>\sqrt{n}),$$

where the second equation uses the decomposition of $\mathbb{P}(\Delta_j>\sqrt{n})$; the second inequality uses $|\mathcal{A}_1|+|\mathcal{A}_2|\le n$; and the last inequality uses $f(n)=(1-8/(Cn))^n$ is a monotonically increasing function for $n\ge 1$. Note that

$$\mathbb{P}(\Delta_j>\sqrt{n})=\mathbb{P}\Big(\frac{\Delta_j-\mathbb{E}[\Delta_j]}{\sqrt{n_\Delta}/2}>\frac{\sqrt{n}-\mathbb{E}[\Delta_j]}{\sqrt{n_\Delta}/2}\Big)$$

$$\ge\bar{\Phi}\Big(\frac{\sqrt{n}-\mathbb{E}[\Delta_j]}{\sqrt{n_\Delta}/2}\Big)-O(\frac{1}{\sqrt{n}})\ge\bar{\Phi}(2(\sqrt{3}+\sqrt{\varepsilon\log(n)}))-O(\frac{1}{\sqrt{n}}),$$

where the first inequality uses Berry-Esseen theorem (Lemma A.19), and the second inequality is from (A.17) and (A.21). If $\sqrt{\varepsilon\log(n)}\le\sqrt{3}$, then $\bar{\Phi}(2(\sqrt{3}+\sqrt{\varepsilon\log(n)}))-O(1/\sqrt{n})=\Omega(1)$, which gives a constant lower bound for $\mathbb{P}(\Delta_j>\sqrt{n})$. If $\sqrt{\varepsilon\log(n)}>\sqrt{3}$, we have

$$\bar{\Phi}(2(\sqrt{3}+\sqrt{\varepsilon\log(n)}))\ge\bar{\Phi}(4\sqrt{\varepsilon\log(n)})\ge\frac{1}{8\sqrt{2\pi\varepsilon\log(n)}}\exp(-8\varepsilon\log(n))$$

$$=\frac{1}{8\sqrt{2\pi\varepsilon\log(n)}n^{8\varepsilon}}\ge\frac{17}{n^{10\varepsilon}},$$

for sufficiently large $n$. Here the second inequality uses $\bar{\Phi}(x)\ge\Phi'(x)/(2x)$ for $x\ge 1$. Combining both situations, we have

$$\mathbb{P}(\Delta_j>\sqrt{n})\ge\frac{17}{n^{10\varepsilon}}-\frac{C_{\text{BE}}}{\sqrt{n/3}}\ge\frac{16}{n^{10\varepsilon}}\qquad(\text{A}.23)$$

for sufficiently large $n$. Combining (A.22) and (A.23), we have

$$\mathbb{P}(D_{+\mu_1,j}^{(0)}>\sqrt{n})\ge(1-\frac{8}{C})\frac{16}{n^{10\varepsilon}}\ge\frac{8}{n^{10\varepsilon}}$$

for $C\ge 16$. It remains to prove

$$\mathbb{P}(d_{\mu,j}^{(0)}\ge\min\{c_{+\mu_1},c_{-\mu_1}\}-2n_{\pm\mu_1}-\sqrt{n})\le\frac{1}{n^{10\varepsilon}},\mu\in\{\pm\mu_1\}.$$

Without loss of generality, below we prove it for $\mu = +\mu_1$. According to condition (B3) in Lemma 4.1, we have

$$\min\{c_{+\mu_1}, c_{-\mu_1}\} - 2n_{\pm\mu_1} - \sqrt{n} \geq (\frac{1}{4} - 5\eta)n - 6\sqrt{n\varepsilon\log(n)} - \sqrt{n} \geq (\frac{1}{5} - \frac{5}{C})n \geq \frac{n}{6} \quad \text{(A.24)}$$

for $C \geq 150$ and sufficiently large $n$. Here the second inequality is from Assumption (A3). Thus it suffices to prove $\mathbb{P}(d_{+\mu_1,j}^{(0)} \geq n/6) \leq n^{-10\varepsilon}$. Note that

$$d_{+\mu_1,j}^{(0)} = \sum_{i \in \mathcal{C}_{+\mu_1}} \mathbb{I}(z_i > 0) - \sum_{i \in \mathcal{N}_{+\mu_1}} \mathbb{I}(z_i > 0).$$

Denote

$$\Delta_j' := \sum_{i \in \mathcal{C}_{+\mu_1}} \mathbb{I}(z_i' > 0) - \sum_{i \in \mathcal{N}_{+\mu_1}} \mathbb{I}(z_i' > 0).$$

Following the same proof procedure for the anti-concentration result of $D_{+\mu_1,j}^{(0)}$, we have

$$\mathbb{P}(d_{+\mu_1,j}^{(0)} \geq \frac{n}{6}) \leq (2\gamma_2)^{c_{+\mu_1}+n_{+\mu_1}} \mathbb{P}(\Delta_j' \geq \frac{n}{6}),$$

where $\gamma_2 = 1/2 + 4/(Cn)$. According to condition (B3) in Lemma 4.1, we have $c_{+\mu_1} - n_{+\mu_1} \leq (1/4 - 2\eta)n + 2\sqrt{n\varepsilon\log(n)}$. It yields that

$$\mathbb{E}[\Delta_j'] = \frac{c_{+\mu_1} - n_{+\mu_1}}{2} \leq (1/8 - \eta)n + \sqrt{n\varepsilon\log(n)} \leq n/7.$$

Applying Hoeffding's inequality (Lemma A.18), we have

$$\mathbb{P}(\Delta_j' \geq n/6) \leq \mathbb{P}(\Delta_j' - \mathbb{E}[\Delta_j'] \geq n/42) \leq \exp(-\Omega(n)).$$

Combining the inequalities above, we have

$$\mathbb{P}(d_{+\mu_1,j}^{(0)} \geq n/6) \leq (1 + \frac{8}{Cn})^{c_{+\mu_1}+n_{+\mu_1}} \mathbb{P}(\Delta_j' \geq n/6) = \exp(-\Omega(n)) \leq \frac{1}{n^{10\varepsilon}}, \quad \text{(A.25)}$$

where the equation uses $(1 + 8/(Cn))^{c_{+\mu_1}+n_{+\mu_1}} \leq (1 + 8/(Cn))^n \leq \exp(8/C)$. Now we have completed the proof for (D2).

**Proof of condition (D3):** Without loss of generality, we only prove the results for $\mathcal{J}_{+\mu_1,\text{P}}^{20\varepsilon} \cup \mathcal{J}_{-\mu_1,\text{P}}^{20\varepsilon}$. By Berry-Essen theorem, we have

$$\mathbb{P}(|\Delta_j| \leq n^{1/2-20\varepsilon}) = \mathbb{P}\left(\frac{\Delta_j - \mathbb{E}[\Delta_j]}{\sqrt{n_\Delta}/2} \in \left[-\frac{\mathbb{E}[\Delta_j]}{\sqrt{n_\Delta}/2} - \frac{2}{n^{20\varepsilon}}, -\frac{\mathbb{E}[\Delta_j]}{\sqrt{n_\Delta}/2} + \frac{2}{n^{20\varepsilon}}\right]\right)$$

$$\leq 2[\Phi(\frac{2}{n^{20\varepsilon}}) - \Phi(0)] + O(\frac{1}{\sqrt{n}}) \leq 4n^{-20\varepsilon},$$

where the first inequality uses $\Phi(b) - \Phi(a) \leq 2(\Phi((b-a)/2) - \Phi(0)), b \geq a$; the second inequality uses $\Phi(x) - \Phi(0) \leq \Phi'(0)x, x \geq 0$ and $20\varepsilon < 1/2$. It yields that

$$\mathbb{P}(|D_{+\mu_1,j}^{(0)}| \leq n^{1/2-20\varepsilon}) \leq 2\mathbb{P}(|\Delta_j| \leq n^{1/2-20\varepsilon}) \leq 8n^{-20\varepsilon},$$

where the first inequality is from Lemma A.15. Combined with (A.24) and (A.25), we have

$$\mathbb{P}(|D_{\nu,j}^{(0)}| > n^{1/2-20\varepsilon}, d_{\nu,j}^{(0)} < \min\{c_\nu, c_{-\nu}\} - 2n_{\pm\nu} - \sqrt{n}, \nu \in \{\pm\mu_1\})$$

$$\geq \mathbb{P}(|D_{\nu,j}^{(0)}| > n^{1/2-20\varepsilon}, d_{\nu,j}^{(0)} < n/6, \nu \in \{\pm\mu_1\})$$

$$\geq 1 - 8n^{-20\varepsilon} - 2\exp(-\Omega(n)) \geq 1 - 9n^{-20\varepsilon},$$

where the second inequality uses $D_{\nu,j}^{(0)} = -D_{-\nu,j}^{(0)}$ and $\mathbb{P}(\cap_{i=1}^n A_i) = 1 - \mathbb{P}(\cup_{i=1}^n A_i^c) \geq 1 - \sum_{i=1}^n \mathbb{P}(A_i^c)$. Note that given $\{a_j\}$ and $\{x_i\}$, $|\mathcal{J}_{\nu,\text{P}} \cup \mathcal{J}_{-\nu,\text{P}}|$ is the summation of i.i.d Bernoulli random variables with expectation larger than $1 - 9n^{-20\varepsilon}$. Applying Hoeffding's inequality (Lemma A.18),

we obtain

$$
\begin{aligned}
\mathbb{P}(|\mathcal{J}_{+\mu_1,\mathrm{P}}^{20\varepsilon} &\cup \mathcal{J}_{-\mu_1,\mathrm{P}}^{20\varepsilon}| < |\mathcal{J}_{\mathrm{P}}|(1 - 10n^{-20\varepsilon})) \\
&\leq \mathbb{P}(|\mathcal{J}_{+\mu_1,\mathrm{P}}^{20\varepsilon} \cup \mathcal{J}_{-\mu_1,\mathrm{P}}^{20\varepsilon}| - \mathbb{E}[|\mathcal{J}_{+\mu_1,\mathrm{P}}^{20\varepsilon} \cup \mathcal{J}_{-\mu_1,\mathrm{P}}^{20\varepsilon}|] < -|\mathcal{J}_{\mathrm{P}}|n^{-20\varepsilon}) \\
&\leq \exp(-2|\mathcal{J}_{\mathrm{P}}|n^{-40\varepsilon}) \leq n^{-\varepsilon},
\end{aligned}
$$

where the first inequality uses $\mathbb{E}[|\mathcal{J}_{+\mu_1,\mathrm{P}}^{20\varepsilon} \cup \mathcal{J}_{-\mu_1,\mathrm{P}}|] \geq |\mathcal{J}_{\mathrm{P}}^{20\varepsilon}|(1 - 9n^{-20\varepsilon})$ and the last inequality is from Assumption (A6) and $40\varepsilon < 0.01$.

**Proof of condition (D4):** Lastly we show that (D4) also holds with probability at least $1 - O(n^{-\varepsilon})$. Without loss of generality, we only prove it for $\mathcal{J}_{+\mu_1,\mathrm{P}}^{\kappa}$. Referring back to the definition of $\mathcal{J}_{+\mu_1,\mathrm{P}}^{\kappa}$ in equation (A.14), it is crucial to note that it solely imposes upper bounds on $d_{-\mu_1,j}^{(0)}$. Consequently, the average of $d_{-\mu_1,j}^{(0)}$ in $\mathcal{J}_{+\mu_1,\mathrm{P}}^{\kappa}$ is no more than the average of $d_{-\mu_1,j}^{(0)}$ in $\mathcal{J}_{\mathrm{P}}$, which imposes no constraints on $d_{-\mu_1,j}^{(0)}$. Armed with this understanding, when $|\mathcal{J}_{+\mu_1,\mathrm{P}}^{\kappa}| > 0$, we have that with probability 1,

$$
\frac{1}{|\mathcal{J}_{+\mu_1,\mathrm{P}}^{\kappa}|} \sum_{j \in \mathcal{J}_{+\mu_1,\mathrm{P}}^{\kappa}} (c_{+\mu_1} - n_{+\mu_1} - d_{-\mu_1,j}^{(0)}) \geq \frac{1}{|\mathcal{J}_{\mathrm{P}}|} \sum_{j \in \mathcal{J}_{\mathrm{P}}} (c_{+\mu_1} - n_{+\mu_1} - d_{-\mu_1,j}^{(0)}).
$$

Thus it suffices to show that

$$
\frac{1}{|\mathcal{J}_{\mathrm{P}}|} \sum_{j \in \mathcal{J}_{\mathrm{P}}} (c_{+\mu_1} - n_{+\mu_1} - d_{-\mu_1,j}^{(0)}) \geq \frac{n}{10} \tag{A.26}
$$

with probability at least $1 - O(\delta)$. Note that given the training data $X$, $\{d_{-\mu_1,j}^{(0)}\}_{j=1}^{m}$ are i.i.d random variables with $\mathbb{E}[d_{-\mu_1,j}^{(0)}] = (c_{-\mu_1} - n_{-\mu_1})/2$, which comes from the symmetry of the distribution of $w_j^{(0)}$. Then we have

$$
\mathbb{E}[c_{+\mu_1} - n_{+\mu_1} - d_{-\mu_1,j}^{(0)}] = c_{+\mu_1} - n_{+\mu_1}(c_{-\mu_1} - n_{-\mu_1})/2 \geq (\frac{1}{8} - 5\eta)n - 5\sqrt{n\varepsilon \log(n)} \geq \frac{n}{9}. \tag{A.27}
$$

Here the first inequality uses (B3) in Lemma 4.1 and the second inequality uses Assumption (A3). Applying Hoeffding's inequality (Lemma A.18), we obtain

$$
\begin{aligned}
\mathbb{P}\Big( &\frac{1}{|\mathcal{J}_{\mathrm{P}}|} \sum_{j \in \mathcal{J}_{\mathrm{P}}} (c_{+\mu_1} - n_{+\mu_1} - d_{-\mu_1,j}^{(0)}) < \frac{n}{10} \Big) \\
&= \mathbb{P}\Big( \sum_{j \in \mathcal{J}_{\mathrm{P}}} (d_{-\mu_1,j}^{(0)} - \mathbb{E}[d_{-\mu_1,j}^{(0)}]) > (c_{+\mu_1} - n_{+\mu_1} - \frac{n}{10} - \mathbb{E}[d_{-\mu_1,j}^{(0)}])|\mathcal{J}_{\mathrm{P}}| \Big) \\
&\leq \mathbb{P}\Big( \sum_{j \in \mathcal{J}_{\mathrm{P}}} (d_{-\mu_1,j}^{(0)} - \mathbb{E}[d_{-\mu_1,j}^{(0)}]) > \frac{n}{90}|\mathcal{J}_{\mathrm{P}}| \Big) \leq \exp\Big( -\frac{n^2|\mathcal{J}_{\mathrm{P}}|}{4050(c_{-\mu_1} + n_{-\mu_1})^2} \Big) \leq \delta,
\end{aligned}
$$

where the first inequality uses (A.27), the second inequality uses Hoeffding's inequality (Lemma A.18) and the bounds of $d_{-\mu_1,j}^{(0)}$, i.e. $-n_{-\mu_1} \leq d_{-\mu_1,j}^{(0)} \leq c_{-\mu_1}$, and the last inequality uses Assumption (A6). It proves (A.26). $\qquad\square$

**Remark A.1.** In the proof of (D2), note that when $\Sigma = I_n$, $z_i$ are independent with each other. Then (A.15) can be proved by applying Hoeffding's inequality (Lemma A.18). In our setting, $\Sigma$ is close to the identity matrix, which means that $\{z_i\}$ are weakly dependent and inspires us to prove similar results.

### A.3.3 Proof of the Probability bound of the "Good run" event

Combining the probability lower bound parts of Lemma 4.1,4.3 and 4.4, we have

$$
\begin{aligned}
&\mathbb{P}((a, W^{(0)}, X) \in \mathcal{G}_{\text{good}}) \\
&\geq \mathbb{P}(a \in \mathcal{G}_A, X \in \mathcal{G}_{\text{data}}, \text{(D1)-(D4) are satisfied}) - \mathbb{P}(W^{(0)} \notin \mathcal{G}_W) \\
&\geq \mathbb{P}(\text{(D1)-(D4) are satisfied} \,|\, a \in \mathcal{G}_A, X \in \mathcal{G}_{\text{data}}) \mathbb{P}(a \in \mathcal{G}_A, X \in \mathcal{G}_{\text{data}}) - O(n^{-\varepsilon}) \\
&\geq (1 - O(n^{-\varepsilon}))(1 - O(n^{-\varepsilon})) - O(n^{-\varepsilon}) = 1 - O(n^{-\varepsilon}),
\end{aligned}
$$

as desired.

### A.4 Trajectory Analysis of the Neurons

Let $t \geq 0$ be an arbitrary step. Denote $z_i^{(t)} := y_i f(x_i; W^{(t)})$, and $h_i^{(t)} := g_i^{(t)} - 1/2$. Then we can decompose (2.2) as

$$
w_j^{(t+1)} - w_j^{(t)} = \frac{\alpha a_j}{2n} \sum_{i=1}^n \phi'(\langle w_j^{(t)}, x_i \rangle) y_i x_i + \frac{\alpha a_j}{n} \sum_{i=1}^n h_i^{(t)} \phi'(\langle w_j^{(t)}, x_i \rangle) y_i x_i. \tag{A.28}
$$

**Remark A.2.** When $|z_i^{(t)}|$ is sufficiently small, we can use $1/2$ as an approximation for the negative derivative of the logistic loss by first-order Taylor's expansion and we will show that the training dynamics is nearly the same in the first $O(p)$ steps.

**Lemma A.3.** *Suppose that Assumptions (A1)-(A6) hold. Under a good run, for $0 \leq t \leq 1/(\sqrt{n}p\alpha) - 2$, we have $\max_{i \in [n]} |h_i^{(t)}| \leq 2/n^{3/2}$.*

**Lemma A.4.** *Suppose that Assumptions (A1)-(A6) hold. Under a good run, for $0 \leq t \leq 1/(\sqrt{n}p\alpha) - 2$, we have that for each $k \in [n]$,*

$$
\left| \langle w_j^{(t+1)} - w_j^{(t)}, x_k \rangle - \frac{\alpha a_j}{2n} \left[ y_k \phi'(\langle w_j^{(t)}, x_k \rangle) p + y_{\bar{x}_k} D_{\bar{x}_k, j}^{(t)} \|\mu\|^2 \right] \right|
$$

$$
\leq \frac{4\alpha}{n^{5/2}\sqrt{m}} \left[ \phi'(\langle w_j^{(t)}, x_k \rangle) p + \frac{C_n n^{1.99} \|\mu\|^2}{3C} \right], \text{ and} \tag{A.29}
$$

$$
\left| \langle w_j^{(t+1)} - w_j^{(t)}, \nu \rangle - \frac{\alpha a_j}{2n} y_\nu D_{\nu, j}^{(t)} \|\mu\|^2 \right| \leq \frac{5\alpha}{n^{3/2}\sqrt{m}} \|\mu\|^2. \tag{A.30}
$$

*where $C_n := 10\sqrt{\log(n)}$, $\bar{x}_k \in$ centers is defined as the cluster mean for sample $(x_k, y_k)$, and $y_\nu$ is defined as the clean label for cluster centered at $\nu$ (i.e. $y_\nu = 1$ for $\nu \in \{\pm\mu_1\}$, $y_\nu = -1$ for $\nu \in \{\pm\mu_2\}$).*

Taking a closer look at (A.29), we see that if $a_j y_k > 0$, and $x_k$ activates neuron $w_j$ at time $s$, then $x_k$ will activate neuron $w_j^{(t)}$ for any $t \in [s, 1/(\sqrt{n}p\alpha) - 2]$. Moreover, if $a_j y_k < 0$, and $x_k$ activates neuron $w_j$ at time $s$, then $x_k$ will not activate neuron $w_j$ at time $s + 1$, which implies that there is an upper bound for the inner product $\langle w_j^{(t)}, x_k \rangle$. These observations are stated as the corollary below:

**Corollary A.5.** *Suppose that Assumptions (A1)-(A6) hold. Under a good run, for any pair $(j, k) \in [m] \times [n]$, the following is true:*

(E1) *When $a_j y_k > 0$, if there exists some $0 \leq s < 1/(\sqrt{n}p\alpha) - 2$ such that $\langle w_j^{(s)}, x_k \rangle > 0$, then for any $s \leq t \leq 1/(\sqrt{n}p\alpha) - 2$, we have $\langle w_j^{(t)}, x_k \rangle > 0$.*

(E2) *When $a_j y_k < 0$, for any $0 \leq t \leq 1/(\sqrt{n}p\alpha) - 2$, we have that $\langle w_j^{(t)}, x_k \rangle \leq \frac{\alpha}{\sqrt{m}} \|\mu\|^2$.*

(E3) *When $a_j y_k < 0$, for any $0 \leq t \leq 1/(\sqrt{n}p\alpha) - 3$, we have that $\langle w_j^{(t)}, x_k \rangle > 0$ implies $\langle w_j^{(t+1)}, x_k \rangle < 0$.*

### A.4.1 PROOF OF LEMMA A.3

**Lemma A.3.** *Suppose that Assumptions (A1)-(A6) hold. Under a good run, for $0 \leq t \leq 1/(\sqrt{n}p\alpha) - 2$, we have $\max_{i \in [n]} |h_i^{(t)}| \leq 2/n^{3/2}$.*

*Proof.* It suffices to show that for $0 \leq t \leq 1/(\sqrt{n}p\alpha) - 2$,

$$\max_{i \in [n]} |h_i^{(t)}| \leq \frac{2\alpha p}{n}(t+2).$$

We prove the result by an induction on $t$. Denote

$$P(t): \quad \max_{i \in [n]} |h_i^{(\tau)}| \leq \frac{2\alpha p}{n}(t+2), \quad \forall \tau \leq t.$$

When $t = 0$, we have

$$|h_i^{(0)}| \leq \frac{p\omega_{\text{init}}\sqrt{3m}}{2} \leq \frac{\sqrt{3}\alpha\|\mu\|^2}{4nm} \leq \frac{4\alpha p}{n}$$

by Lemma A.10, Assumption (A2) and (A5). Thus $P(0)$ holds. Suppose $P(t)$ holds and $t \leq 1/(\sqrt{n}p\alpha) - 3$, then we have

$$|h_i^{(\tau)}| \leq \frac{2\alpha p}{\sqrt{n}}(\tau+2) \leq \frac{2}{\sqrt{n}}; \quad \frac{1}{2} - \frac{2}{\sqrt{n}} \leq g_i^{(\tau)} \leq \frac{1}{2} + \frac{2}{\sqrt{n}}, \quad \forall \tau \leq t,$$

which yields that $\max_{i \in [n]} g_i^{(\tau)} \leq 1$. Further we have that for each pair $(j, k) \in [m] \times [n]$,

$$|\langle w_j^{(\tau+1)} - w_j^{(\tau)}, x_k \rangle| = \left|\frac{\alpha a_j}{n} \sum_{i=1}^{n} g_i^{(\tau)} \phi'(\langle w_j^{(\tau)}, x_i \rangle) y_i \langle x_i, x_k \rangle\right|$$

$$\leq \frac{\alpha}{n\sqrt{m}} \max_{i \in [n]} g_i^{(\tau)} (2p + 2n\|\mu\|^2) \leq \frac{4\alpha p}{n\sqrt{m}},$$

where the first inequality uses $\|x_i\|^2 \leq 2p$, $|\langle x_i, x_j \rangle| \leq 2\mu^2$, which comes from Lemma 4.1, and the second inequality uses Assumption (A2). It yields that for each pair $(j, k) \in [m] \times [n]$,

$$|\langle w_j^{(t+1)}, x_k \rangle| \leq \sum_{\tau=0}^{t} |\langle w_j^{(\tau+1)} - w_j^{(\tau)}, x_k \rangle| + |\langle w_j^{(0)}, x_k \rangle| \leq \frac{4\alpha p}{n\sqrt{m}}(t+1) + \sqrt{2p}\|w_j^{(0)}\| \leq \frac{4\alpha p}{n\sqrt{m}}(t+2),$$

where the last inequality uses Lemma 4.3 and Assumption (A5). Then we have that for each $k \in [n]$,

$$|f(x_k; W^{(t+1)})| \leq \sum_{j=1}^{m} |a_j \langle w_j^{(t+1)}, x_k \rangle| \leq \sqrt{m} \max_{j \in [m]} |\langle w_j^{(t+1)}, x_k \rangle| \leq \frac{4\alpha p}{n}(t+2).$$

By $|1/(1 + \exp(z)) - 1/2| \leq |z|/2, \forall z$, we have for each $i \in [n]$,

$$|h_i^{(t+1)}| \leq \frac{1}{2}|z_i^{(t+1)}| = \frac{1}{2}|f(x_i; W^{(t+1)})| \leq \frac{2\alpha p}{n}(t+2).$$

Thus $P(t+1)$ is proved. $\square$

As a consequence of Lemma A.3, we have $g_i^{(t)} \in [1/4, 1]$ for $0 \leq t \leq 1/(\sqrt{n}p\alpha) - 2$.

### A.4.2 PROOF OF LEMMA A.4

**Lemma A.4.** *Suppose that Assumptions (A1)-(A6) hold. Under a good run, for $0 \leq t \leq 1/(\sqrt{n}p\alpha) - 2$, we have that for each $k \in [n]$,*

$$\left| \langle w_j^{(t+1)} - w_j^{(t)}, x_k \rangle - \frac{\alpha a_j}{2n} \left[ y_k \phi'(\langle w_j^{(t)}, x_k \rangle) p + y_{\bar{x}_k} D_{\bar{x}_k, j}^{(t)} \|\mu\|^2 \right] \right|$$

$$\leq \frac{4\alpha}{n^{5/2}\sqrt{m}} \left[ \phi'(\langle w_j^{(t)}, x_k \rangle) p + \frac{C_n n^{1.99} \|\mu\|^2}{3C} \right], \ and \qquad (A.29)$$

$$\left| \langle w_j^{(t+1)} - w_j^{(t)}, \nu \rangle - \frac{\alpha a_j}{2n} y_\nu D_{\nu, j}^{(t)} \|\mu\|^2 \right| \leq \frac{5\alpha}{n^{3/2}\sqrt{m}} \|\mu\|^2. \qquad (A.30)$$

*where $C_n := 10\sqrt{\log(n)}$, $\bar{x}_k \in$ centers is defined as the cluster mean for sample $(x_k, y_k)$, and $y_\nu$ is defined as the clean label for cluster centered at $\nu$ (i.e. $y_\nu = 1$ for $\nu \in \{\pm\mu_1\}$, $y_\nu = -1$ for $\nu \in \{\pm\mu_2\}$).*

*Proof.* First we have

$$\left| \frac{\alpha a_j}{n} \sum_{i=1}^n h_i^{(t)} \phi'(\langle w_j^{(t)}, x_i \rangle) y_i \langle x_i, x_k \rangle \right| \leq \frac{2\alpha}{n^{5/2}\sqrt{m}} \sum_{i=1}^n \phi'(\langle w_j^{(t)}, x_i \rangle) |\langle x_i, x_k \rangle|$$

$$\leq \frac{2\alpha}{n^{5/2}\sqrt{m}} \left[ \phi'(\langle w_j^{(t)}, x_k \rangle) \|x_k\|^2 + \sum_{i \neq k} |\langle x_i, x_k \rangle| \right] \quad (A.31)$$

$$\leq \frac{4\alpha}{n^{5/2}\sqrt{m}} \left[ \phi'(\langle w_j^{(t)}, x_k \rangle) p + n\|\mu\|^2 \right],$$

where the first inequality uses $\max_i h_i^{(t)} \leq 2n^{-3/2}$, which is from Lemma A.3; the third inequality uses $\|x_k\|^2 \leq 2p, |\langle x_i, x_k \rangle| \leq 2\|\mu\|^2$, which is induced by Lemma 4.1. Next we have the following decomposition:

$$\sum_{i=1}^n \phi'(\langle w_j^{(t)}, x_i \rangle) \langle y_i x_i, x_k \rangle$$

$$= y_k \phi'(\langle w_j^{(t)}, x_k \rangle)(\|x_k\|^2 - p - \|\mu\|^2) + \sum_{i \neq k} \phi'(\langle w_j^{(t)}, x_i \rangle) y_i (\langle x_i, x_k \rangle - \langle \bar{x}_i, \bar{x}_k \rangle)$$

$$+ y_k \phi'(\langle w_j^{(t)}, x_k \rangle)(p + \|\mu\|^2) + \sum_{i \neq k} \phi'(\langle w_j^{(t)}, x_i \rangle) y_i \langle \bar{x}_i, \bar{x}_k \rangle \qquad (A.32)$$

$$= y_k \phi'(\langle w_j^{(t)}, x_k \rangle)(\|x_k\|^2 - p - \|\mu\|^2) + \sum_{i \neq k} \phi'(\langle w_j^{(t)}, x_i \rangle) y_i (\langle x_i, x_k \rangle - \langle \bar{x}_i, \bar{x}_k \rangle)$$

$$+ y_k \phi'(\langle w_j^{(t)}, x_k \rangle) p + y_{\bar{x}_k} D_{\bar{x}_k, j}^{(t)} \|\mu\|^2 + \sum_{i: \bar{x}_i \notin \{\pm \bar{x}_k\}} \phi'(\langle w_j^{(t)}, x_i \rangle) y_i \langle \bar{x}_i, \bar{x}_k \rangle,$$

where the second equation uses the definition of $D_{\nu, j}^{(t)}$. Recall that $C_n = 10\sqrt{\log(n)}$. Combining with results in Lemma 4.1, (A.32) yields that

$$\left| \sum_{i=1}^n \phi'(\langle w_j^{(t)}, x_i \rangle) \langle y_i x_i, x_k \rangle - \left[ y_k \phi'(\langle w_j^{(t)}, x_k \rangle) p + y_{\bar{x}_k} D_{\bar{x}_k, j}^{(t)} \|\mu\|^2 \right] \right| \leq nC_n\sqrt{p} + 2n\|\mu\| \leq 2nC_n\sqrt{p},$$

$$(A.33)$$

where the first inequality uses (B1) and (B2) in Lemma 4.1 and the second inequality uses Assumption (A2). Recall the decomposition (A.28) of the gradient descent update, we have

$$\langle w_j^{(t+1)} - w_j^{(t)}, x_k \rangle = \frac{\alpha a_j}{2n} \sum_{i=1}^n \phi'(\langle w_j^{(t)}, x_i \rangle) \langle y_i x_i, x_k \rangle + \frac{\alpha a_j}{n} \sum_{i=1}^n h_i^{(t)} \phi'(\langle w_j^{(t)}, x_i \rangle) \langle y_i x_i, x_k \rangle$$

$$(A.34)$$

Then combining (A.31), (A.33), and (A.34), we have

$$\left| \langle w_j^{(t+1)} - w_j^{(t)}, x_k \rangle - \frac{\alpha a_j}{2n} \big[ y_k \phi'(\langle w_j^{(t)}, x_k \rangle) p + y_{\bar{x}_k} D_{\bar{x}_k, j}^{(t)} \|\mu\|^2 \big] \right|$$

$$\leq \frac{4\alpha}{n^{5/2}\sqrt{m}} \big[ \phi'(\langle w_j^{(t)}, x_k \rangle) p + n\|\mu\|^2 \big] + \frac{\alpha C_n \sqrt{p}}{\sqrt{m}}$$

$$\leq \frac{4\alpha}{n^{5/2}\sqrt{m}} \big[ \phi'(\langle w_j^{(t)}, x_k \rangle) p + n\|\mu\|^2 + \frac{C_n n^{2-0.01}\|\mu\|^2}{4C} \big]$$

$$\leq \frac{4\alpha}{n^{5/2}\sqrt{m}} \big[ \phi'(\langle w_j^{(t)}, x_k \rangle) p + \frac{C_n n^{2-0.01}\|\mu\|^2}{3C} \big],$$

where the second inequality uses Assumption (A1) and the last inequality holds for large enough $n$.

Now we turn to prove (A.30). Similar to (A.34), we have a decomposition for $\langle w_j^{(t+1)} - w_j^{(t)}, \nu \rangle$:

$$\langle w_j^{(t+1)} - w_j^{(t)}, \nu \rangle = \frac{\alpha a_j}{2n} \sum_{i=1}^{n} \phi'(\langle w_j^{(t)}, x_i \rangle) \langle y_i x_i, \nu \rangle + \frac{\alpha a_j}{n} \sum_{i=1}^{n} h_i^{(t)} \phi'(\langle w_j^{(t)}, x_i \rangle) \langle y_i x_i, \nu \rangle.$$

Similar to (A.31), we have

$$\left| \frac{\alpha a_j}{n} \sum_{i=1}^{n} h_i^{(t)} \phi'(\langle w_j^{(t)}, x_i \rangle) y_i \langle x_i, \nu \rangle \right| \leq \frac{4\alpha}{n^{3/2}\sqrt{m}} \|\mu\|^2$$

by Lemma A.3 and $|\langle x_i, \nu \rangle| \leq 2\|\mu\|^2$, which induced by (B1) in Lemma 4.1. Similar to (A.33), we have

$$\left| \sum_{i=1}^{n} \phi'(\langle w_j^{(t)}, x_i \rangle) \langle y_i x_i, \nu \rangle - y_\nu D_{\nu, j}^{(t)} \|\mu\|^2 \right| = \left| \sum_{i=1}^{n} \phi'(\langle w_j^{(t)}, x_i \rangle) y_i \langle x_i - \bar{x}_i, \nu \rangle \right| \leq n C_n \|\mu\|$$

(A.35)

by (B1) in Lemma 4.1. Combining the inequalities above, we have

$$\left| \langle w_j^{(t+1)} - w_j^{(t)}, \nu \rangle - \frac{\alpha a_j}{2n} y_\nu D_{\nu, j}^{(t)} \|\mu\|^2 \right| \leq \frac{4\alpha}{n^{3/2}\sqrt{m}} \|\mu\|^2 + \frac{\alpha C_n}{2\sqrt{m}} \|\mu\| \leq \frac{5\alpha}{n^{3/2}\sqrt{m}} \|\mu\|^2$$

for large enough $n$. Here the last inequality uses

$$\|\mu\|^2 \geq C n^{0.51} \sqrt{p} \geq C^{3/2} n^{1.51} \|\mu\|,$$

which comes from Assumptions (A1)-(A2). □

### A.4.3 Proof of Corollary A.5

**Corollary A.5.** *Suppose that Assumptions (A1)-(A6) hold. Under a good run, for any pair $(j, k) \in [m] \times [n]$, the following is true:*

(E1) *When $a_j y_k > 0$, if there exists some $0 \leq s < 1/(\sqrt{n} p \alpha) - 2$ such that $\langle w_j^{(s)}, x_k \rangle > 0$, then for any $s \leq t \leq 1/(\sqrt{n} p \alpha) - 2$, we have $\langle w_j^{(t)}, x_k \rangle > 0$.*

(E2) *When $a_j y_k < 0$, for any $0 \leq t \leq 1/(\sqrt{n} p \alpha) - 2$, we have that $\langle w_j^{(t)}, x_k \rangle \leq \frac{\alpha}{\sqrt{m}} \|\mu\|^2$.*

(E3) *When $a_j y_k < 0$, for any $0 \leq t \leq 1/(\sqrt{n} p \alpha) - 3$, we have that $\langle w_j^{(t)}, x_k \rangle > 0$ implies $\langle w_j^{(t+1)}, x_k \rangle < 0$.*

*Proof.* **(E1):** It suffices to show the result holds for $t = s + 1$, then by induction we can prove it for all $s \leq t \leq 1/(\sqrt{n} p \alpha) - 2$. Note that $a_j y_k = 1/\sqrt{m}$ and $\langle w_j^{(s)}, x_k \rangle > 0$, by (A.29), we have

$$\langle w_j^{(s+1)} - w_j^{(s)}, x_k \rangle \geq \frac{\alpha}{2n\sqrt{m}} (p - n\|\mu\|^2) - \frac{4\alpha}{n^{5/2}\sqrt{m}} \big[ p + \frac{C_n n^{1.99}\|\mu\|^2}{3C} \big] \geq \frac{\alpha p}{4n\sqrt{m}} > 0, \quad (A.36)$$

where the second inequality uses Assumption (A2).

**(E2):** We prove (E2) by induction. Denote

$$Q(t): \quad \langle w_j^{(t)}, x_k \rangle \le \frac{\alpha}{\sqrt{m}} \|\mu\|^2.$$

When $t = 0$, by the definition of a good run, we have

$$|\langle w_j^{(0)}, x_k \rangle| \le \|w_j^{(0)}\| \cdot \|x_k\| \le \|W^{(0)}\|_F \cdot \sqrt{2p} \le \omega_{\text{init}} p \sqrt{3m} \le \frac{\alpha}{Cn\sqrt{m}} \|\mu\|^2, \qquad (A.37)$$

where the second inequality uses Lemma 4.1; the third inequality uses Lemma 4.3; and the last inequality is from Assumption (A5). Thus $Q(0)$ holds. Suppose $Q(t)$ holds and $t \le 1/(\sqrt{n}p\alpha) - 3$. If $\langle w_j^{(t)}, x_k \rangle < 0$, we have

$$\langle w_j^{(t+1)}, x_k \rangle \le \langle w_j^{(t+1)} - w_j^{(t)}, x_k \rangle \le \frac{\alpha a_j y_{\bar{x}_k}}{2n} D_{\bar{x}_k,j}^{(t)} \|\mu\|^2 + \frac{4\alpha C_n}{3Cn^{0.51}\sqrt{m}} \|\mu\|^2 \le \frac{\alpha}{\sqrt{m}} \|\mu\|^2,$$

where the second inequality uses (A.29) and $\phi'(\langle w_j^{(t)}, x_k \rangle) = 0$; and the third inequality uses $D_{\nu,j}^{(t)} \le n$ and $n$ is large enough. If $\langle w_j^{(t)}, x_k \rangle > 0$, we have

$$\langle w_j^{(t+1)} - w_j^{(t)}, x_k \rangle \le -\frac{\alpha}{2n\sqrt{m}}(p - n\|\mu\|^2) + \frac{4\alpha}{n^{5/2}\sqrt{m}}\left[p + \frac{C_n n^{1.99}\|\mu\|^2}{3C}\right]$$

$$\le -\frac{\alpha}{2n\sqrt{m}}(p - n\|\mu\|^2) + \frac{8\alpha p}{n^{5/2}\sqrt{m}},$$

where the first inequality uses (A.29) and $\phi'(\langle w_j^{(t)}, x_k \rangle) = 1$; and the second inequality uses Assumption (A2). Combined with the inductive hypothesis, we have

$$\langle w_j^{(t+1)}, x_k \rangle = \langle w_j^{(t)}, x_k \rangle + \langle w_j^{(t+1)} - w_j^{(t)}, x_k \rangle \le \frac{\alpha}{\sqrt{m}}\|\mu\|^2 - \frac{\alpha}{2n\sqrt{m}}(p - n\|\mu\|^2) + \frac{8\alpha p}{n^{5/2}\sqrt{m}} < 0$$

by Assumption (A2). Thus $Q(t+1)$ holds. And (E3) is also proved by the last inequality. □

### A.4.4 PROOF OF LEMMA A.6

Since the analysis on one cluster can be similarly replicated on other clusters, below we will focus on analyzing the cluster centered at $+\mu_1$. Given the training set, $D_{+\mu_1,j}^{(0)}$ is a function of the random initialization $w_j^{(0)}$. $D_{+\mu_1,j}^{(0)}$ plays an important role in determining the direction that $w_j^{(t)}, t \ge 1$ aligns with and the sign of the inner product $\langle w_j^{(t)}, x_k \rangle$. For $\bar{x}_k \in \{\pm\mu_1\}$, $y_{\bar{x}_k} = 1$. Then for each $t \le 1/(\sqrt{n}p\alpha) - 2$, (A.29) is simplified to

$$\left|\langle w_j^{(t+1)} - w_j^{(t)}, x_k \rangle - \frac{\alpha a_j y_k p}{2n}\right| \le \frac{4\alpha p}{n^{5/2}\sqrt{m}} + \frac{\alpha}{2\sqrt{m}}\|\mu\|^2, \quad \text{when } \langle w_j^{(t)}, x_k \rangle > 0; \quad (A.38)$$

$$\left|\langle w_j^{(t+1)} - w_j^{(t)}, x_k \rangle - \frac{\alpha a_j}{2n} D_{\bar{x}_k,j}^{(t)} \|\mu\|^2\right| \le \frac{4\alpha C_n}{3Cn^{0.01}\sqrt{mn}}\|\mu\|^2, \quad \text{when } \langle w_j^{(t)}, x_k \rangle \le 0. \quad (A.39)$$

Here $C_n = 10\sqrt{\log(n)}$ is defined in Lemma A.4. We will elaborate on the outcomes for neurons with $a_j > 0$ and $a_j < 0$ separately in the following lemmas.

**Lemma A.6.** *Suppose that Assumptions (A1)-(A6) hold. Under a good run, we have that for any $j \in \mathcal{J}_{+\mu_1,\text{P}}^{20\varepsilon}$ (or equivalently, for any neuron $j \in \mathcal{J}_{\text{Pos}}$ that is $(\mu_1, 20\varepsilon)$-aligned) ), the followings hold for $1 \le t \le 1/(\sqrt{n}p\alpha) - 2$:*

*(F1)*

$$\mathcal{C}_{+\mu_1,j}^{(t)} = \mathcal{C}_{+\mu_1}; \quad \mathcal{C}_{-\mu_1,j}^{(t)} = \mathcal{C}_{-\mu_1,j}^{(0)}; \quad \mathcal{N}_{-\mu_1,j}^{(t)} = \varnothing; \quad D_{+\mu_1,j}^{(t)} > c_{+\mu_1} - n_{+\mu_1} - d_{-\mu_1,j}^{(0)}.$$

*(F2)*
$$\langle w_j^{(t)} - w_j^{(t-1)}, \mu_1 \rangle \geq \frac{\alpha}{4n\sqrt{m}} D_{+\mu_1,j}^{(t-1)} \|\mu\|^2.$$

*Proof.* Given $j \in \mathcal{J}_{+\mu_1,\mathrm{P}}^{20\varepsilon}$, when $t = 0$, for $x_k \in \mathcal{C}_{+\mu_1,j}^{(0)}$, we have $a_j y_k > 0$. Thus by Corollary A.5, we have

$$x_k \in \mathcal{C}_{+\mu_1,j}^{(t)}, \quad 0 \leq t \leq 1/(\sqrt{n}p\alpha) - 2. \tag{A.40}$$

Similarly we have that for $x_k \in \mathcal{C}_{-\mu_1,j}^{(0)}$,

$$x_k \in \mathcal{C}_{-\mu_1,j}^{(t)}, \quad 0 \leq t \leq 1/(\sqrt{n}p\alpha) - 2; \tag{A.41}$$

and for $x_k \in \mathcal{N}_{-\mu_1,j}^{(0)}$, $x_k \notin \mathcal{N}_{-\mu_1,j}^{(1)}$ since $a_j y_k < 0$.

Next for $x_k \in \mathcal{C}_{+\mu_1} \backslash \mathcal{C}_{+\mu_1,j}^{(0)}$, we have

$$
\begin{aligned}
\langle w_j^{(1)} - w_j^{(0)}, x_k \rangle &\geq \frac{\alpha a_j}{2n} D_{+\mu_1,j}^{(0)} \|\mu\|^2 - \frac{4\alpha C_n}{3Cn^{0.01}\sqrt{mn}} \|\mu\|^2 \\
&\geq \frac{\alpha}{2n^{20\varepsilon}\sqrt{mn}} \|\mu\|^2 - \frac{4\alpha C_n}{3Cn^{0.01}\sqrt{mn}} \|\mu\|^2 \geq \frac{\alpha}{4n^{20\varepsilon}\sqrt{mn}} \|\mu\|^2,
\end{aligned}
\tag{A.42}
$$

where the first inequality is from (A.39); the second inequality uses $D_{+\mu_1,j}^{(0)} > n^{1/2-20\varepsilon}$, which is from $j \in \mathcal{J}_{+\mu_1,\mathrm{P}}^{20\varepsilon}$; and the last inequality uses $40\varepsilon < 0.01$. It yields that

$$\langle w_j^{(1)}, x_k \rangle \geq \langle w_j^{(1)} - w_j^{(0)}, x_k \rangle - \|w_j^{(0)}\| \cdot \|x_k\| \geq \frac{\alpha}{4n^{20\varepsilon}\sqrt{mn}} \|\mu\|^2 - \frac{\alpha}{Cn\sqrt{m}} \|\mu\|^2 > 0, \tag{A.43}$$

where the second inequality uses (A.37). Thus we have

$$\mathcal{C}_{+\mu_1} \backslash \mathcal{C}_{+\mu_1,j}^{(0)} \subseteq \mathcal{C}_{+\mu_1,j}^{(1)}.$$

Combined with (A.40), we obtain $\mathcal{C}_{+\mu_1,j}^{(1)} = \mathcal{C}_{+\mu_1}$. Then by Corollary A.5, we have

$$\mathcal{C}_{+\mu_1,j}^{(t)} = \mathcal{C}_{+\mu_1}, \quad 0 \leq t \leq 1/(\sqrt{n}p\alpha) - 2.$$

For $x_k \in \left(\mathcal{C}_{-\mu_1} \backslash \mathcal{C}_{-\mu_1,j}^{(0)}\right) \cup \left(\mathcal{N}_{-\mu_1} \backslash \mathcal{N}_{-\mu_1,j}^{(0)}\right)$, Following similar analysis of (A.43), we have

$$\langle w_j^{(1)}, x_k \rangle \leq \langle w_j^{(1)} - w_j^{(0)}, x_k \rangle + \|w_j^{(0)}\| \cdot \|x_k\| \leq -\left(\frac{\alpha}{4n^{20\varepsilon}\sqrt{mn}} \|\mu\|^2 - \frac{\alpha}{Cn\sqrt{m}} \|\mu\|^2\right) < 0. \tag{A.44}$$

Thus we have $\mathcal{C}_{-\mu_1} \backslash \mathcal{C}_{-\mu_1,j}^{(0)} \notin \mathcal{C}_{-\mu_1,j}^{(1)}$, and $\mathcal{N}_{-\mu_1} \backslash \mathcal{N}_{-\mu_1,j}^{(0)} \notin \mathcal{N}_{-\mu_1,j}^{(1)}$. Combined with (A.41) and $\mathcal{N}_{-\mu_1,j}^{(0)} \notin \mathcal{N}_{-\mu_1,j}^{(1)}$, we obtain

$$\mathcal{C}_{-\mu_1,j}^{(1)} = \mathcal{C}_{-\mu_1,j}^{(0)}; \quad \mathcal{N}_{-\mu_1,j}^{(1)} = \varnothing.$$

It yields that

$$D_{+\mu_1,j}^{(1)} = c_{+\mu_1} - |\mathcal{N}_{+\mu_1,j}^{(1)}| - |\mathcal{C}_{-\mu_1,j}^{(0)}| > c_{+\mu_1} - n_{+\mu_1} - d_{-\mu_1,j}^{(0)} > \sqrt{n},$$

where the last inequality uses $d_{+\mu_1,j}^{(0)} < \min\{c_{+\mu_1}, c_{-\mu_1}\} - 2n_{\pm\mu_1} - \sqrt{n}$ and

$$c_{+\mu_1} - n_{+\mu_1} - d_{-\mu_1,j}^{(0)} > \sqrt{n} + d_{+\mu_1,j}^{(0)} - d_{-\mu_1,j}^{(0)} > \sqrt{n}.$$

Thus (F1) holds for $t = 1$. Then (F1) is proved by replicating the same analysis and employing induction.

For the inner product with the cluster mean $+\mu_1$, by (A.30) we have

$$\langle w_j^{(t+1)} - w_j^{(t)}, \mu_1 \rangle \geq \frac{\alpha}{2n\sqrt{m}} D_{+\mu_1,j}^{(t)} \|\mu\|^2 - \frac{5C_n\alpha}{n^{3/2}\sqrt{m}} \|\mu\|^2 \geq \frac{\alpha}{4n\sqrt{m}} D_{+\mu_1,j}^{(t)} \|\mu\|^2,$$

where the last inequality uses $D_{+\mu_1,j}^{(t)} > 0$. □

### A.4.5 PROOF OF LEMMA A.7

**Lemma A.7.** *Suppose that Assumptions (A1)-(A6) hold. Under a good run, for any $j \in \mathcal{J}_{+\mu_1,\mathbb{N}}^{20\varepsilon} \cup \mathcal{J}_{-\mu_1,\mathbb{N}}^{20\varepsilon}$ (or equivalently, for any neuron $j \in \mathcal{J}_{\mathtt{Neg}}$ that is $(\pm\mu_1, 20\varepsilon)$-aligned), the followings hold for $2 \le t \le 1/(\sqrt{n}p\alpha) - 2$.*

$$\mathcal{N}_{+\mu_1,j}^{(t)} = \mathcal{N}_{+\mu_1}, \mathcal{N}_{-\mu_1,j}^{(t)} = \mathcal{N}_{-\mu_1}; \tag{A.45}$$

$$-n - \Delta_{\mu_1}(t-2) \le \sum_{s=0}^{t} D_{\nu,j}^{(s)} \le n + \Delta_{\mu_1}(t-2), \quad \nu \in \{\pm\mu_1\}, \tag{A.46}$$

*where $\Delta_{\mu_1} := |n_{+\mu_1} - n_{-\mu_1}| + \sqrt{n}$.*

*Proof.* For a given $\nu \in \{\pm\mu_1\}$, suppose $j \in \mathcal{J}_{\nu,\mathbb{N}}^{20\varepsilon}$. Then we have

$$a_j < 0; \quad D_{\nu,j}^{(0)} > n^{1/2-20\varepsilon}; \quad d_{\nu,j}^{(0)} \le \min\{c_\nu, c_{-\nu} - 2n_{\pm\nu} - \sqrt{n}\} \tag{A.47}$$

according to the definition (A.14). Note that we study the same data as in Lemma A.6 and only $\mathrm{sgn}(a_j)$ is flipped in the trajectory analysis compared to the setting in Lemma A.6, our analysis in the first two iterations follows similar procedures in Lemma A.6. For $x_k \in \mathcal{C}_{\nu,j}^{(0)} \cup \mathcal{C}_{-\nu,j}^{(0)}$, $a_j y_k < 0$, by Corollary A.5, we have

$$\langle w_j^{(1)}, x_k \rangle < 0. \tag{A.48}$$

For $x_k \in \mathcal{N}_{\nu,j}^{(0)} \cup \mathcal{N}_{-\nu,j}^{(0)}$, $a_j y_k > 0$, by Corollary A.5, we have

$$\langle w_j^{(t)}, x_k \rangle > 0 \tag{A.49}$$

for any $t \le 1/(\sqrt{n}p\alpha) - 2$. For $x_k \in (\mathcal{C}_\nu \backslash \mathcal{C}_{\nu,j}^{(0)}) \cup (\mathcal{N}_\nu \backslash \mathcal{N}_{\nu,j}^{(0)})$, similar to (A.42), we have

$$\langle w_j^{(1)} - w_j^{(0)}, x_k \rangle \le -\left(\frac{\alpha a_j}{2n} D_{+\mu_1,j}^{(0)} \|\mu\|^2 - \frac{4\alpha C_n}{3Cn^{0.01}\sqrt{mn}} \|\mu\|^2\right) \le -\frac{\alpha}{4n^{20\varepsilon}\sqrt{mn}} \|\mu\|^2 < 0,$$

then similar to (A.43), we have

$$\langle w_j^{(1)}, x_k \rangle \le -\langle w_j^{(1)} - w_j^{(0)}, x_k \rangle + \|w_j^{(0)}\| \cdot \|x_k\| \le -\frac{\alpha}{4n^{20\varepsilon}\sqrt{mn}} \|\mu\|^2 + \frac{\alpha}{Cn\sqrt{m}} \|\mu\|^2 < 0. \tag{A.50}$$

For $x_k \in (\mathcal{C}_{-\nu} \backslash \mathcal{C}_{-\nu,j}^{(0)}) \cup (\mathcal{N}_{-\nu} \backslash \mathcal{N}_{-\nu,j}^{(0)})$, similar to (A.44), we have

$$\langle w_j^{(1)}, x_k \rangle \ge \langle w_j^{(1)} - w_j^{(0)}, x_k \rangle - \|w_j^{(0)}\| \cdot \|x_k\| \ge \frac{\alpha}{4n^{20\varepsilon}\sqrt{mn}} \|\mu\|^2 - \frac{\alpha}{Cn\sqrt{m}} \|\mu\|^2 > 0. \tag{A.51}$$

Combining (A.48)-(A.51), we have

$$\mathcal{C}_{\nu,j}^{(1)} = \varnothing; \quad \mathcal{C}_{-\nu,j}^{(1)} = \mathcal{C}_{-\nu} \backslash \mathcal{C}_{-\nu,j}^{(0)}; \quad \mathcal{N}_{\nu,j}^{(1)} = \mathcal{N}_{\nu,j}^{(0)}; \quad \mathcal{N}_{-\nu,j}^{(1)} = \mathcal{N}_{-\nu}. \tag{A.52}$$

Thus by the definition of $D_{\nu,j}^{(1)}$, we have

$$D_{\nu,j}^{(1)} = -|\mathcal{N}_{\nu,j}^{(0)}| - c_{-\nu} + |\mathcal{C}_{-\nu,j}^{(0)}| + n_{-\nu} \le -|\mathcal{N}_{\nu,j}^{(0)}| - c_{-\nu} + d_{-\nu,j}^{(0)} + 2n_{-\nu}. \tag{A.53}$$

It further yields that

$$D_{\nu,j}^{(1)} + D_{\nu,j}^{(0)} \le -|\mathcal{N}_{\nu,j}^{(0)}| - c_{-\nu} + 2n_{-\nu} + d_{\nu,j}^{(0)} \le -c_{-\nu} + 2n_{-\nu} + d_{\nu,j}^{(0)} < -\sqrt{n},$$

where the first inequality uses (A.53) and the definition of $D_{\nu,j}^{(0)}$, and the third inequality uses (A.47).

After the second iteration, for $x_k \in \mathcal{N}_\nu \backslash \mathcal{N}_{\nu,j}^{(1)}$, $\langle w_j^{(0)}, x_k \rangle < 0$, $\langle w_j^{(1)}, x_k \rangle < 0$. Then we have

$$\langle w_j^{(2)} - w_j^{(0)}, x_k \rangle \geq -\frac{\alpha}{2n\sqrt{m}}(D_{\nu,j}^{(0)} + D_{\nu,j}^{(1)})\|\mu\|^2 - \frac{4\alpha C_n}{3Cn^{0.01}\sqrt{mn}}\|\mu\|^2$$
$$> \frac{\alpha}{2\sqrt{mn}}\|\mu\|^2 - \frac{4\alpha C_n}{3Cn^{0.01}\sqrt{mn}}\|\mu\|^2,$$

where the first inequality uses (A.39), and the second inequality uses $D_{\nu,j}^{(1)} + D_{\nu,j}^{(0)} < -\sqrt{n}$. It further yields that

$$\langle w_j^{(2)}, x_k \rangle \geq \langle w_j^{(2)} - w_j^{(0)}, x_k \rangle - \|w_j^{(0)}\| \cdot \|x_k\| \geq \frac{\alpha}{2\sqrt{mn}}\|\mu\|^2 - \frac{4\alpha C_n}{3Cn^{0.01}\sqrt{mn}}\|\mu\|^2 - \frac{\alpha}{Cn\sqrt{m}}\|\mu\|^2 > 0.$$
(A.54)

For $x_k \in \mathcal{N}_{\nu,j}^{(1)} \cup \mathcal{N}_{-\nu}$, note that $a_j y_k > 0$. Then by Corollary A.5, we have $\langle w_j^{(2)}, x_k \rangle > 0$. Combined with (A.54), we obtain $\mathcal{N}_{\nu,j}^{(2)} = \mathcal{N}_\nu$, $\mathcal{N}_{-\nu,j}^{(2)} = \mathcal{N}_{-\nu}$. Again by Corollary A.5, we have that for $2 \leq t \leq 1/(\sqrt{n}p\alpha) - 2$,

$$\mathcal{N}_{\nu,j}^{(t)} = \mathcal{N}_\nu, \quad \mathcal{N}_{-\nu,j}^{(t)} = \mathcal{N}_{-\nu}, \tag{A.55}$$

i.e. for $t \geq 2$, neurons with $j \in \mathcal{J}_{\nu,\mathbf{N}}^{20\varepsilon} \cup \mathcal{J}_{-\nu,\mathbf{N}}^{20\varepsilon}$ are active for all noisy points in $\mathcal{N}_{\pm\mu_1}$, which proves (A.45).

For $x_k \in \mathcal{C}_{-\nu,j}^{(1)}$, note that $a_j y_k < 0$ and $\langle w_j^{(1)}, x_k \rangle > 0$. Then by Corollary A.5, we have $\langle w_j^{(2)}, x_k \rangle < 0$. For $x_k \in \mathcal{C}_{-\nu} \backslash \mathcal{C}_{-\nu,j}^{(1)}$, by (A.52) we have $\langle w_j^{(0)}, x_k \rangle > 0$, $\langle w_j^{(1)}, x_k \rangle < 0$. It yields that

$$\langle w_j^{(2)} - w_j^{(0)}, x_k \rangle \leq -\frac{\alpha}{2n\sqrt{m}}(p + D_{\nu,j}^{(1)}\|\mu\|^2) + \frac{4\alpha p}{n^{5/2}\sqrt{m}} + \frac{\alpha}{2\sqrt{m}}\|\mu\|^2 + \frac{4\alpha C_n}{3Cn^{0.01}\sqrt{mn}}\|\mu\|^2 \leq -\frac{\alpha p}{4n\sqrt{m}},$$

where the first inequality uses (A.38) and (A.39), and the second inequality uses Assumption (A2). It further yields that

$$\langle w_j^{(2)}, x_k \rangle < \langle w_j^{(2)} - w_j^{(0)}, x_k \rangle + \|w_j^{(0)}\| \cdot \|x_k\| \leq -\frac{\alpha p}{4n\sqrt{m}} + \frac{\alpha}{Cn\sqrt{m}}\|\mu\|^2 < 0 \tag{A.56}$$

by Assumption (A2). Thus we have $\mathcal{C}_{-\nu,j}^{(2)} = \varnothing$.

For $x_k \in \mathcal{C}_{\nu,j}^{(0)}$, $\langle w_j^{(0)}, x_k \rangle > 0$, $\langle w_j^{(1)}, x_k \rangle < 0$, which is similar to the setting of $\mathcal{C}_{-\nu} \backslash \mathcal{C}_{-\nu,j}^{(1)}$. Repeating the analysis above, we have

$$\langle w_j^{(2)}, x_k \rangle < 0.$$

For $x_k \in \mathcal{C}_\nu \backslash \mathcal{C}_{\nu,j}^{(0)}$, note that $\langle w_j^{(0)}, x_k \rangle < 0$, $\langle w_j^{(1)}, x_k \rangle < 0$, then we have

$$\langle w_j^{(2)} - w_j^{(0)}, x_k \rangle \geq -\frac{\alpha}{2n\sqrt{m}}(D_{\nu,j}^{(0)} + D_{\nu,j}^{(1)})\|\mu\|^2 - \frac{4\alpha C_n}{3Cn^{0.01}\sqrt{mn}}\|\mu\|^2$$
$$> \frac{\alpha}{2\sqrt{mn}}\|\mu\|^2 - \frac{4\alpha C_n}{3Cn^{0.01}\sqrt{mn}}\|\mu\|^2 > 0,$$

where the first inequality uses (A.39) and the second inequality uses (A.53). Combining the inequalities above, we obtain

$$\mathcal{C}_{\nu,j}^{(2)} = \mathcal{C}_\nu \backslash \mathcal{C}_{\nu,j}^{(0)}; \quad \mathcal{C}_{-\nu,j}^{(2)} = \varnothing; \quad \mathcal{N}_{\nu,j}^{(2)} = \mathcal{N}_\nu; \quad \mathcal{N}_{-\nu,j}^{(2)} = \mathcal{N}_{-\nu}. \tag{A.57}$$

Combining (A.52) and (A.57), we have

$$\sum_{s=0}^{2} D_{\nu,j}^{(s)} = c_\nu - c_{-\nu} - n_\nu + 3n_{-\nu} - 2|\mathcal{N}_\nu^{(0)}|,$$

and it yields that

$$c_\nu - c_{-\nu} - 3n_\nu + 3n_{-\nu} \le \sum_{s=0}^{2} D_{\nu,j}^{(s)} \le c_\nu - c_{-\nu} + 3n_{-\nu} - n_\nu.$$

It remains to prove (A.46). It suffices to prove

$$c_\nu - 2c_{-\nu} - 4n_\nu + 3n_{-\nu} - \Delta_{\mu_1}(t-2) \le \sum_{s=0}^{t} D_{\nu,j}^{(s)} \le (2c_\nu - c_{-\nu} + 4n_{-\nu} - n_\nu) + \Delta_{\mu_1}(t-2), \nu \in \{\pm\mu_1\},$$

since $2c_\nu - c_{-\nu} + 4n_{-\nu} - n_\nu \le n$ and $c_\nu - 2c_{-\nu} - 4n_\nu + 3n_{-\nu} \ge -n$ by Lemma 4.1. Without loss of generality, below we only show the proof of the right-hand side. Denote $\mathcal{T} = \{t \in [T], t \ge 3, D_{\nu,j}^{(t)} > \Delta_{\mu_1}\} = \{t_i\}_{i=1}^{K}, t_1 < t_2 < \cdots < t_K$. To prove the right-hand side of (A.46), it suffices to show that the followings hold

$$\sum_{t=t_i}^{s} D_{\nu,j}^{(t)} \le c_\nu + n_{-\nu} + \Delta_{\mu_1}(s - t_i); \tag{A.58}$$

$$\sum_{t=t_i}^{t_{i+1}-1} D_{\nu,j}^{(t)} \le \Delta_{\mu_1}(t_{i+1} - t_i) \tag{A.59}$$

for any $i \in [K]$ and all $s \in [t_i, t_{i+1} - 2]$. (A.58) directly follows from the definition of the set $\mathcal{T}$ and the fact that $D_{\nu,j}^{(t)} \le c_\nu + n_{-\nu}$ for any $j, t$. For a given $t_i, t_i \in \mathcal{T}$, we have $D_{\nu,j}^{(t_i)} > \Delta_{\mu_1} \ge \sqrt{n}$. By (A.39), we have that for any $x_k \in \mathcal{C}_\nu \backslash \mathcal{C}_\nu^{(t_i)}(j)$,

$$\langle w_j^{(t_i+1)}, x_k \rangle \le \langle w_j^{(t_i+1)} - w_j^{(t_i)}, x_k \rangle \le -\frac{\alpha}{2n\sqrt{m}} D_{\nu,j}^{(t_i)} \|\mu\|^2 + \frac{4\alpha C_n}{3Cn^{0.01}\sqrt{mn}} \|\mu\|^2$$

$$\le -\frac{\alpha}{4n\sqrt{m}} D_{\nu,j}^{(t_i)} \|\mu\|^2 < 0, \tag{A.60}$$

which implies that $w_j^{(t_i+1)}$ is still inactive for those $x_k$ that didn't activate $w_j^{(t_i)}$. For any $x_k \in \mathcal{C}_{\nu,j}^{(t_i)}$, since $a_j y_k < 0$, by Corollary A.5, we have

$$\langle w_j^{(t_i)}, x_k \rangle \le \frac{\alpha \|\mu\|^2}{\sqrt{m}}.$$

Combined with (A.38), we have

$$\langle w_j^{(t_i+1)}, x_k \rangle = \langle w_j^{(t_i+1)} - w_j^{(t_i)}, x_k \rangle + \langle w_j^{(t_i)}, x_k \rangle$$

$$\le -\frac{\alpha p}{2n\sqrt{m}} + \frac{4\alpha p}{n^{5/2}\sqrt{m}} + \frac{3\alpha}{2\sqrt{m}} \|\mu\|^2 \le -\frac{\alpha p}{4n\sqrt{m}} < 0 \tag{A.61}$$

where the second inequality uses Assumption (A2). Combining (A.60) and (A.61), we have $\mathcal{C}_{\nu,j}^{(t_i+1)} = \varnothing$, and

$$\langle w_j^{(t_i+1)}, x_k \rangle \le -\frac{\alpha}{2n\sqrt{m}} D_{\nu,j}^{(t_i)} \|\mu\|^2 + \frac{4\alpha C_n}{3Cn^{0.01}\sqrt{mn}} \|\mu\|^2 \tag{A.62}$$

for all $x_k \in \mathcal{C}_\nu$. It yields that

$$D_{\nu,j}^{(t_i+1)} = |\mathcal{C}_{\nu,j}^{(t_i+1)}| - |\mathcal{C}_{-\nu,j}^{(t_i+1)}| + n_{-\nu} - n_\nu = -|\mathcal{C}_{-\nu,j}^{(t_i+1)}| + n_{-\nu} - n_\nu \le |n_{+\mu_1} - n_{-\mu_1}|,$$

where the first equation uses (A.45). It implies that $t_{i+1} - t_i > 1$. Let $t_i^\star = \min\{t \in \mathbb{N} : t_i + 1 < t \le t_{i+1}, \mathcal{C}_\nu^{(t)}(j) \ne \varnothing\}$. We claim that $t_i^\star$ is well-defined for each $i$, because $\mathcal{C}_\nu^{(t_{i+1})}(j) \ne \varnothing$. Otherwise we have $D_{\nu,j}^{(t_{i+1})} \le |n_{+\mu_1} - n_{-\mu_1}| < \Delta_{\mu_1}$, which contradicts to the definition of the set $\mathcal{T}$. Thus $t_i^\star$ always exists. Choose one point from the set $\mathcal{C}_{\nu,j}^{(t_i^\star)}$ and denote it as $x_k^\star$. Note that for any

$t \in [t_i + 1, t_i^\star - 1]$, we have $\mathcal{C}_\nu^{(t)}(j) = \varnothing$, $D_{\nu,j}^{(t)} \leq |n_{+\mu_1} - n_{-\mu_1}|$, and by (A.39),

$$\langle w_j^{(t+1)} - w_j^{(t)}, x_k^\star \rangle \leq -\frac{\alpha}{2n\sqrt{m}} D_{\nu,j}^{(t)} \|\mu\|^2 + \frac{4\alpha C_n}{3Cn^{0.01}\sqrt{mn}} \|\mu\|^2.$$

Combined with (A.62), it yields that

$$0 \leq \langle w_j^{(t_i^\star)}, x_k^\star \rangle = \sum_{t=t_i+1}^{t_i^\star-1} \langle w_j^{(t+1)} - w_j^{(t)}, x_k^\star \rangle + \langle w_j^{(t_i+1)}, x_k^\star \rangle$$

$$\leq -\frac{\alpha \|\mu\|^2}{2n\sqrt{m}} \Big( D_{\nu,j}^{(t_i)} + \sum_{t=t_i+1}^{t_i^\star-1} D_{\nu,j}^{(t)} - \frac{4\sqrt{n}C_n}{3Cn^{0.01}}(t_i^\star - t_i) \Big).$$

It further yields that

$$\sum_{t=t_i}^{t_i^\star-1} D_{\nu,j}^{(t)} \leq \frac{4\sqrt{n}C_n}{3Cn^{0.01}}(t_i^\star - t_i) \leq \sqrt{n}(t_i^\star - t_i).$$

If $t_i^\star = t_{i+1}$, then we've proved (A.59). If $t_i^\star < t_{i+1}$, then we have

$$\sum_{t=t_i}^{t_{i+1}-1} D_{\nu,j}^{(t)} = \sum_{t=t_i}^{t_i^\star-1} D_{\nu,j}^{(t)} + \sum_{t=t^\star}^{t_{i+1}-1} D_{\nu,j}^{(t)} \leq \sqrt{n}(t^\star - t_i) + \Delta_{\mu_1}(t_{i+1} - t^\star) \leq \Delta_{\mu_1}(t_{i+1} - t_i),$$

which proves the right side. For the left side, similarly we denote $\mathcal{T}_- = \{t \in [T], t \geq 3, D_{\nu,j}^{(t)} < -\Delta_{\mu_1}\} = \{t_i\}_{i=1}^K, t_1 < t_2 < \cdots < t_K$. Following the same analysis, we can prove that the followings hold

$$\sum_{t=t_i}^{s} D_{\nu,j}^{(t)} \geq -c_{-\nu} - n_\nu - \Delta_{\mu_1}(s - t_i); \quad \sum_{t=t_i}^{t_{i+1}-1} D_{\nu,j}^{(t)} \geq -\Delta_{\mu_1}(t_{i+1} - t_i)$$

for any $i \in [K]$ and all $s \in [t_i, t_{i+1} - 2]$. It proves the left-hand side of (A.46). $\qquad \square$

## A.5 Proof of the Main Theorem

We rigorously prove Theorem 3.1 in this section. The upper bound of $t$ in the theorems below is $1/(\sqrt{n}p\alpha) - 2$, which by Assumption (A4), is larger than $\sqrt{n}$, the upper bound of $t$ in Theorem 3.1.

### A.5.1 Proof of Theorem A.8: 1-step Overfitting

**Theorem A.8.** *Suppose that Assumptions (A1)-(A6) hold. Under a good run, the classifier* $\mathrm{sgn}(f(x, W^{(t)}))$ *can correctly classify all training datapoints for* $1 \leq t \leq 1/(\sqrt{n}p\alpha) - 2$.

*Proof.* Without loss of generality, we only consider datapoints in the cluster $\mathcal{C}_{+\mu_1} \cup \mathcal{N}_{+\mu_1}$. According to (D1) in Lemma 4.4, we have that under a good run, $|\mathcal{J}_P^{i,(0)}| \geq m/7, |\mathcal{J}_N^{i,(0)}| \geq m/7$ for each $i \in [n]$. For $x_k \in \mathcal{C}_{+\mu_1}$, by Corollary A.5, we have

$$\langle w_j^{(s)}, x_k \rangle > 0$$

for all $j \in \mathcal{J}_P^{k,(0)}$ and $0 \leq s \leq 1/(\sqrt{n}p\alpha) - 2$; and

$$\langle w_j^{(s)}, x_k \rangle \leq \frac{\alpha}{\sqrt{m}} \|\mu\|^2$$

for all $j \in \mathcal{J}_{\mathbb{N}}$ and $0 \le s \le 1/(\sqrt{n}p\alpha) - 2$. Then for $1 \le t \le 1/(\sqrt{n}p\alpha) - 2$, we have

$$\sum_{j=1}^{m} a_j \phi(\langle w_j^{(t)}, x_k \rangle) \ge \sum_{j \in \mathcal{J}_{\mathbb{P}}^{k,(0)}} \frac{1}{\sqrt{m}} \phi(\langle w_j^{(t)}, x_k \rangle) - \sum_{j:a_j<0} \frac{1}{\sqrt{m}} \phi(\langle w_j^{(t)}, x_k \rangle)$$

$$\ge \sum_{j \in \mathcal{J}_{\mathbb{P}}^{k,(0)}} \sum_{s=0}^{t-1} \frac{1}{\sqrt{m}} \langle w_j^{(s+1)} - w_j^{(s)}, x_k \rangle - \sum_{j:a_j<0} \frac{\alpha}{m} \|\mu\|^2$$

$$\ge \frac{\alpha p t}{4nm} |\mathcal{J}_{\mathbb{P}}^{k,(0)}| - \frac{\alpha |\mathcal{J}_{\mathbb{N}}|}{m} \|\mu\|^2$$

$$\ge \frac{\alpha p t}{28n} - \alpha \|\mu\|^2 > 0,$$

where the first inequality uses $\phi(x) \ge 0, \forall x$; the second inequality uses the definition of $\mathcal{J}_{\mathbb{P}}^{k,(0)}$ and (E2) in Corollary A.5; the third inequality uses (A.36) in Corollary A.5; and the last inequality is from Assumption (A2). For $x_k \in \mathcal{N}_{+\mu_1}$, similarly we have

$$\sum_{j=1}^{m} a_j \phi(\langle w_j^{(t)}, x_k \rangle) \le - \sum_{j \in \mathcal{J}_{\mathbb{N}}^{k,(0)}} \frac{1}{\sqrt{m}} \phi(\langle w_j^{(t)}, x_k \rangle) + \sum_{j:a_j>0} \frac{1}{\sqrt{m}} \phi(\langle w_j^{(t)}, x_k \rangle)$$

$$\le - \sum_{j \in \mathcal{J}_{\mathbb{N}}^{k,(0)}} \sum_{s=1}^{t} \frac{1}{\sqrt{m}} \langle w_j^{(s)} - w_j^{(s-1)}, x_k \rangle + \sum_{j:a_j>0} \frac{\alpha}{\sqrt{m}} \|\mu\|^2$$

$$\le -(\frac{\alpha p t}{28n} - \alpha \|\mu\|^2) < 0.$$

Thus our classifier can correctly classify all training datapoints for $1 \le t \le 1/(\sqrt{n}p\alpha) - 2$. $\qquad \square$

### A.5.2 PROOF OF THEOREM 4.7: GENERALIZATION

Before proceeding with the proof of Theorem 4.7, we first state a technical lemma:

**Lemma A.9.** *Suppose that $\|W\| > 0$. Then there exists a constant $c > 0$ such that*

$$\mathbb{P}_{(x,\widetilde{y}) \sim P_{clean}}(\widetilde{y} \ne \mathrm{sgn}(f(x;W))) \le \max_{\nu \in \mathsf{centers}} 2 \exp\left( -c \left( \frac{\mathbb{E}_{x \sim N(\nu, I_p)}[f(x;W)]}{\|W\|_F} \right)^2 \right).$$

*Proof.* It suffices to prove that for each $\nu \in \mathsf{centers}$,

$$\mathbb{P}_{x \sim N(\nu, I_p)}(y_\nu f(x;W) < 0) \le 2 \exp\left( -c \left( \frac{\mathbb{E}_{x \sim N(\nu, I_p)}[f(x;W)]}{\|W\|_F} \right)^2 \right). \tag{A.63}$$

Then applying the law of total expectation, we have

$$\mathbb{P}_{(x,\widetilde{y}) \sim P_{\mathrm{clean}}}(\widetilde{y} \ne \mathrm{sgn}(f(x;W))) = \frac{1}{4} \sum_{\nu \in \mathsf{centers}} \mathbb{P}_{x \sim N(\nu, I_p)}(y_\nu \ne \mathrm{sgn}(f(x;W)))$$

$$\le \frac{1}{2} \sum_{\nu \in \mathsf{centers}} \exp\left( -c \left( \frac{\mathbb{E}_{x \sim N(\nu, I_p)}[f(x;W)]}{\|W\|_F} \right)^2 \right)$$

$$\le \max_{\nu \in \mathsf{centers}} 2 \exp\left( -c \left( \frac{\mathbb{E}_{x \sim N(\nu, I_p)}[f(x;W)]}{\|W\|_F} \right)^2 \right).$$

Since for each $\nu$, $N(\nu, I_p)$ is 1-strongly log-concave, we plug in $\lambda = 1$ in the proof of Lemma 4.1 in Frei et al. (2022b). Then (A.63) is obtained.

$\qquad \square$

Our next theorem shows that the generalization risk is small for large $t$. Recall the definition of $\mathcal{J}_1$ and $\mathcal{J}_2$, we equivalently write them as

$$\mathcal{J}_1 = \mathcal{J}_{+\mu_1,\mathrm{P}}^{20\varepsilon} = \{j \in [m] : a_j > 0, D_{+\mu_1,j}^{(0)} > n^{1/2-20\varepsilon}, d_{+\mu_1,j}^{(0)} < \min\{c_{+\mu_1}, c_{-\mu_1}\} - 2n_{\pm\mu_1} - \sqrt{n}\};$$

$$\mathcal{J}_2 = \mathcal{J}_{+\mu_1,\mathrm{N}}^{20\varepsilon} \cup \mathcal{J}_{-\mu_1,\mathrm{N}}^{20\varepsilon} = \{j \in [m] : a_j < 0, D_{\nu,j}^{(0)} > n^{1/2-20\varepsilon},$$
$$d_{\nu,j}^{(0)} < \min\{c_\nu, c_{-\nu}\} - 2n_{\pm\mu_1} - \sqrt{n}, \nu \in \{\pm\mu_1\}\}.$$

Here $\mathcal{J}_{+\mu_1,\mathrm{P}}^{20\varepsilon}, \mathcal{J}_{+\mu_1,\mathrm{N}}^{20\varepsilon}$, and $\mathcal{J}_{-\mu_1,\mathrm{N}}^{20\varepsilon}$ are defined in (A.14). By Lemma 4.4, we know that under a good run,

$$|\mathcal{J}_1| \geq \frac{m}{n^{10\varepsilon}}, \quad |\mathcal{J}_2| \geq (1 - \frac{10}{n^{20\varepsilon}})|\mathcal{J}_\mathrm{N}|. \tag{A.64}$$

**Theorem 4.7.** *Suppose that Assumptions (A1)-(A6) hold. Under a good run, for $Cn^{10\varepsilon} \leq t \leq \sqrt{n}$, the generalization error of classifier $\mathrm{sgn}(f(x, W^{(t)}))$ has an upper bound*

$$\mathbb{P}_{(x,y)\sim P_{clean}}(y \neq \mathrm{sgn}(f(x; W^{(t)}))) \leq \exp\left(-\Omega\left(\frac{n^{1-20\varepsilon}\|\mu\|^4}{p}\right)\right).$$

*Proof.* Without loss of generality, we consider $x$ follows $N(+\mu_1, I_p)$. Then we have

$$\mathbb{E}_x[yf(x, W^{(t)})] = \sum_{j=1}^m a_j \mathbb{E}_x[\phi(\langle w_j^{(t)}, x \rangle)]$$

$$\geq \frac{1}{\sqrt{m}}\Big[\sum_{j:a_j>0} \phi(\langle w_j^{(t)}, \mathbb{E}[x] \rangle) - \sum_{j:a_j<0} \mathbb{E}_x[\phi(\langle w_j^{(t)}, x \rangle)]\Big] \tag{A.65}$$

$$\geq \frac{1}{\sqrt{m}} \sum_{j:j\in\mathcal{J}_1} \phi(\langle w_j^{(t)}, \mu_1 \rangle) - \frac{1}{\sqrt{m}} \sum_{j:a_j<0} \mathbb{E}_x[\phi(\langle w_j^{(t)}, x \rangle)],$$

where the first inequality uses Jensen's inequality. By Lemma A.6, we have that for $j \in \mathcal{J}_1$,

$$\langle w_j^{(t)}, \mu_1 \rangle = \sum_{s=0}^{t-1} \langle w_j^{(s+1)} - w_j^{(s)}, \mu_1 \rangle + \langle w_j^{(0)}, \mu_1 \rangle$$

$$\geq \frac{\alpha}{4n\sqrt{m}} \sum_{s=0}^{t-1} D_{+\mu_1,j}^{(s)} \|\mu\|^2 - \omega_{\mathrm{init}}\sqrt{3mp/2}\|\mu\|$$

$$\geq \frac{\alpha\|\mu\|^2}{4n\sqrt{m}}\left[n^{1/2-20\varepsilon} + (c_{+\mu_1} - n_{+\mu_1} - d_{-\mu_1,j}^{(0)})(t-1)\right] - \omega_{\mathrm{init}}\sqrt{3mp/2}\|\mu\| \tag{A.66}$$

$$\geq \frac{\alpha\|\mu\|^2}{4n\sqrt{m}}(c_{+\mu_1} - n_{+\mu_1} - d_{-\mu_1,j}^{(0)})(t-1) > 0,$$

where the first inequality is from Lemma A.6 and (C1) in Lemma 4.3; the second inequality uses the property that for $j \in \mathcal{J}_1$, $D_{+\mu_1,j}^{(s)} \geq c_{+\mu_1} - n_{+\mu_1} - d_{-\mu_1}^{(0)}(j), s \geq 1$, which is also from Lemma A.6; and the third inequality uses Assumption (A5). It yields that

$$\sum_{j:j\in\mathcal{J}_1} \phi(\langle w_j^{(t)}, \mu_1 \rangle) \geq \frac{\alpha\|\mu\|^2(t-1)}{4n\sqrt{m}} \sum_{j\in\mathcal{J}_1} (c_{+\mu_1} - d_{-\mu_1}^{(0)}(j) - n_{+\mu_1}) \geq \frac{\alpha\|\mu\|^2(t-1)}{40\sqrt{m}}|\mathcal{J}_1|, \tag{A.67}$$

where the last inequality uses (D4) in Lemma 4.4. For the second term in (A.65), note that we have $\phi(\lambda x) = \lambda\phi(x), \forall \lambda > 0$, and by Jensen's inequality, $\phi(x_1 + x_2) \leq \phi(x_1) + \phi(x_2), \forall x_1, x_2 \in \mathbb{R}$. Then we have

$$\mathbb{E}_x[\phi(\langle w, x \rangle)] \leq \phi(\langle w, \mu_1 \rangle) + \mathbb{E}_x[\phi(\langle w, x - \mu_1 \rangle)] = \phi(\langle w, \mu_1 \rangle) + \sqrt{\frac{1}{2\pi}}\|w\|, \tag{A.68}$$

where the last equation uses the expectation of half-normal distribution. By Lemma A.3, we have $g_i^{(t)} \leq 1$, and

$$\|w_j^{(t+1)} - w_j^{(t)}\| = \|\frac{\alpha a_j}{n} \sum_{i=1}^{n} g_i^{(t)} \phi'(\langle w_j^{(t)}, x_i \rangle) y_i x_i\|$$

$$\leq \frac{\alpha}{n\sqrt{m}} \max_{i \in [n]} g_i^{(t)} \sqrt{\sum_{i=1}^{n} \|x_i\|^2 + \sum_{i \neq j} |\langle x_i, x_j \rangle|} \leq \frac{2\alpha\sqrt{p}}{\sqrt{mn}}, \quad 0 \leq t \leq 1/(\sqrt{n}p\alpha) - 2,$$

where the last inequality uses $\|x_i\|^2 \leq 2p$, $|\langle x_i, x_j \rangle| \leq 2\mu^2$, which comes from Lemma 4.1, and Assumption (A2). It yields that for each $j \in [m]$,

$$\|w_j^{(t)}\| \leq \sum_{\tau=0}^{t-1} \|w_j^{(\tau+1)} - w_j^{(\tau)}\| + \|w_j^{(0)}\| \leq \frac{2\alpha\sqrt{p}t}{\sqrt{nm}} + \|w_j^{(0)}\| \leq \frac{3\alpha\sqrt{p}t}{\sqrt{mn}}, \tag{A.69}$$

where the last inequality uses Lemma 4.3. Then we consider the decomposition of $\sum_{j:a_j<0} \phi(\langle w_j^{(t)}, \mu_1 \rangle)$:

$$\sum_{j:a_j<0} \phi(\langle w_j^{(t)}, \mu_1 \rangle) = \sum_{j \in \mathcal{J}_2} \phi(\langle w_j^{(t)}, \mu_1 \rangle) + \sum_{j \in \mathcal{J}_\mathbb{N}, j \notin \mathcal{J}_2} \phi(\langle w_j^{(t)}, \mu_1 \rangle).$$

For the first term, we have

$$\sum_{j \in \mathcal{J}_2} \phi(\langle w_j^{(t)}, \mu_1 \rangle) \leq \sum_{j \in \mathcal{J}_2} |\langle w_j^{(t)}, \mu_1 \rangle|$$

$$\leq \sum_{j \in \mathcal{J}_2} \left[ \left| \sum_{s=0}^{t-1} \langle w_j^{(s+1)} - w_j^{(s)}, \mu_1 \rangle \right| + |\langle w_j^{(0)}, \mu_1 \rangle| \right]$$

$$\leq \sum_{j \in \mathcal{J}_2} \left[ \left| \sum_{s=0}^{t-1} (\frac{\alpha\|\mu\|^2}{2n\sqrt{m}} D_{+\mu_1,j}^{(s)} + \frac{5\alpha\|\mu\|^2}{n\sqrt{mn}}) \right| + \omega_{\text{init}}\sqrt{3mp/2}\|\mu\| \right] \tag{A.70}$$

$$\leq \sum_{j \in \mathcal{J}_2} \left[ \frac{\alpha\|\mu\|^2}{2n\sqrt{m}}(n + \Delta_{\mu_1}(t-2)) + \frac{5\alpha\|\mu\|^2 t}{n\sqrt{mn}} + \omega_{\text{init}}\sqrt{3mp/2}\|\mu\| \right]$$

$$= \sum_{j \in \mathcal{J}_2} \frac{\alpha\|\mu\|^2}{2n\sqrt{m}}[n + 1 + (\Delta_{\mu_1} + 1)(t-2)] \leq \frac{\alpha\|\mu\|^2}{2n\sqrt{m}}[n + 1 + (\Delta_{\mu_1} + 1)(t-2)]|\mathcal{J}_2|,$$

where the third inequality uses (A.30) in Lemma A.4; the fourth inequality uses Lemma A.7; and the fiveth inequality uses Assumptions (A1) and (A5). For the second term, we have

$$\sum_{j \in \mathcal{J}_\mathbb{N}, j \notin \mathcal{J}_2} \phi(\langle w_j^{(t)}, \mu_1 \rangle)$$

$$\leq \sum_{j \in \mathcal{J}_\mathbb{N}, j \notin \mathcal{J}_2} \left[ \sum_{s=0}^{t-1} |\langle w_j^{(s+1)} - w_j^{(s)}, \mu_1 \rangle| + |\langle w_j^{(0)}, \mu_1 \rangle| \right]$$

$$\leq \sum_{j \in \mathcal{J}_\mathbb{N}, j \notin \mathcal{J}_2} \left[ \sum_{s=0}^{t-1} (\frac{\alpha\|\mu\|^2}{2n\sqrt{m}} |D_{+\mu_1,j}^{(s)}| + \frac{5\alpha\|\mu\|^2}{n\sqrt{mn}}) + \omega_{\text{init}}\sqrt{3mp/2}\|\mu\| \right] \tag{A.71}$$

$$\leq \sum_{j \in \mathcal{J}_\mathbb{N}, j \notin \mathcal{J}_2} \frac{\alpha t(\max_{\nu \in \{\pm\mu_1\}} \{c_\nu + n_{-\nu}\} + 1)\|\mu\|^2}{n\sqrt{m}}$$

$$\leq \frac{\alpha t n \|\mu\|^2}{n\sqrt{m}}(|\mathcal{J}_\mathbb{N}| - |\mathcal{J}_2|) \leq \frac{10\alpha t\|\mu\|^2}{n^{20\varepsilon}\sqrt{m}}|\mathcal{J}_\mathbb{N}|,$$

where the second inequality uses (A.30) in Lemma A.4; the third inequality uses Assumption (A5) and $|D_{\nu,j}^{(t)}| \leq \max\{c_\nu + n_{-\nu}, c_{-\nu} + n_\nu\}$, which comes from the definition of $D_{\nu,j}^{(t)}$; the fourth inequality uses $c_\nu + n_{-\nu} + 1 \leq n$ for all $\nu \in$ centers, and the last inequality uses (A.64). Combining (A.68), (A.69), (A.70), and (A.71), we have

$$
\sum_{j:a_j<0} \mathbb{E}_x[\phi(\langle w_j^{(t)}, x\rangle)] \leq \sum_{j:a_j<0} \phi(\langle w_j^{(t)}, \mu_1\rangle) + \sqrt{\frac{1}{2\pi}} \sum_{j:a_j<0} \|w_j^{(t)}\|
$$

$$
= \sum_{j \in \mathcal{J}_2} \phi(\langle w_j^{(t)}, \mu_1\rangle) + \sum_{j \in \mathcal{J}_\mathbb{N}, j \notin \mathcal{J}_2} \phi(\langle w_j^{(t)}, \mu_1\rangle) + \sqrt{\frac{1}{2\pi}} \sum_{j:a_j<0} \|w_j^{(t)}\|
$$

$$
\leq \frac{\alpha\|\mu\|^2 t\sqrt{m}}{2n}\Big[\frac{n+1}{t} + (\Delta_{\mu_1}+1) + \frac{20n}{n^{20\varepsilon}} + \frac{3\sqrt{2np}}{\sqrt{\pi}\|\mu\|^2}\Big].
$$

It follows that

$$
\mathbb{E}_{x\sim N(+\mu_1, I_p)}[yf(x, W^{(t)})]
$$
$$
\geq \frac{\alpha\|\mu\|^2(t-1)}{40m}|\mathcal{J}_1| - \frac{\alpha\|\mu\|^2 t}{2n}\Big[\frac{n+1}{t} + (\Delta_{\mu_1}+1) + \frac{20n}{n^{20\varepsilon}} + \frac{3\sqrt{2np}}{\sqrt{\pi}\|\mu\|^2}\Big]
$$
$$
\geq \frac{\alpha\|\mu\|^2 t}{2}\Big[\frac{1}{20n^{10\varepsilon}}(1-\frac{1}{t}) - \frac{2}{t} - \frac{\Delta_{\mu_1}+1}{n} - \frac{20}{n^{20\varepsilon}} - \frac{6\sqrt{p}}{\sqrt{2\pi n}\|\mu\|^2}\Big] \quad \text{(A.72)}
$$
$$
\geq \frac{\alpha\|\mu\|^2 t}{2}\Big[\frac{1}{20n^{10\varepsilon}}(1-\frac{1}{t}) - \frac{2}{t} - \frac{2\eta\sqrt{n\varepsilon\log(n)}+1}{n} - \frac{20}{n^{20\varepsilon}} - \frac{6}{\sqrt{2\pi}Cn}\Big] \geq \frac{\alpha\|\mu\|^2 t}{80n^{10\varepsilon}}
$$

for $t \geq Cn^{10\varepsilon}$ when $C$ is large enough. Here the second inequality uses $|\mathcal{J}_1| \geq mn^{-10\varepsilon}$; the third inequality uses (B3) in Lemma 4.1 and Assumption (A1); and the last inequality uses $\varepsilon < 0.01$. By (A.69), it follows that $\|W^{(t)}\|_F \leq 3\alpha t\sqrt{p/n}$. Thus we have

$$
\frac{\mathbb{E}_{x\sim N(+\mu_1, I_p)}[yf(x, W^{(t)})]}{\|W^{(t)}\|_F} \geq \frac{\sqrt{n}\|\mu\|^2}{240\sqrt{p}n^{10\varepsilon}}.
$$

This lower bound for the normalized margin can be easily extended to the other $\nu$'s. Applying Lemma A.9, we have

$$
\mathbb{P}_{(x,y)\sim P_{\text{clean}}}(y \neq \text{sgn}(f(x; W^{(t)}))) \leq 2\exp\Big(-\frac{cn^{1-20\varepsilon}\|\mu\|^4}{240^2 p}\Big) = \exp\big(-\Omega(\frac{n^{1-20\varepsilon}\|\mu\|^4}{p})\big).
$$

$\square$

**Lemma 4.6.** *Suppose that Assumptions (A1)-(A6) hold. Under a good run, we have that for $1 \leq t \leq \sqrt{n}$,*

$$
\frac{1}{|\mathcal{J}_1|}\sum_{j\in\mathcal{J}_1}\langle w_j^{(t)}, +\mu_1\rangle = \Omega\Big(\frac{\alpha\|\mu\|^2}{\sqrt{m}}t\Big);
$$

$$
\frac{1}{|\mathcal{J}_2|}\sum_{j\in\mathcal{J}_2}|\langle w_j^{(t)}, \mu_1\rangle| = O\Big(\frac{\alpha\|\mu\|^2}{\sqrt{m}} + \frac{\alpha\|\mu\|^2\sqrt{\log(n)}}{\sqrt{mn}}t\Big).
$$

*Proof.* This lemma is essentially implied by the proof of Lemma 4.7. By (A.66), we know that for all $j \in \mathcal{J}_1$,

$$
\langle w_j^{(t)}, +\mu_1\rangle > 0.
$$

Then note that $\langle w_j^{(t)}, +\mu_1\rangle = \phi(\langle w_j^{(t)}, +\mu_1\rangle)$. From this we have

$$
\frac{1}{|\mathcal{J}_1|}\sum_{j:j\in\mathcal{J}_1}\langle w_j^{(t)}, +\mu_1\rangle = \frac{1}{|\mathcal{J}_1|}\sum_{j:j\in\mathcal{J}_1}\phi(\langle w_j^{(t)}, +\mu_1\rangle) \geq \frac{\alpha\|\mu\|^2(t-1)}{40\sqrt{m}} = \Omega\Big(\frac{\alpha\|\mu\|^2 t}{\sqrt{m}}\Big),
$$

where the first inequality comes from (A.67). Recall that in Lemma A.7, $\Delta_{\mu_1}$ is defined as $|n_{+\mu_1} - n_{-\mu_1}| + \sqrt{n}$. Applying (B3) in Lemma 4.1, we have

$$
\begin{aligned}
|n_{+\mu_1} - n_{-\mu_1}| \leq & |n_{+\mu_1} - \eta(n_{+\mu_1} + c_{+\mu_1})| + |\eta(n_{+\mu_1} + c_{+\mu_1} - n/4)| \\
& + |\eta(n_{-\mu_1} + c_{-\mu_1} - n/4)| + |n_{-\mu_1} - \eta(n_{-\mu_1} + c_{-\mu_1})| \\
\leq & 4\sqrt{\varepsilon n \log(n)}.
\end{aligned}
$$

Then $\Delta_{\mu_1}$ is upper bounded by

$$
\Delta_{\mu_1} \leq \sqrt{n} + 4\sqrt{\varepsilon n \log(n)} = O(\sqrt{n \log(n)}).
$$

Combining the inequality above with equation (A.70), we have

$$
\frac{1}{|\mathcal{J}_2|} \sum_{j \in \mathcal{J}_2} |\langle w_j^{(t)}, \mu_1 \rangle| \leq \frac{\alpha \|\mu\|^2}{2n\sqrt{m}} [n + 1 + (\Delta_{\mu_1} + 1)(t-2)] = O\Big( \frac{\alpha\|\mu\|^2}{\sqrt{m}} + \frac{\alpha\|\mu\|^2 \sqrt{\log(n)} t}{\sqrt{mn}} \Big).
$$

$\square$

### A.5.3 PROOF OF THEOREM A.13: 1-STEP TEST ACCURACY

Before stating the proof, we begin with the necessary definitions and a preliminary result. Recall that $h_i^{(t)} = g_i^{(t)} - 1/2$ and the decomposition (A.28). When $t = 0$, we denote

$$
w_{j,\mathrm{T}}^{(1)} := w_j^{(0)} + \frac{\alpha a_j}{2n} \sum_{i=1}^n \phi'(\langle w_j^{(0)}, x_i \rangle) y_i x_i, \quad j \in [m] \tag{A.73}
$$

and $W_{\mathrm{T}}^{(1)} := [w_{1,\mathrm{T}}^{(1)}, \cdots, w_{m,\mathrm{T}}^{(1)}]^\top$. Next lemma shows that $W_{\mathrm{T}}^{(1)}$ is a good approximation of $W^{(1)}$ with a large probability.

**Lemma A.10.** *Suppose Assumptions (A1) and (A2) hold. Given $\{x_i\} \in \mathcal{G}_{data}$ and $W^{(0)} \in \mathcal{G}_W$, we have*

$$
|h_i^{(0)}| \leq p\omega_{init}\sqrt{3m}/2;
$$

$$
\|W_{\mathrm{T}}^{(1)} - W^{(1)}\|_F = \sqrt{\sum_{j=1}^m \|w_{j,\mathrm{T}}^{(1)} - w_j^{(1)}\|^2} \leq \frac{\alpha\omega_{init} p^{3/2}\sqrt{3m}}{\sqrt{n}}.
$$

*Proof.* Let $z_i^{(t)} = y_i f(x_i; W^{(t)})$. Note that $\ell'(z) = -1/(1 + \exp(z))$, we have $|-\ell'(z) - 1/2| \leq |z|/2$. It yields that

$$
\begin{aligned}
|h_i^{(0)}| &\leq \frac{1}{2}|z_i^{(0)}| \leq \frac{1}{2} \sum_{j=1}^m |a_j \langle w_j^{(0)}, x_i \rangle| \leq \frac{1}{2} \sqrt{\sum_{j=1}^m a_j^2 \sum_{j=1}^m \|w_j^{(0)}\|^2 \cdot \|x\|^2} \\
&= \frac{1}{2}\|W^{(0)}\|_F \cdot \|x_i\| \leq \frac{1}{2} p\omega_{\mathrm{init}} \sqrt{3m},
\end{aligned} \tag{A.74}
$$

where the first inequality uses $h_i^{(t)} = g_i^{(t)} - 1/2$ and $g_i^{(t)} := -\ell'(z_i^{(t)})$; the second inequality uses triangle inequality; the third inequality uses Cauchy-Schwarz inequality; and the last inequality uses (B1) in Lemma 4.1 and (C1) in Lemma 4.3. Denote $h_{\max} = \max_{i \in [n]} |h_i^{(0)}|$. Then we have

$$
\begin{aligned}
\|w_{j,\mathrm{T}}^{(1)} - w_j^{(1)}\| &= \frac{\alpha}{n\sqrt{m}} \|\sum_{i=1}^n h_i^{(0)} \phi'(\langle w_j^{(0)}, x_i \rangle) y_i x_i\| \\
&\leq \frac{\alpha h_{\max}}{n\sqrt{m}} \sqrt{\sum_{i=1}^n \|x_i\|^2 + n(n-1) \max_{i \neq j} |x_i^\top x_j|} \\
&\leq \frac{\alpha h_{\max}}{n\sqrt{m}} \sqrt{4np} \leq \frac{\sqrt{3}\alpha\omega_{\mathrm{init}} p^{3/2}}{\sqrt{n}},
\end{aligned}
$$

where the second inequality uses $\|x_i\|^2 \leq 2p$ and $p \geq Cn^2\|\mu\|^2$, which come from (B1) and (B2) in Lemma 4.1 and Assumption (A2) respectively, and the third inequality uses (A.74). Further we have

$$\|W_{\text{T}}^{(1)} - W^{(1)}\|_F = \sqrt{\sum_{j=1}^{m} \|w_{j,\text{T}}^{(1)} - w_j^{(1)}\|^2} \leq \frac{\alpha\omega_{\text{init}}p^{3/2}\sqrt{3m}}{\sqrt{n}}.$$

$\square$

**Lemma A.11.** *Suppose that Assumptions (A1)-(A6) hold. Given $X \in \mathcal{G}_{data}$, for each $j \in [m]$, we have*

$$n/24 \leq \text{Var}(D_{+\mu_1,j}^{(0)}) \leq n/2;$$

$$\mathbb{E}\big[|D_{+\mu_1,j}^{(0)}) - \mathbb{E}[D_{+\mu_1,j}^{(0)}]|^3\big] \leq n^{3/2}.$$

*Proof.* Recall that $\mathcal{A}_1 = \mathcal{C}_{+\mu_1} \cup \mathcal{N}_{-\mu_1}$, $\mathcal{A}_2 = \mathcal{C}_{-\mu_1} \cup \mathcal{N}_{+\mu_1}$. According to equation (A.18), we have

$$D_{+\mu_1,j}^{(0)} = \sum_{i \in \mathcal{A}_1} \mathbb{I}(z_i > 0) - \sum_{i \in \mathcal{A}_2} \mathbb{I}(z_i > 0). \tag{A.75}$$

According to Lemma A.15, we have

$$\text{Var}(D_{+\mu_1,j}^{(0)}) = \mathbb{E}_B[f_1(b_1, \cdots, b_n)] \geq \frac{1}{2}\mathbb{E}_{B'}[f_1(b_1', \cdots, b_n')]$$

$$= \frac{1}{2}\text{Var}_{B'}\big(\sum_{i \in \mathcal{A}_1} b_i' - \sum_{i \in \mathcal{A}_2} b_i'\big) = \frac{|\mathcal{A}_1| + |\mathcal{A}_2|}{8} \geq \frac{n}{24},$$

where $f_1(b_1, \cdots, b_n) := (\sum_{i \in \mathcal{A}_1} b_i - \sum_{i \in \mathcal{A}_2} b_i - (|\mathcal{A}_1| - |\mathcal{A}_2|)/2)^2 \geq 0$, and $b_i'$ are i.i.d Bernoulli random variables defined in Lemma A.15, and the last inequality is from (A.17). On the other side, similarly we have

$$\text{Var}(D_{+\mu_1,j}^{(0)}) \leq 2\mathbb{E}_{B'}[f_1(b_1', \cdots, b_n')] = (|\mathcal{A}_1| + |\mathcal{A}_2|)/2 \leq n/2, \tag{A.76}$$

where the last inequality is from (B3) in Lemma 4.1. Denote $f_2(b_1, \cdots, b_n) := (\sum_{i \in \mathcal{A}_1} b_i - \sum_{i \in \mathcal{A}_2} b_i - (|\mathcal{A}_1| - |\mathcal{A}_2|)/2)^4 \geq 0$, then we have

$$\mathbb{E}[|D_{+\mu_1,j}^{(0)} - \mathbb{E}[D_{+\mu_1,j}^{(0)}]|^4] = \mathbb{E}_B[f_2(b_1, \cdots, b_n)] \leq 2\mathbb{E}_{B'}[f_2(b_1', \cdots, b_n')]$$

$$= 2\mathbb{E}_{B'}\Big[\big[\sum_{i \in \mathcal{A}_1}(b_i' - \frac{1}{2}) - \sum_{i \in \mathcal{A}_2}(b_i' - \frac{1}{2})\big]^4\Big]$$

$$\leq 16\mathbb{E}_{B'}\Big[\big[\sum_{i \in \mathcal{A}_1}(b_i' - \frac{1}{2})\big]^4 + \big[\sum_{i \in \mathcal{A}_2}(b_i' - \frac{1}{2})\big]^4\Big]$$

$$\leq 4(|\mathcal{A}_1|^2 + |\mathcal{A}_2|^2) \leq n^2, \tag{A.77}$$

where the first inequality uses Lemma A.15; the second inequality uses $(a + b)^4 \leq 8(a^4 + b^4)$; the third inequality uses the formula of the fourth central moment of a binomial distribution with parameter equal to $1/2$, i.e. $\mu_4(\text{B}(n, 1/2)) = n(1 + (3n - 6)/4)/4 \leq n^2/4$; and the last inequality is from (B3) in Lemma 4.1. Combining (A.76) and (A.77), we have

$$\mathbb{E}\big[|D_{+\mu_1,j}^{(0)}) - \mathbb{E}[D_{+\mu_1,j}^{(0)}]|^3\big] \leq \sqrt{\text{Var}(D_{+\mu_1,j}^{(0)})\mathbb{E}[|D_{+\mu_1,j}^{(0)} - \mathbb{E}[D_{+\mu_1,j}^{(0)}]|^4]} \leq n^{3/2}$$

by applying the Cauchy-Schwarz inequality.

$\square$

**Lemma A.12.** *Suppose that Assumptions (A1)-(A6) hold. Given $X = [x_1, \cdots, x_n]^\top \in \mathcal{G}_{data}$, we have*

$$\mathbb{P}\big(\big|\sum_{j=1}^{m} a_j\phi(a_j D_{+\mu_1,j}^{(0)}) - \frac{1}{2}\mathbb{E}[D_{+\mu_1,j}^{(0)}]\big| > t\big) \leq 2\bar{\Phi}\big(\frac{t\sqrt{m}}{3C_n\sqrt{n}\varepsilon}\big) + \frac{C}{\sqrt{m}};$$

$$\mathbb{P}\Big(\Big|\sum_{j=1}^{m} a_j |a_j D_{+\mu_1,j}^{(0)}|\Big| > t\Big) \le 2\bar{\Phi}\Big(\frac{t\sqrt{m}}{3C_n\sqrt{n\varepsilon}}\Big) + \frac{C}{\sqrt{m}}.$$

*Proof.* In this proof, by convention all $\mathbb{P}(\cdot), \mathbb{E}[\cdot], \mathrm{Var}(\cdot), \rho(\cdot)$ are implicitly conditioned on a fixed $X$. Denote the expectation of $D_{+\mu_1,j}^{(0)}$ by $e_{+\mu_1}$. Note that conditioning on $X$, $\{a_j\phi(a_j D_{+\mu_1,j}^{(0)})\}_{j\ge 1}$ are i.i.d, and the expectation of $D_{+\mu_1,j}^{(0)}$ is

$$e_{+\mu_1} = (c_{+\mu_1} - n_{+\mu_1} - c_{-\mu_1} + n_{-\mu_1})/2 \le 2C_n\sqrt{n\varepsilon}, \tag{A.78}$$

where the inequality uses (B3) in Lemma 4.1. By Lemma A.11, we have

$$\frac{n}{24} \le \mathrm{Var}\big(D_{+\mu_1,j}^{(0)}\big) \le \frac{n}{2}; \quad \rho(D_{+\mu_1,j}^{(0)}) \le n^{3/2}. \tag{A.79}$$

Denote

$$\sigma_{+\mu_1}^2 = \mathrm{Var}\big(ma_j\phi(a_j D_{+\mu_1,j}^{(0)})\big); \quad \rho_{+\mu_1} = \rho(ma_j\phi(a_j D_{+\mu_1,j}^{(0)})).$$

Combining (A.79) and results in Lemma A.14, we have

$$\mathbb{E}[ma_j\phi(a_j D_{+\mu_1,j}^{(0)})] = \frac{e_{+\mu_1}}{2}; \quad \max\Big\{\frac{n}{48}, \frac{e_{+\mu_1}^2}{4}\Big\} \le \sigma_{+\mu_1}^2 \le \max\Big\{\frac{n}{2}, \frac{e_{+\mu_1}^2}{2}\Big\}; \quad \rho_{+\mu_1} \le 32\max\{n^{3/2}, |e_{+\mu_1}|^3\}. \tag{A.80}$$

Applying Berry-Esseen theorem, we have

$$\mathbb{P}\Big(\Big|\sum_{j=1}^{m} a_j\phi(a_j D_{+\mu_1,j}^{(0)}) - \frac{1}{2}e_{+\mu_1}\Big| > t\Big) \le 2\bar{\Phi}\Big(\frac{t\sqrt{m}}{\sigma_{+\mu_1}}\Big) + \frac{C_{\mathrm{BE}}\rho_{+\mu_1}}{\sigma_{+\mu_1}^3\sqrt{m}} \le 2\bar{\Phi}\Big(\frac{t\sqrt{m}}{\sqrt{n} + 2C_n\sqrt{n\varepsilon}}\Big) + \frac{C}{\sqrt{m}}$$

for some universal constant $C > 0$. Here the second inequality uses $\sigma_{+\mu_1}^2 \le (\sqrt{n} + |e_{+\mu_1}|)^2$, which comes from (A.80), and the last inequality uses (A.78). By the symmetry of $a_j$, we have

$$\mathbb{E}[ma_j|a_j D_{+\mu_1,j}^{(0)}|] = 0; \quad \mathrm{Var}(ma_j|a_j D_{+\mu_1,j}^{(0)}|) = \mathbb{E}[(D_{+\mu_1,j}^{(0)})^2]; \quad \rho(ma_j|a_j D_{+\mu_1,j}^{(0)}|) = \mathbb{E}[|D_{+\mu_1,j}^{(0)}|^3].$$

By (A.79), we have

$$\frac{n}{24} + e_{+\mu_1}^2 \le \mathbb{E}[(D_{+\mu_1,j}^{(0)})^2] \le \frac{n}{2} + e_{+\mu_1}^2; \quad \mathbb{E}[|D_{+\mu_1,j}^{(0)}|^3] \le 8(\rho(D_{+\mu_1,j}^{(0)}) + |e_{+\mu_1}|^3) \le 8(n^{3/2} + |e_{+\mu_1}|^3). \tag{A.81}$$

Similarly, applying Berry-Esseen theorem, we have

$$\mathbb{P}\Big(\Big|\sum_{j=1}^{m} a_j |a_j D_{+\mu_1,j}^{(0)}|\Big| > t\Big) \le 2\bar{\Phi}\Big(\frac{t\sqrt{m}}{\sqrt{n} + 2C_n\sqrt{n\varepsilon}}\Big) + \frac{C}{\sqrt{m}},$$

where the inequality uses $\mathrm{Var}(ma_j|a_j D_{+\mu_1,j}^{(0)}|) \le (\sqrt{n} + |e_{+\mu_1}|)^2$ and (A.78). Then the results of this lemma are proved by noting that $C_n\sqrt{\varepsilon} \ge 1$ for large enough $n$. $\qquad\square$

**Theorem A.13.** *Suppose that Assumptions (A1)-(A6) hold. With probability at least $1 - O(1/\sqrt{m}) - O(n^{-\varepsilon})$ over the initialization of the weights and the generation of training data, after one iteration, the classifier $\mathrm{sgn}(f(x, W^{(1)}))$ exhibits a generalization risk with the following bounds:*

$$\tfrac{1}{2}(1 - n^{-\varepsilon}) \le \mathbb{P}_{(x,y)\sim P_{clean}}(y \ne \mathrm{sgn}(f(x; W^{(1)}))) \le \tfrac{1}{2}(1 + n^{-\varepsilon}).$$

*Proof.* For any given training data $X \in \mathcal{G}_{\mathrm{data}}$, denote the expectation of $D_{\nu,j}^{(0)}$ by $e_\nu$, i.e.

$$e_\nu := \mathbb{E}[D_{\nu,j}^{(0)}|X] = (c_\nu - n_\nu - c_{-\nu} + n_{-\nu})/2, \quad \nu \in \{\pm\mu_1, \pm\mu_2\}, \tag{A.82}$$

and a set of parameters $\mathcal{G}_X$:

$$\mathcal{G}_X := \Big\{ (a, W^{(0)}) : \big| \sum_{j=1}^{m} a_j \phi(a_j D_{\nu,j}^{(0)}) - e_\nu / 2 \big| \le 3 C_n \sqrt{n\varepsilon/m} \log(m),$$

$$\big| \sum_{j=1}^{m} a_j |a_j D_{\nu,j}^{(0)}| \big| \le 3 C_n \sqrt{n\varepsilon/m} \log(m), a \in \mathcal{G}_A, W^{(0)} \in \mathcal{G}_W \Big\}.$$

Applying the union bound, we have

$$\mathbb{P}(\mathcal{G}_X | X \in \mathcal{G}_{\text{data}}) \ge 1 - \exp(-\Omega(\log^2(m))) - \frac{2C}{\sqrt{m}} - n^{-\varepsilon}$$

by Lemma A.12 and 4.3. Further we have

$$\mathbb{P}((a, W^{(0)}) \in \mathcal{G}_X, X \in \mathcal{G}_{\text{data}}) \ge \mathbb{P}(\mathcal{G}_X | X \in \mathcal{G}_{\text{data}}) \mathbb{P}(X \in \mathcal{G}_{\text{data}})$$

$$\ge 1 - \exp(-\log^2(m)/2) - \frac{2C}{\sqrt{m}} - 2n^{-\varepsilon}$$

$$\ge 1 - \frac{3C}{\sqrt{m}} - 2n^{-\varepsilon}.$$

Define events $\mathcal{F}_{\text{test},\nu}$ for test data:

$$\mathcal{F}_{\text{test},\nu} = \{ x \in \mathbb{R}^p : |\|x\|^2 - p - \|\mu\|^2| \le 10\sqrt{p\log(n)};$$

$$|\langle x, x_i \rangle - \langle \nu, \bar{x}_i \rangle| \le 10\sqrt{p\log(n)} \text{ for all } i \in [n]\}, \quad \nu \in \{\pm\mu_1, \pm\mu_2\}.$$

Treat $\{x\} \cup \{x_i\}_{i=1}^n$ as a new 'training' set with $n+1$ datapoints. Following the proof procedure in Lemma 4.1, we can show that $\mathbb{P}_{x \sim N(\nu, I_p)}(x \in \mathcal{F}_{\text{test}} | X \in \mathcal{G}_{\text{data}}) \ge 1 - n^{-\varepsilon}$, where $\mathcal{F}_{\text{test}} := \cup_{\nu \in \{\pm\mu_1, \pm\mu_2\}} \mathcal{F}_{\text{test},\nu}$. And $\mathcal{F}_{\text{test}}$ is a symmetric set for $x$, i.e., if $x \in \mathcal{F}_{\text{test}}$, then $-x$ also belongs to $\mathcal{F}_{\text{test}}$. In the remaining proof, by convention all probabilities and expectations are implicitly conditioned on fixed $X \in \mathcal{G}_{\text{data}}$ and $a, W^{(0)} \in \mathcal{G}_X$. Therefore, to simplify notation, we write $\mathbb{P}(\cdot)$ and $\mathbb{E}[\cdot]$ to denote $\mathbb{P}(\cdot | a, W^{(0)}, \{x_i\})$ and $\mathbb{E}[\cdot | a, W^{(0)}, \{x_i\}]$, respectively. In other words, the randomness is over the test data $(x, y)$, conditioned on a fixed initialization and training data. We first look at the clusters centered at $\pm\mu_1$, i.e. $x \sim N(\pm\mu_1, I_p), y = 1$. Then we have

$$\mathbb{P}_{x \sim N(\pm\mu_1, I_p)}(y \ne \text{sgn}(f(x, W^{(1)}))) = \mathbb{P}_{x \sim N(\pm\mu_1, I_p)}(f(x, W^{(1)}) \le 0)$$
$$= \frac{1}{2}\mathbb{P}_{x \sim N(\mu_1, I_p)}(f(x, W^{(1)}) \le 0) + \frac{1}{2}\mathbb{P}_{x \sim N(\mu_1, I_p)}(f(-x, W^{(1)}) \le 0). \tag{A.83}$$

Note that given $W^{(0)}$ and $X$, we have with probability 1 that

$$|f(x; W^{(1)}) - f(x; W^{(1)} - W^{(0)})| = \Big| \sum_{j=1}^{m} a_j [\phi(\langle w_j^{(1)}, x \rangle) - \phi(\langle w_j^{(1)} - w_j^{(0)}, x \rangle)] \Big|$$

$$\le \sum_{j=1}^{m} |a_j \langle w_j^{(0)}, x \rangle| \le \sqrt{\sum_{j=1}^{m} a_j^2 \sum_{j=1}^{m} \|w_j^{(0)}\|^2 \cdot \|x\|^2} \tag{A.84}$$

$$= \|W^{(0)}\|_F \cdot \|x\| \le \omega_{\text{init}} \sqrt{3mp/2} \|x\|,$$

where the first inequality comes from the 1-Lipschitz continuity of $\phi(\cdot)$; the second inequality uses Cauchy-Schwarz inequality; and the last inequality uses Lemma 4.3. Next, recall that $W_{\text{T}}$ is defined

as in (A.73). By the same argument above, we have

$$
\begin{aligned}
& |f(x; W^{(1)} - W^{(0)}) - f(x; W_{\mathtt{T}}^{(1)} - W^{(0)})| \\
& = \Big| \sum_{j=1}^{m} a_j [\phi(\langle w_j^{(1)} - w_j^{(0)}, x\rangle) - \phi(\langle w_{j,\mathtt{T}}^{(1)} - w_j^{(0)}, x\rangle)] \Big| \\
& \leq \sum_{j=1}^{m} |a_j \langle w_j^{(1)} - w_{j,\mathtt{T}}^{(1)}, x\rangle| \leq \sqrt{\sum_{j=1}^{m} a_j^2 \sum_{j=1}^{m} \|w_j^{(1)} - w_{j,\mathtt{T}}^{(1)}\|^2 \cdot \|x\|^2} = \|W^{(1)} - W_{\mathtt{T}}^{(1)}\|_F \cdot \|x\| \\
& \leq \alpha \omega_{\text{init}} p \sqrt{3mp/n}\|x\| \leq \omega_{\text{init}} \sqrt{3mp/n}\|x\|,
\end{aligned}
\tag{A.85}
$$

where the first inequality comes from the 1-Lipschitz continuity of $\phi(\cdot)$; the second inequality uses Cauchy-Schwarz inequality; the third inequality uses Lemma A.10; and the last inequality uses Assumption (A3). Using (A.84) and (A.85), we have by the triangle inequality that

$$
|f(x; W^{(1)}) - f(x; W_{\mathtt{T}}^{(1)} - W^{(0)})| \leq 2\omega_{\text{init}}\sqrt{mp}\|x\| =: \epsilon_x, \quad \text{that for any } x \in \mathbb{R}^p. \tag{A.86}
$$

Recall that

$$
\langle w_{j,\mathtt{T}}^{(1)} - w_j^{(0)}, x\rangle = \frac{\alpha a_j}{2n} \sum_{i=1}^{n} \phi'(\langle w_j^{(0)}, x_i\rangle)\langle y_i x_i, x\rangle.
$$

Then under a good run, for $x \in \mathcal{F}_{\text{test}}$, we have that with probability 1,

$$
\Big| \langle w_{j,\mathtt{T}}^{(1)} - w_j^{(0)}, x\rangle - \frac{\alpha a_j}{2n} D_{+\mu_1,j}^{(0)}\|\mu\|^2 \Big| \leq \frac{\alpha}{\sqrt{m}} C_n \sqrt{p},
$$

where $C_n = 10\sqrt{\log(n)}$ and the inequality uses the definition of $\mathcal{F}_{\text{test}}$. It yields that

$$
\Big| f(x; W_{\mathtt{T}}^{(1)} - W^{(0)}) - \sum_{j=1}^{m} \frac{\alpha a_j}{2n} \phi(a_j D_{+\mu_1,j}^{(0)})\|\mu\|^2 \Big| \leq \alpha C_n \sqrt{p}. \tag{A.87}
$$

According to the definition of $\mathcal{G}_X$, we have

$$
\Big| \sum_{j=1}^{m} \frac{\alpha a_j}{2n} \phi(a_j D_{+\mu_1,j}^{(0)})\|\mu\|^2 - \frac{\alpha\|\mu\|^2}{4n} e_{+\mu_1} \Big| \leq \frac{3\alpha C_n \sqrt{\varepsilon}\log(m)}{2\sqrt{mn}}\|\mu\|^2. \tag{A.88}
$$

Combining (A.86)-(A.88), we have

$$
\Big| f(x; W^{(1)}) - \frac{\alpha\|\mu\|^2}{4n} e_{+\mu_1} \Big| \leq \epsilon_x + \alpha C_n \sqrt{p} + \frac{3\alpha C_n \sqrt{\varepsilon}\log(m)}{2\sqrt{mn}}\|\mu\|^2. \tag{A.89}
$$

The above inequality immediately implies that

$$
\mathbb{P}(f(x; W^{(1)}) \leq 0 | \mathcal{F}_{\text{test}}) \geq \mathbb{P}\Big( \frac{\alpha\|\mu\|^2}{2n} e_{+\mu_1} \leq -\epsilon_x - \alpha C_n \sqrt{p} - \frac{3\alpha C_n \sqrt{\varepsilon}\log(m)}{2\sqrt{mn}}\|\mu\|^2 \Big| \mathcal{F}_{\text{test}} \Big).
\tag{A.90}
$$

Similar to (A.89), for $-x \sim N(-\mu_1, I_p)$, we have

$$
\Big| f(-x; W^{(1)}) - \frac{\alpha\|\mu\|^2}{2n} e_{-\mu_1} \Big| \leq \epsilon_x + \alpha C_n \sqrt{p} + \frac{3\alpha C_n \sqrt{\varepsilon}\log(m)}{2\sqrt{mn}}\|\mu\|^2.
$$

Note that by definition, $e_{-\mu_1} = -e_{+\mu_1}$, the above inequality immediately implies that

$$
\mathbb{P}(f(-x; W^{(1)}) \leq 0 | \mathcal{F}_{\text{test}}) \geq \mathbb{P}\Big( \frac{\alpha\|\mu\|^2}{2n} e_{+\mu_1} \geq \epsilon_x + \alpha C_n \sqrt{p} + \frac{3\alpha C_n \sqrt{\varepsilon}\log(m)}{2\sqrt{mn}}\|\mu\|^2 \Big| \mathcal{F}_{\text{test}} \Big).
\tag{A.91}
$$

According to the definition of $\mathcal{G}_{\text{test}}$, we have $\epsilon_x \leq 4\omega_{\text{init}}\sqrt{m}p^{3/2}$. According to the definition of $\mathcal{G}_{\text{data}}$, we have

$$|c_\nu - n_\nu - c_{-\nu} + n_{-\nu}| \geq |c_\nu - c_{-\nu}| - |n_\nu - n_{-\nu}| \geq |c_\nu + n_\nu - c_{-\nu} - n_{-\nu}| - 2|n_\nu - n_{-\nu}|$$
$$\geq (1 - 2\eta)n^{1/2-\varepsilon} \geq n^{1/2-\varepsilon}/2.$$

Thus we have $|e_{+\mu_1}| \geq n^{1/2-\varepsilon}/4$. It yields that

$$\frac{\alpha\|\mu\|^2}{2n}|e_{+\mu_1}| - \epsilon_x - \alpha C_n\sqrt{p} - \frac{3\alpha C_n\sqrt{\varepsilon}\log(m)}{2\sqrt{mn}}\|\mu\|^2$$
$$\geq \frac{\alpha\|\mu\|^2}{\sqrt{n}}\left(\frac{1}{8n^\varepsilon} - 4\sqrt{mn}p^{3/2}\frac{\omega_{\text{init}}}{\alpha\|\mu\|^2} - C_n\sqrt{\frac{np}{\|\mu\|^4}} - \frac{3C_n\sqrt{\varepsilon}\log(m)}{2\sqrt{m}}\right) \quad \text{(A.92)}$$
$$\geq \frac{\alpha\|\mu\|^2}{\sqrt{n}}\left(\frac{1}{8n^\varepsilon} - \frac{2}{m\sqrt{n}} - \frac{C_n}{3Cn^{0.01}} - \frac{3C_n}{2\sqrt{C}n^{0.01}}\right) > 0,$$

where the first inequality uses $|e_{+\mu_1}| \geq n^{1/2-\varepsilon}/4$ and $\epsilon_x \leq 4\omega_{\text{init}}\sqrt{m}p^{3/2}$; the second inequality uses Assumption (A5), (A1) and (A6); and the last inequality uses $n$ is large enough. Combining (A.90)-(A.92), we have

$$\mathbb{P}(f(x; W^{(1)}) \leq 0|\mathcal{F}_{\text{test}}) + \mathbb{P}(f(-x; W^{(1)}) \leq 0|\mathcal{F}_{\text{test}})$$
$$\geq \mathbb{P}(\frac{\alpha\|\mu\|^2}{2n}|e_{+\mu_1}| \geq \epsilon_x + \alpha C_n\sqrt{p} + \frac{3\alpha C_n\sqrt{\varepsilon}\log(m)}{2\sqrt{mn}}\|\mu\|^2|\mathcal{F}_{\text{test}}) = 1, \quad \text{(A.93)}$$

where the inequality uses $\epsilon_x \geq 0$. Following a similar procedure, for the other side, we have

$$\mathbb{P}(f(x; W^{(1)}) \leq 0|\mathcal{F}_{\text{test}}) + \mathbb{P}(f(-x; W^{(1)}) \leq 0|\mathcal{F}_{\text{test}})$$
$$\leq \mathbb{P}(\frac{\alpha\|\mu\|^2}{2n}|e_{+\mu_1}| \geq -\epsilon_x - \alpha C_n\sqrt{p} - \frac{3\alpha C_n\sqrt{\varepsilon}\log(m)}{2\sqrt{mn}}\|\mu\|^2|\mathcal{F}_{\text{test}}) = 1. \quad \text{(A.94)}$$

Combining (A.93) and (A.94), we have

$$\mathbb{P}(f(x; W^{(1)}) \leq 0|\mathcal{F}_{\text{test}}) + \mathbb{P}(f(-x; W^{(1)}) \leq 0|\mathcal{F}_{\text{test}}) = 1.$$

Following the same procedure, we have that for any $\nu \in \{\pm\mu_1, \pm\mu_2\}$,

$$\mathbb{P}_{x\sim N(\nu,I_p)}(yf(x; W^{(1)}) \leq 0|\mathcal{F}_{\text{test}}) + \mathbb{P}_{x\sim N(\nu,I_p)}(yf(-x; W^{(1)}) \leq 0|\mathcal{F}_{\text{test}}) = 1.$$

Then for $(x, y) \sim P_{\text{clean}}$, we have

$$\mathbb{P}_{(x,y)\sim P_{\text{clean}}}(yf(x; W^{(1)}) \leq 0) \geq \mathbb{P}(yf(x; W^{(1)}) \leq 0|\mathcal{F}_{\text{test}})\mathbb{P}(\mathcal{F}_{\text{test}}) \geq \frac{1}{2}(1 - n^{-\varepsilon});$$

$$\mathbb{P}_{(x,y)\sim P_{\text{clean}}}(yf(x; W^{(1)}) \leq 0) \leq \mathbb{P}(yf(x; W^{(1)}) \leq 0|\mathcal{F}_{\text{test}})\mathbb{P}(\mathcal{F}_{\text{test}}) + \mathbb{P}(\mathcal{F}_{\text{test}}^c) \leq \frac{1}{2}(1 + n^{-\varepsilon}).$$

$\square$

### A.6 Probability Lemmas

**Lemma A.14.** *Suppose we have a random variable $g$ that has finite $L_3$ norm and a Rademacher variable $a$ that is independent with $g$. Then we have*

$$\max\{\frac{1}{2}Var(g), \frac{1}{4}(\mathbb{E}[g])^2\} \leq Var(a\phi(ag)) \leq \max\{Var(g), \frac{1}{2}(\mathbb{E}[g])^2\}; \quad \text{(A.95)}$$

$$\mathbb{E}\left[\left|a\phi(ag) - \mathbb{E}[a\phi(ag))]\right|^3\right] \leq 32\max\{\mathbb{E}[|g - \mathbb{E}[g]|^3], |\mathbb{E}[g]|^3\}. \quad \text{(A.96)}$$

*Proof.* The expectation of the random variable $a\phi(ag)$ is

$$\mathbb{E}[a\phi(ag)] = \frac{1}{2}\mathbb{E}[\phi(g) - \phi(-g)] = \frac{1}{2}\mathbb{E}[g], \quad \text{(A.97)}$$

where the first equation uses the law of expectation, and the second equation uses $\phi(x) - \phi(-x) = x$. The second moment of $a\phi(ag)$ is

$$\mathbb{E}[(a\phi(ag))^2] = \mathbb{E}[\phi(ag)^2] = \frac{1}{2}\mathbb{E}[\phi(g)^2 + \phi(-g)^2] = \frac{1}{2}\mathbb{E}[g^2], \tag{A.98}$$

where the last equation uses $\phi(x)^2 + \phi(-x)^2 = x^2$. Combining (A.97) and (A.98), we have

$$\mathrm{Var}(a\phi(ag)) = \frac{1}{2}\mathbb{E}[g^2] - \frac{1}{4}(\mathbb{E}[g])^2 = \frac{1}{2}\mathrm{Var}(g) + \frac{1}{4}(\mathbb{E}[g])^2,$$

which implies (A.95). Moreover, for a random variable $X$ that has finite $L_3$ norm, we have

$$\|X - \mathbb{E}[X]\|_3 \le \|X\|_3 + \|\mathbb{E}[X]\|_3 \le \|X\|_3 + \mathbb{E}[|X|] \le 2\|X\|_3,$$

where the second inequality is due to $\|\mathbb{E}[X]\|_3 = |\mathbb{E}[X]|$ and the last inequality is due to $\|X\|_1 \le \|X\|_3$. Thus we have

$$\mathbb{E}\big[\big|a\phi(ag) - \frac{1}{2}\mathbb{E}[g]\big|^3\big] \le 8\mathbb{E}[|a\phi(ag)|^3] = 4\mathbb{E}[\phi(g)^3 + \phi(-g)^3] = 4\mathbb{E}[|g|^3],$$

where the last equation is due to $\phi(x)^3 + \phi(-x)^3 = |x|^3$. Then by $\|g\|_3 \le \|g - \mathbb{E}[g]\|_3 + |\mathbb{E}[g]|$, we have

$$\mathbb{E}\big[\big|a\phi(ag) - \frac{1}{2}\mathbb{E}[g]\big|^3\big] \le 4\big(\|g - \mathbb{E}[g]\|_3 + |\mathbb{E}[g]|\big)^3 \le 32\max\{\mathbb{E}[|g - \mathbb{E}[g]|^3], |\mathbb{E}[g]|^3\}.$$

$\square$

**Lemma A.15.** *Suppose $Z = [z_1, \cdots, z_n]^\top \sim N(0, \Sigma)$, where $\Sigma_{ii} = 1$, and $|\Sigma_{ij}| \le 1/(Cn^2), 1 \le i \ne j \le n$. And $Z' = [z'_1, \cdots, z'_n]^\top \sim N(0, \mathbb{I}_n)$. Let $b_i = \mathbb{I}(z_i > 0)$ and $b'_i = \mathbb{I}(z'_i > 0), i \in [n]$ be Bernoulli random variables. Let $B = [b_1, \cdots, b_n]^\top$ and $B' = [b'_1, \cdots, b'_n]^\top$. Then we have that for any non-negative function $f : \mathbb{R}^n \to \mathbb{R}^+ \cup \{0\}$,*

$$\frac{1}{2}\mathbb{E}_{B'}[f(b'_1, \cdots, b'_n)] \le \mathbb{E}_B[f(b_1, \cdots, b_n)] \le 2\mathbb{E}_{B'}[f(b'_1, \cdots, b'_n)].$$

*Proof.* Note that for any fixed value $(b_1, \cdots, b_n) \in \{0,1\}^n$, $\mathbb{P}_{B'}(b'_1, \cdots, b'_n) = (1/2)^n$. Then we have

$$\begin{aligned}
\mathbb{E}_B[f(b_1, \cdots, b_n)] &= \sum_{b_1, \cdots, b_n} f(b_1, \cdots, b_n)\mathbb{P}_B(b_1, \cdots, b_n) \\
&\ge (2\gamma_1)^n \sum_{b_1, \cdots, b_n} f(b_1, \cdots, b_n)\mathbb{P}_{B'}(b_1, \cdots, b_n) \\
&= (2\gamma_1)^n \mathbb{E}_{B'}[f(b_1, \cdots, b_n)],
\end{aligned} \tag{A.99}$$

where the inequality comes from Lemma A.16. On the other side, similarly we have

$$\mathbb{E}_B[f(b_1, \cdots, b_n)] \le (2\gamma_2)^n \mathbb{E}_{B'}[f(b_1, \cdots, b_n)]. \tag{A.100}$$

By $C > 8$, we have $(2\gamma_1)^n = (1 - 4/(Cn))^n \ge 1 - 4/(Cn) \ge 1/2$ and $(2\gamma_2)^n = (1 + 4/(Cn))^n \le \exp(4/C) \le \exp(1/2) \le 2$. Combining these results with (A.99) and (A.100), we have

$$\frac{1}{2}\mathbb{E}_{B'}[f(b'_1, \cdots, b'_n)] \le \mathbb{E}_B[f(b_1, \cdots, b_n)] \le 2\mathbb{E}_{B'}[f(b'_1, \cdots, b'_n)].$$

$\square$

**Lemma A.16.** *Suppose $Z = [z_1, \cdots, z_n]^\top \sim N(0, \Sigma)$, where $\Sigma_{ii} = 1$, and $|\Sigma_{ij}| \le 1/(Cn^2), 1 \le i \ne j \le n$. Then we have that for any subset $\mathcal{A} \subseteq [n]$,*

$$\gamma_1^n \le \mathbb{E}[\prod_{i \in \mathcal{A}} \mathbb{I}(z_i > 0) \cdot \prod_{i \in [n]\setminus\mathcal{A}} \mathbb{I}(z_i < 0)] \le \gamma_2^n$$

*for $\gamma_1 = 1/2 - 2/(Cn)$ and $\gamma_2 = 1/2 + 2/(Cn)$.*

*Proof.* We first prove the result for $\mathcal{A} = [n]$. Note that

$$\mathbb{P}(z_1 > 0, \cdots, z_n > 0) = \mathbb{P}(z_1 > 0) \prod_{k=2}^{n} \mathbb{P}(z_k > 0 | z_{k-1} > 0, \cdots, z_1 > 0). \qquad \text{(A.101)}$$

Let $Z_{k-1} = [z_1, \cdots, z_{k-1}]^\top$ and denote the covariance matrix of $[z_1, \cdots, z_k]$ as

$$\begin{bmatrix} \Sigma_{k-1} & \epsilon_k \\ \epsilon_k^\top & 1 \end{bmatrix},$$

where $\Sigma_{k-1} = \mathrm{Cov}(Z_{k-1})$ and $\epsilon_k = \mathrm{Cov}(Z_{k-1}, z_k)$. Then $|\epsilon_{kj}| \leq 1/(Cn^2)$ for $j \in [k-1]$, and the conditional distribution of $z_k | Z_{k-1}$ is $N(\epsilon_k^\top \Sigma_{k-1}^{-1} Z_{k-1}, 1 - \epsilon_k^\top \Sigma_{k-1}^{-1} \epsilon_k)$. By Gershgorin circle theorem, we have

$$1 - \frac{1}{Cn} \leq \lambda_{\min}(\Sigma_{k-1}) \leq \lambda_{\max}(\Sigma_{k-1}) \leq 1 + \frac{1}{Cn}.$$

Denote $f_{k-1}(\cdot)$ as the density function of $Z_{k-1}$. Then we have

$$\mathbb{P}(z_k > 0 | z_{k-1} > 0, \cdots, z_1 > 0) = \int_0^\infty \cdots \int_0^\infty f_{k-1}(Z_{k-1}) \bar{\Phi}\left( \frac{-\epsilon_k^\top \Sigma_{k-1}^{-1} Z_{k-1}}{\sqrt{1 - \epsilon_k^\top \Sigma_{k-1}^{-1} \epsilon_k}} \right) dz_1 \cdots dz_{k-1}$$

$$\geq \int_{\|\Sigma_{k-1}^{-1/2} Z_{k-1}\| \leq 2\sqrt{n}} f_{k-1}(Z_{k-1}) \bar{\Phi}\left( \frac{-\epsilon_k \Sigma_{k-1}^{-1} Z_{k-1}}{\sqrt{1 - \epsilon_k^\top \Sigma_{k-1}^{-1} \epsilon_k}} \right) dz_1 \cdots dz_{k-1}$$

$$\geq \left( \frac{1}{2} - \frac{\|\Sigma_{k-1}^{-1/2} \epsilon_k\| \cdot 2\sqrt{n}}{\sqrt{2\pi(1 - \epsilon_k^\top \Sigma_{k-1}^{-1} \epsilon_k)}} \right) \mathbb{P}(\|\Sigma_{k-1}^{-1/2} Z_{k-1}\| \leq 2\sqrt{n})$$

$$\geq \left( \frac{1}{2} - \frac{2\sqrt{2}}{nC\sqrt{\pi}} \right) \mathbb{P}(\|\Sigma_{k-1}^{-1/2} Z_{k-1}\| \leq 2\sqrt{n})$$

$$\geq \left( \frac{1}{2} - \frac{2\sqrt{2}}{nC\sqrt{\pi}} \right)(1 - \exp(-n)) \geq \frac{1}{2} - \frac{2}{Cn}$$

$$\text{(A.102)}$$

for sufficiently large $n$. Here the second inequality uses $|\Phi(x) - \Phi(0)| \leq \Phi'(0)|x|$ and Cauchy-Schwarz inequality; the third inequality uses $\sigma_{\min}(\Sigma_{k-1}) = \lambda_{\min}(\Sigma_{k-1}) \geq 1/2$ and $\|\Sigma_{k-1}^{-1/2} \epsilon_k\| \leq \sqrt{2}\|\epsilon_k\| \leq \sqrt{2}n^{-3/2}/C$; and the fourth inequality uses the concentration inequality for chi-square random variables in Lemma A.17. Then the result is proved by combining (A.101) and (A.102). On the other side, we have

$$\mathbb{P}(z_k > 0 | z_{k-1} > 0, \cdots, z_1 > 0) \leq \int_{\|\Sigma_{k-1}^{-1/2} Z_{k-1}\| \leq 2\sqrt{n}} f_{k-1}(Z_{k-1}) \bar{\Phi}\left( \frac{-\epsilon_k \Sigma_{k-1}^{-1} Z_{k-1}}{\sqrt{1 - \epsilon_k^\top \Sigma_{k-1}^{-1} \epsilon_k}} \right) dz_1 \cdots dz_{k-1}$$

$$+ \mathbb{P}(\|\Sigma_{k-1}^{-1/2} Z_{k-1}\| > 2\sqrt{n})$$

$$\leq \left( \frac{1}{2} + \frac{\|\Sigma_{k-1}^{-1/2} \epsilon_k\| \cdot 2\sqrt{n}}{\sqrt{2\pi(1 - \epsilon_k^\top \Sigma_{k-1}^{-1} \epsilon_k)}} \right) + \mathbb{P}(\|\Sigma_{k-1}^{-1/2} Z_{k-1}\| > 2\sqrt{n})$$

$$\leq \frac{1}{2} + \frac{2\sqrt{2}}{nC\sqrt{\pi}} + \exp(-n) \leq \frac{1}{2} + \frac{2}{Cn}.$$

Note that our proof does not use any information related to $\mathcal{A}$, thus we can extend the result for any subset $\mathcal{A} \subseteq [n]$. $\qquad \square$

**Lemma A.17.** *For $X_k$ i.i.d $\sim N(0, \sigma^2), 1 \leq k \leq n$, we have*

$$\Phi'(t)/t \leq \mathbb{P}(|X_1| \geq t\sigma) \leq \exp(-t^2/2), \quad \forall t \geq 1;$$

$$\mathbb{P}(|\frac{1}{n\sigma^2}\sum_{k=1}^{n}X_k^2 - 1| \geq t) \leq 2\exp(-nt^2/8), \quad \forall t \in (0,1).$$

*Proof.* For the first inequality, we note that

$$\bar{\Phi}(t) = \int_t^{+\infty}\frac{x}{\sqrt{2\pi}x}\exp(-\frac{1}{2}x^2)dx \leq \int_t^{+\infty}\frac{1}{2\sqrt{2\pi}t}\exp(-\frac{1}{2}x^2)dx^2 = \frac{\Phi'(t)}{t}.$$

It yields that for any $t \geq 1$,

$$\mathbb{P}(|X_1| \geq t\sigma) = 2\bar{\Phi}(t) \leq 2\Phi'(t)/t \leq \exp(-t^2/2).$$

On the other side, we have

$$\bar{\Phi}(t) \geq \int_t^{+\infty}\frac{\frac{1+x^2}{x^2}}{\sqrt{2\pi}\frac{1+t^2}{t^2}}\exp(-\frac{1}{2}x^2)dx = \frac{1}{\sqrt{2\pi}}\frac{t^2}{1+t^2}\Big(-\frac{\exp(-\frac{x^2}{2})}{x}\Big)\Big|_{x=t}^{+\infty} = \frac{t}{1+t^2}\Phi'(t).$$

When $t \geq 1$, it further yields that $\bar{\Phi}(t) \geq \Phi'(t)/(2t)$. Thus we have

$$\mathbb{P}(|X_1| \geq t\sigma) = 2\bar{\Phi}(t) \geq \Phi'(t)/t.$$

The second inequality is Example 2.11 in Wainwright (2019) $\qquad\qquad\square$

**Lemma A.18** (Hoeffding's inequality, Equation (2.11) in Wainwright (2019))**.** *Let $X_k, 1 \leq k \leq n$ be a series of independent random variables with $X_k \in [a,b]$. Then*

$$\mathbb{P}(\sum_{k=1}^{n}(X_k - \mathbb{E}[X_k]) \geq t) \leq \exp\Big(-\frac{2t^2}{n(b-a)^2}\Big), \quad \forall t \geq 0.$$

**Lemma A.19.** *[Berry-Esseen Theorem, Theorem 3.4.17 in Durrett (2019)] Let $X_1, \cdots, X_n$ are i.i.d. random variables with $\mathbb{E}[X_i] = 0, Var(X_i) = \sigma^2$, and $\mathbb{E}[|X_i|^3] = \rho < \infty$. If $F_n(x)$ is the distribution of $\sum_{i=1}^{n}X_i/(\sigma\sqrt{n})$, then*

$$|F_n(x) - \Phi(x)| \leq \frac{3\rho}{\sigma^3\sqrt{n}}.$$

## A.7 EXPERIMENTAL DETAILS

In our experiments, dimension $p = 40000$, number of train/test samples $n = 200$ $\mu = 2.5\sqrt{p/n}$, number of neurons $m = 1000$, label noise rate $\eta = 0.05$, and initial weight scale $\omega_{\text{init}} = 10^{-15}$. For Figure 3, 2, and 1-left, the step size $\alpha = 10^{-12}$. For Figure 4 and 1-right, $\alpha = 10^{-16}$.

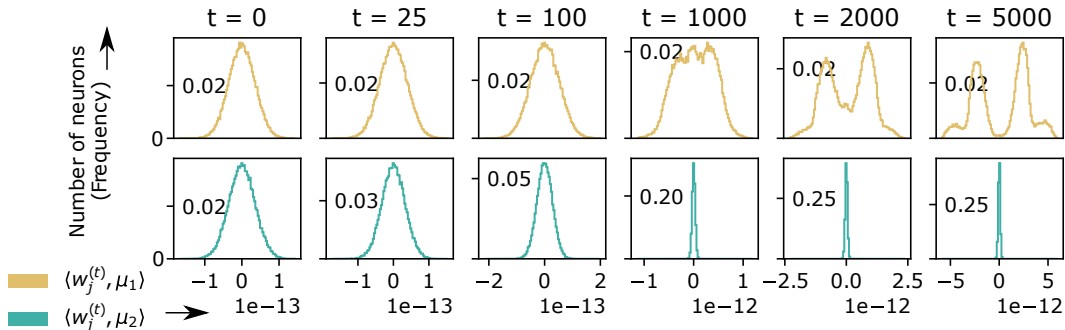

Figure 4: Histograms of inner products between positive neurons and $\mu$'s pooled over 100 independent runs under the same setting as in Figure 1 but with a smaller step size. *Top (resp. bottom) row:* Inner products between positive neurons and $\mu_1$ (resp. $\mu_2$). While the projections of positive neurons $w_j^{(t)}$ onto the $\mu_1$ and $\mu_2$ directions have nearly the same distribution when the network cannot generalize, they become much more aligned with $\pm\mu_1$ when the network can generalize.

### A.8 ADDITIONAL EXPERIMENTS

In this section, we show additional experiments with setups that are variations on that of Figure 1. See Appendix A.7 for details of the setup of the experiments in Figure 1.

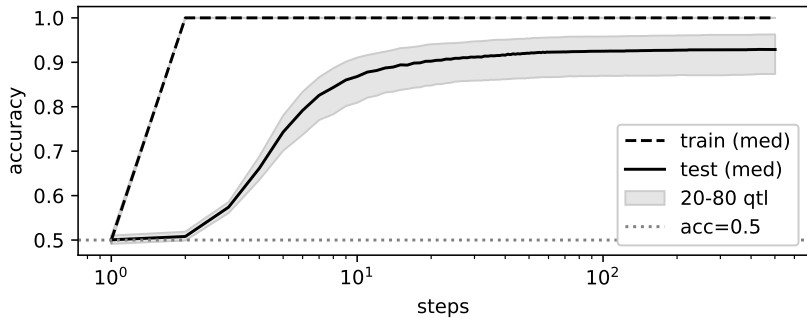

Figure 5: **More samples hurt generalization**. Same experimental setup as in Figure 1 (Left) except that the sample size $n = 2000$ is 10 times larger. Even running over $10\times$ number of steps, the median test accuracy remains near $90\%$, while in Figure 1 the median test accuracy is $100\%$.

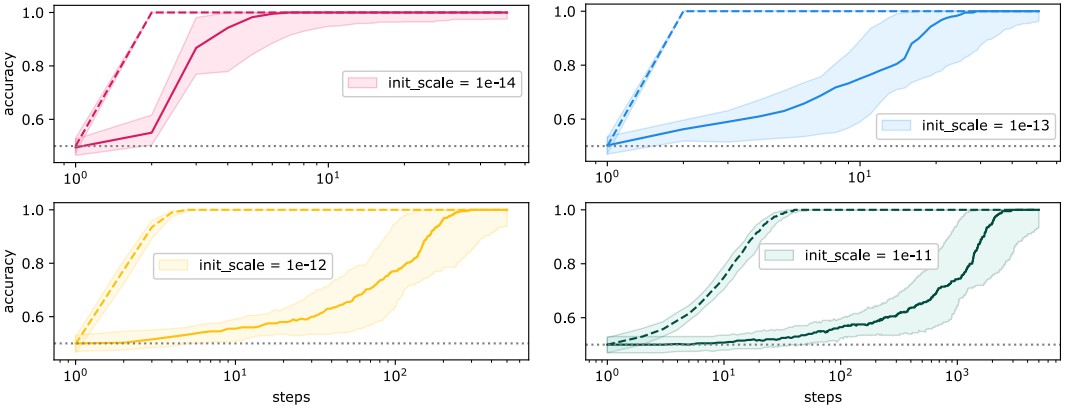

Figure 6: **Larger initial weight scale**. Same experimental setup as in Figure 1 except that the initial weight scale $\omega_{\text{init}}$ are larger. (Recall that $\omega_{\text{init}} = 10^{-15}$ in Figure 1). Moreover, the experiments corresponding to $\omega_{\text{init}} = 10^{-12}$ (resp. $\omega_{\text{init}} = 10^{-11}$) are ran with $10\times$ (resp. $100\times$) number of steps.

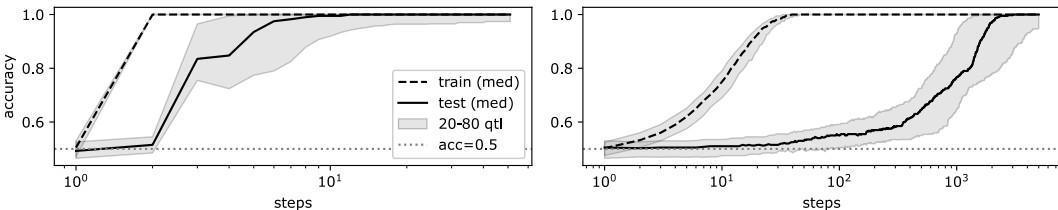

Figure 7: **Train both layers**. Same experimental setup as in Figure 1 except that the outer layer weights are also trained. The accuracy dynamics is similar to that of Figure 1.

