# OpenReview forum: "Benign Overfitting and Grokking in ReLU Networks for XOR Cluster Data"
_ICLR.cc/2024/Conference — ICLR 2024 poster_

### Official Review · Reviewer_AQUA · 2023-10-31

**Soundness:** 3 good
**Presentation:** 3 good
**Contribution:** 3 good
**Rating:** 5
**Confidence:** 4

**Summary:**

This paper theoretically combines the phenomena of benign overfitting and Grokking by training a two-layer ReLU neural network using gradient descent on XOR cluster data. The authors demonstrate that after the first step of gradient descent, the network achieves 100% training accuracy, perfectly fitting the noisy labels in the training data, but exhibits nearly random performance on the test data. However, after some time in training, the Grokking phenomenon occurs, and the network achieves near-optimal test accuracy while still adapting to the random labels in the training data, demonstrating benign overfitting.

**Strengths:**

This paper is the first to study the combination of benign overfitting and the Grokking phenomenon in neural networks.

**Weaknesses:**

My main concern about this paper lies in its assumptions. Combining assumptions A1 and A2, we can obtain $p\geq C^4 n^{5.02}$. This is an extremely high-dimensional setting.

**Questions:**

1: Do the authors have any ideas for improvements in the high-dimensional setting?

2: What behavior does the test error exhibit when the training time is between 1 and $n^{0.01}$?

---

> ### Author Response · Authors · 2023-11-18
> **Author Response to Reviewer AQUA**
>
> We thank the reviewer for their review and address their comments and questions below. We hope that the reviewer will consider raising their score in light of our response.
>
> **1: My main concern about this paper lies in its assumptions… Do the authors have any ideas for improvements in the high-dimensional setting?**
>
> Note that all existing benign overfitting results require input dimension much larger than the number of samples ($p>\mathrm{poly}(n)$), e.g. [[Bartlett et al. (2019)](https://www.pnas.org/doi/10.1073/pnas.1907378117), [Frei et al. (2022)](https://arxiv.org/abs/2202.05928), [Cao et al. (2022)](https://arxiv.org/abs/2202.06526), [Frei et al. (2023)](https://jmlr.org/beta/papers/v24/22-1132.html), [Xu & Gu (2023)](https://proceedings.mlr.press/v206/xu23k.html), [Kou et al. (2023)](https://arxiv.org/abs/2303.04145)]. Compared with these papers, our non-linearly separable setting is significantly harder. One major technical challenge is to give concentration bounds for summation of dependent random variables (see Equations (A.17)-(A.22) and Lemma A.17 in Section A.6). There are few techniques that can analyze the concentration of such summation, which is necessary for our trajectory analysis. We note that such difficulty is due to the non-linearly separable nature of our data distribution, which was not present in previous papers on linearly separable data.
>
> Furthermore, we are not sure what the reviewer means regarding potential ideas for “improvements in the high-dimensional setting.” We do not have a complete understanding of whether grokking or benign overfitting happen in lower-dimensional settings, and we believe a full characterization of whether these two phenomena occur as a function of the dimension, number of samples, step-size, initialization variance, and signal-to-noise ratio (determined by $\|\mu_i\|$) is an extremely challenging problem. Indeed, prior work by [Frei et al. (2023)](https://jmlr.org/beta/papers/v24/22-1132.html) (Figure 2, page 15) showed that in the high-SNR, low-dimensional setting for two-layer ReLU nets trained on the XOR cluster distribution we consider, overfitting is not benign and grokking does not occur.
>
> We believe it is significant that we have established a proof of both benign overfitting and grokking in ReLU networks for nonlinear data, and that our work helps elucidate how high-dimensionality and low-SNR can play a crucial role in both of these phenomena. We believe these contributions are of substantial interest to the ICLR community. We hope the reviewer will consider raising their score.
>
> **2: What behavior does the test error exhibit when the training time is between 1 and $n^{0.01}$?**
>
> Regarding the test error behavior between time $1$ and time $n^{0.01}$: for technical reasons, we cannot prove a generalization bound for the behavior between times 1 and $n^{0.01}$. That said, we are able to prove that the expectation of the margin keeps increasing with step $t$, which intuitively should help generalization. Empirically, Figure 1 shows that the test accuracy increases gradually from near-random to near-perfect.

---

> > ### Comment · Reviewer_AQUA · 2023-11-20
> >
> > Dear Authors,
> >
> > Thank you for the response.
> >
> > I apologize for the ambiguity in my question. The original intent of Question 1 was to inquire whether the high-dimensional condition in this paper, $p\geq n^5$, could be improved to something similar to the conditions in the previous papers, such as $p\geq n^2 \log n$ in [1].
> >
> > Thank you,
> >
> > Reviewer AQUA
> >
> > Reference:
> >
> > [1] Spencer Frei, et al. Benign Overfitting without Linearity: Neural Network Classifiers Trained by Gradient Descent for Noisy Linear Data. COLT 2022.

---

> > > ### Author Response · Authors · 2023-11-21
> > > **Author Response to Reviewer AQUA**
> > >
> > > Thank you for your question. It is important to note that previous studies on benign overfitting, including the one you referred to, only focused on linearly separable data. Our data distribution, however, is not linearly separable, which leads to qualitatively different behavior and a more complex trajectory analysis. Specifically, a new technical challenge in our analysis is providing concentration bounds for sum of dependent random variables that are not weakly dependent (i.e. $\text{Cor}(X_t, X_s) \rightarrow 0$ as |t-s| goes to infinity), as detailed in Equations (A.17)-(A.22) and Lemma A.17 in Section A.6. There are few probabilistic tools that can handle such a summation, and in order to handle this concentration we need the stronger condition $p>n^5$. The proof of this concentration inequality is the only place that requires $p > n^5$, and $p>n^2 \log(n)$ suffices for the remaining  proofs.
> > >
> > > Our paper focuses on theoretically capturing the qualitative behavior of (non-linear) benign overfitting and grokking (which we believe is a significant contribution, as Reviewers 6CDP and mTpn both noted in their reviews), instead of optimizing all the technical assumptions. We leave the relaxation of this assumption for future work.

---

> > > > ### Comment · Reviewer_AQUA · 2023-11-22
> > > >
> > > > Dear Authors,
> > > >
> > > > Thank you for the response.
> > > >
> > > > I still have a doubt about linearly separable. Can the author explain why you think the setting [1] is linearly separable? Thank you!
> > > >
> > > > Reviewer AQUA
> > > >
> > > > Reference:
> > > >
> > > > [1] Spencer Frei, et al. Benign Overfitting without Linearity: Neural Network Classifiers Trained by Gradient Descent for Noisy Linear Data. COLT 2022.

---

> > > > > ### Author Response · Authors · 2023-11-22
> > > > > **Author Response to Reviewer AQUA**
> > > > >
> > > > > Thank you for your question. In that paper, they consider a setting where $x = y\mu + z$, with $y = \pm 1$ representing the label and $z$ being the noise term.  This results in two distinct clusters, each with opposite labels: one centered around $+\mu$ and the other around $-\mu$.  We call this a linearly separable distribution because the optimal classifier $f(x)$ for this setting  is a linear one, i.e. $f(x) = \text{sgn}(\langle \mu, x\rangle)$.
> > > > >
> > > > > In our setting, clusters centered at $+\mu_1$ and $-\mu_1$ are both assigned the label $+1$, and clusters centered at $+\mu_2$ and $-\mu_2$ are both assigned the label $-1$. Since opposing clusters share the same sign, linear classifiers achieve 50% test error for the distribution we consider, so the distribution we consider is one which is not linearly separable.

---

> > > > > > ### Comment · Reviewer_AQUA · 2023-11-22
> > > > > >
> > > > > > Dear Authors,
> > > > > >
> > > > > > Thank you for the response. I will increase the score. And I recommended the author add some relevant remarks to the paper.
> > > > > >
> > > > > > Reviewer AQUA

---

### Official Review · Reviewer_mTpn · 2023-11-01

**Soundness:** 3 good
**Presentation:** 3 good
**Contribution:** 3 good
**Rating:** 6
**Confidence:** 4

**Summary:**

This paper delves into the exploration of benign overfitting and "grokking" in two-layer ReLU neural networks when trained on XOR cluster data with noisy labels. The authors demonstrate that networks can perfectly fit noisy training data (benign overfitting) and, after a period, transition from harmful overfitting to a stage where they generalize near-optimally ("grokking"). Through rigorous theoretical analysis and proofs, the study reveals that these surprising phenomena are evident in the networks' training trajectories, providing a nuanced understanding of overfitting and generalization in neural network models.

The paper's contributions lie in providing the first theoretical insights into benign overfitting in non-linearly separable data distributions, unraveling the feature learning dynamics under gradient descent. These findings illuminate the pathways through which neural networks navigate the complexities of noisy data, offering a fresh perspective on their capacity to generalize despite apparent overfitting.

**Strengths:**

* The paper offers a new theoretical examination of benign overfitting and "grokking" in two-layer ReLU neural networks. It focuses on XOR cluster data with noisy labels, giving a detailed exploration and proofs related to these phenomena. The authors use existing concepts and new theories to better explain the behavior of neural networks with noisy training data.

* The paper is structured and clear, effectively communicating the authors’ work and results. It has a logical organization that makes it easy for readers to follow the ideas and analyses. The presentation of definitions, explanations, and proofs is mostly straightforward, helping readers understand the complex concepts and findings.

* The paper is important because it helps understand overfitting and generalization in neural networks better. It explains the concepts of benign overfitting and "grokking" in one framework.

**Weaknesses:**

* The assumption made in A1 seems to contradict common understanding. Generally, increasing the number of samples, even with limited noisy labels, tends to enhance the generalization capability of neural networks. However, in Assumption A1, having a larger number of training samples seems to adversely affect the model, as indicated by its presence on the right-hand side of the inequality. This aspect might require further clarification or justification within the context of the study.

* The mechanism of overfitting, once the neural network learns the directions of $u_1$ and $u_2$. is not explicitly clear. The paper mentions that post the initial gradient step, positive neurons learn $u_1$ while negative neurons learn $u_2$. However, the explanation seems lacking in how the network overfits to samples with noisy (flipped) labels in this condition. A more detailed discussion or clarification on this would be beneficial.

* The role of Lemma 4.6 in the paper is unclear. A clearer explanation of how it relates to other parts of the paper and how it contributes to the overall arguments and conclusions is needed for better understanding.

* The paper seems to lack a comparative discussion with some relevant works, specifically references [1,2].


[1] Meng, Xuran, Difan Zou, and Yuan Cao. "Benign Overfitting in Two-Layer ReLU Convolutional Neural Networks for XOR Data." arXiv preprint arXiv:2310.01975 (2023).

[2] Glasgow, Margalit. "SGD Finds then Tunes Features in Two-Layer Neural Networks with near-Optimal Sample Complexity: A Case Study in the XOR problem." arXiv preprint arXiv:2309.15111 (2023).

**Questions:**

* Could you provide more insights or justification regarding Assumption A1? Specifically, could you clarify why an increase in the number of training samples seems to negatively influence the model, contrary to the common understanding that more samples generally improve a model's generalization capability?

* Could you elaborate on the mechanism of overfitting after the neural network learns the directions of $u_1$ and $u_2$.

* Could you clarify the role and significance of Lemma 4.6 in the context of the paper's objectives and findings?

* Could you make a comparsion your results and techniques with [1] and [2]?

* Have you considered variations in the network architecture, such as not fixing the second layer, and if so, how do these variations influence the results?

* In the model settings used in the paper, there doesn’t appear to be an upper bound on the network width. If an extremely wide network setting were used, would the findings align with the "lazy training" regime? Could you discuss how the results might be influenced by varying the width of the network to such extremes?

I would increase my score if the authors could clarify my concerns demonstrated in the above questions.

--------------------------------------
I increase my score to 6 after rebuttal.

---

> ### Author Response · Authors · 2023-11-18
> **Author Response to Reviewer mTpn (part I)**
>
> We thank the reviewer the detailed review and for appreciating the contributions of our paper. It is encouraging to see that the reviewer thinks our paper is “important” and “structured and clear” and offers “a new theoretical examination of benign overfitting and grokking.”
>
> We are happy to address the reviewer’s questions below, and hope that the reviewer will consider raising the score in light of our response.
>
> **Could you provide more insights or justification regarding Assumption A1? Specifically, could you clarify why an increase in the number of training samples seems to negatively influence the model, contrary to the common understanding that more samples generally improve a model's generalization capability?**
>
> We appreciate the question regarding the potential negative impact of increasing the number of samples on the model's performance, and agree that it is a bit counterintuitive. We wish to mention that an assumption on the upper bound of $n$ was needed in all previous benign overfitting results (since they all require high-dimensional data), e.g. [[Bartlett et al. (2019)](https://www.pnas.org/doi/10.1073/pnas.1907378117), [Frei et al. (2022)](https://arxiv.org/abs/2202.05928), [Cao et al. (2022)](https://arxiv.org/abs/2202.06526), [Frei et al. (2023)](https://jmlr.org/beta/papers/v24/22-1132.html), [Xu & Gu (2023)](https://proceedings.mlr.press/v206/xu23k.html), [Kou et al. (2023)](https://arxiv.org/abs/2303.04145)]. Also note that in practical settings it is possible that increasing the sample size could hurt generalization (see e.g. “deep double descent” [[Nakkiran et al. (2019)](https://arxiv.org/abs/1912.02292)]).
>
> Empirically, such an upper bound for sample size is indeed necessary. We have added an experiment for large sample size $n$ (see Figure 5 in Appendix A.8), which empirically shows that increasing $n$ could harm generalization, so it is not just a limitation of the analysis.
>
> Additionally, we also refer the reviewer to Figure 2 in [Frei et al. (2023)](https://jmlr.org/beta/papers/v24/22-1132.html) (page 15), where they showed empirically that an upper bound for the number of samples is required (see the discussion section there).
>
> **Could you elaborate on the mechanism of overfitting after the neural network learns the directions of u1 and u2?**
>
> A high-level intuition of this mechanism is: the training data points can be easily memorized by the neural network due to the near-orthogonality property ($\| x_i \|^2 \gg |\langle x_i, x_j \rangle|, \forall j \neq i$) and the high-dimensionality ($p\gg n$). The network has the capacity to learn the correct directions $\mu_1$ and $\mu_2$ and to memorize all the training data points.
>
> In our paper, we prove that once the model memorizes a data point, it will never forget it. Specifically, for $a_j y_i > 0$, if at step $s$, the neuron $w_j$ positively correlates with the data point $x_i$, i.e. $\langle w_j^{(s)}, x_i\rangle > 0$, then it will stay that way afterward since $\langle w_j^{(t+1)} – w_j^{(t)}, x_i \rangle > 0, \forall t \ge s$. We refer the reviewer to (F1)-(F2) in Corollary A.5, which are the key properties used to prove the overfitting result. We note that previous benign overfitting results [[Cao et al. (2022)](https://arxiv.org/abs/2202.06526), [Xu & Gu (2023)](https://proceedings.mlr.press/v206/xu23k.html)] also utilized analogous properties to prove the overfitting result.
>
> We think a more interesting question is how features can continue to be learned after the network has already achieved a perfect fit to data - this question seems to be the crux of why grokking has received so much attention. What we show is that even though the network has fitted the data, the small signal in the data continues to be picked up by gradient descent.  In high dimensions, the noise components are close to completely orthogonal, so over time the small increments in the signal learned by each step of gradient descent grow more than the fixed level of noise coming from the nearly-orthogonal noise variables.
>
> **Could you clarify the role and significance of Lemma 4.6 in the context of the paper's objectives and findings?**
>
> The role of this technical lemma is solely to explain the approximation in Equation (4.3). We will make this point clear in the updated version of the paper.
>
> To provide some further explanation, recall that (4.3) is the key equation which shows that the network’s decision boundary at step 1 is approximately linear and thus it fails to generalize. The key step in (4.3) (with the “a.s.”) follows from Lemma 4.6. In the formal proof of the non-generalization result, we use concentration properties that have a similar form to Lemma 4.6 (see Lemma A.12 in Section A.5.2, which is used to prove Theorem 4.8).

---

> > ### Author Response · Authors · 2023-11-18
> > **Author Response to Reviewer mTpn (part II)**
> >
> > **Could you make a comparison of your results and techniques with [1] and [2]?**
> >
> > Thank you for bringing up the related papers. We note that these papers appeared on arXiv *after* the ICLR submission deadline. That said, we are happy to make a comparison with these papers:
> >
> > [1] uses a simplified convolutional neural network on the XOR problem, and proves benign overfitting with a similar generalization upper bound as ours. Compared with [1], a key difference in our paper is that our setting also exhibits the delayed generalization/grokking phenomenon, which was not observed in [1] or any other benign overfitting papers before. Our results give a full characterization of this grokking phenomenon, offering the first rigorous proof of grokking in a neural network setting.
> >
> > [2] uses a two-layer neural network on the (p, 2) parity task and gives a generalization bound for that problem. It’s worth noting that their XOR data distribution is different from ours: their noise terms are Boolean $\pm 1$, while our noise terms are Gaussian. As they mentioned in the Conclusion section, their results only hold for Boolean data and cannot be extended to Gaussian noise XOR data. [2] belongs to another line of work studying (p, 2) parity [[Ji & Telgarsky (2019)](https://arxiv.org/abs/1909.12292); [Wei et al. (2019)](https://arxiv.org/abs/1810.05369); [Barak et al. (2022)](https://arxiv.org/abs/2207.08799); [Telgarsky (2023)](https://openreview.net/forum?id=swEskiem99); [Suzuki et al. (2023)](https://openreview.net/forum?id=tj86aGVNb3&referrer=%5BAuthor%20Console%5D(%2Fgroup%3Fid%3DNeurIPS.cc%2F2023%2FConference%2FAuthors%23your-submissions)] and is not directly comparable with the Gaussian noise XOR problem we study.
> >
> > **Have you considered variations in the network architecture, such as not fixing the second layer, and if so, how do these variations influence the results?**
> >
> > We appreciate the reviewer’s question. We have added an experiment that uses gradient descent to train both layers with the same initialization as our existing experiments. The results (Figure 7 in Appendix A.8) show that the learning dynamics is similar to that with the second layer fixed and that the benign overfitting and grokking phenomena are still observed.
> >
> > Theoretically, our developed framework cannot be directly extended to gradient descent on both layers, as some concentration results for random variables with complicated dependency structure need to be obtained. The techniques used in [2] also cannot be directly extended to our setting. As [2] mentioned in their Conclusion Section, for Boolean data, it is easier to compute $\nabla_a L$ and $\nabla_W L$, and the dynamics of the gradients do not depend on interactions within neurons. But for Gaussian noise XOR, the dynamics can be quite different and the neurons corresponding to the signal may go in the wrong directions. It is definitely worth exploring the learning dynamics of neural networks trained by gradient descent on both layers for future research.
> >
> >
> > **In the model settings used in the paper, there doesn’t appear to be an upper bound on the network width. If an extremely wide network setting were used, would the findings align with the "lazy training" regime? Could you discuss how the results might be influenced by varying the width of the network to such extremes?**
> >
> > Yes, our result holds for any width $m \geq C n^{0.01}$ and does not require an upper bound on $m$. The same results hold even if $m$ is extremely large.
> >
> > We emphasize that the setting we consider is different from the “lazy training” / “neural tangent kernel (NTK)” regime, even if the width goes to infinity. This is because we consider a smaller initialization scale than that in the NTK regime. One way to see this difference is that our weight matrix W goes far from its initialization after one step of gradient descent (since step size $\alpha$ is much larger than the initialization scale), while in the NTK regime the weight matrix will always stay close to its initialization. Therefore, even with infinite width, the setting we consider is not in the lazy training/NTK regime.
> >
> >
> > ** **

---

> > > ### Comment · Reviewer_mTpn · 2023-11-22
> > >
> > > I extend my gratitude to the authors for their comprehensive response. Consequently, I am inclined to revise my rating upwards.

---

> > > > ### Author Response · Authors · 2023-11-22
> > > >
> > > > Thank you for your support in our paper!

---

### Official Review · Reviewer_6CDP · 2023-11-04

**Soundness:** 4 excellent
**Presentation:** 4 excellent
**Contribution:** 4 excellent
**Rating:** 6
**Confidence:** 4

**Summary:**

This paper analyzes two-layer ReLU network trained by gradient descent on XOR cluster data with a high-dimensional input space, and rigorously proves grokking and benign overfitting occurs.

Specifically, with one sufficiently large gradient step, the network is almost a linear classifier and achieves perfect overfitting to the training data, which contains label-flipping noise. If training is continued, the model almost perfectly predicts the clean label, while keeping perfect overfit to the training data.

**Strengths:**

### Notable results on benign overfitting of neural networks beyond linearly separable data

As far as I understand, proving benign overfitting for neural network involves several difficulties due to its nonlinearity, and especially I agree that showing the superiority of neural network to linear methods by learning nonlinear target function has been largely open in this context. I think XOR cluster data is a good starting point to this problem and this paper proves benign overfitting under moderate assumptions.

### Providing useful theoretical understandings on grokking

As well as benign overfitting result, this paper also proves that after one large (compared to the initialization scale) gradient step, the neural network approximately behaves as the linear model and perfectly overfit to the training data. The phenomena that the early stage of training only produces the linear model but there is a transition to the nonlinear neural network with richer features is particularly interesting, providing one of the first rigorous theoretical demonstrations of grokking.

### The paper is well written and clearly explaining its theoretical key points.

The proof sketch section is well-written and provides a sufficient understanding of the overall theoretical contributions. It is expected that the techniques presented in this paper will also be valuable in demonstrating the potential for more enriching feature learning in the future.

**Weaknesses:**

### Justification of the small initialization

In my understanding, it is crucial to take a small initialization scale compared to the step size $\alpha$ to obtain the perfect overfitting at the first gradient step. I think this is acceptable as theory, but it should be better to justify such a small initialization. I also want to know what happens if the initialization scale is much larger than used in Figure 3.

### Large signal-to-noise ratio

Compared to [Ji & Telgarsky (2019)](https://arxiv.org/abs/1909.12292); [Wei et al. (2019)](https://arxiv.org/abs/1810.05369); [Barak et al. (2022)](https://arxiv.org/abs/2207.08799); [Telgarsky (2023)](https://openreview.net/forum?id=swEskiem99); [Suzuki et al. (2023)](https://openreview.net/forum?id=tj86aGVNb3&referrer=%5BAuthor%20Console%5D(%2Fgroup%3Fid%3DNeurIPS.cc%2F2023%2FConference%2FAuthors%23your-submissions), where $x$ is a $d$-dimensional (essentially) rotationally invariant input and $y=\rm{sgn}(x_1x_2)$, this paper considers large signal $\|\mu\|$ as an input.

**Questions:**

- Is it possible to see an additional experiment when the initialization scale is not so small?

- When $\|\mu\|$ gets small, is this grokking phenomena still observed?

---

> ### Author Response · Authors · 2023-11-18
> **Author Response to Reviewer 6CDP**
>
> We thank the reviewer for the detailed review and for appreciating the contributions of our paper. It is encouraging to see that the reviewer thinks our paper is well-written and regards our paper as providing “notable results on benign overfitting” and “useful theoretical understanding on grokking.”
>
> We are happy to address the reviewer’s questions below.
>
>
>
> **Justification of the small initialization:**
>
> The small initialization is indeed crucial as it makes the analysis of the training dynamics easier. Such an assumption has been made in a number of related theory papers, such as [[Frei et al. (2022)](https://arxiv.org/abs/2202.05928), [Frei et al. (2023)](https://jmlr.org/beta/papers/v24/22-1132.html), [Xu & Gu (2023)](https://proceedings.mlr.press/v206/xu23k.html)].
>
> We share the reviewer’s desire for understanding what happens under a larger initialization scale and have done some experiments in this regime. Specifically, in our paper, Figure 1 (right) gives the train and test accuracy under large initialization (relative to step size). The large initialization version of Figure 3 is given in Figure 4 in Appendix A.7. We observe similar benign overfitting and grokking behaviors for large initialization, though the overfitting and generalization require more steps than the case of small initialization.
>
> We have also run additional experiments with a fixed learning rate and varying initialization scales; see Figure 6 in Appendix A.8. We again observe similar qualitative phenomena.
>
>
>
> **Large signal-to-noise (SNR) ratio:**
>
> We agree that our SNR is larger than that in the papers that the reviewer mentioned. However, those papers consider a different problem and are not directly comparable to our work. In particular, the XOR problem they studied is the discrete (p, 2) parity task, while we study the Gaussian noise XOR problem. Instead of a large SNR, those papers all require a large sample size $n$:  [Ji & Telgarsky (2019)](https://arxiv.org/abs/1909.12292), [Wei et al. (2019)](https://arxiv.org/abs/1810.05369), [Barak et al. (2022)](https://arxiv.org/abs/2207.08799), and [Telgarsky (2023)](https://openreview.net/forum?id=swEskiem99) all need $n = \Omega(p^2)$, and [Suzuki et al. (2023)](https://openreview.net/forum?id=tj86aGVNb3&referrer=%5BAuthor%20Console%5D(%2Fgroup%3Fid%3DNeurIPS.cc%2F2023%2FConference%2FAuthors%23your-submissions) need $n = \Omega(p)$. On the other hand, we study the high-dimensional regime $n \ll p$, which is in line with all existing benign overfitting results (e.g. [[Bartlett et al. (2019)](https://www.pnas.org/doi/10.1073/pnas.1907378117), [Frei et al. (2022)](https://arxiv.org/abs/2202.05928), [Cao et al. (2022)](https://arxiv.org/abs/2202.06526), [Frei et al. (2023)](https://jmlr.org/beta/papers/v24/22-1132.html), [Xu & Gu (2023)](https://proceedings.mlr.press/v206/xu23k.html), [Kou et al. (2023)](https://arxiv.org/abs/2303.04145)]) and intuitively explains why a relatively large SNR is needed.
>
> We also note that there is a concurrent work studying (p, 2) parity [[Glasgow (2023)](https://arxiv.org/abs/2309.15111)] which discusses the difference between the (p, 2) parity problem and the Gaussian noise XOR problem in their Conclusion and Discussion section and why their results cannot be extended to Gaussian noise XOR.
>
> ** **

---

> ### Comment · Reviewer_6CDP · 2023-11-22
>
> Dear authors,
>
> I appreciate authors' efforts on the extensive clarification.
>
> Best,

---

### Meta-Review · Area_Chair_EtoB · 2023-12-16

**Metareview:**

This paper shows the Grokking phenomenon after overfitting when training a 2 layer ReLU network for XOR problem in a high dimensional space.

The analysis is novel and gives interesting insight in the problem. The analysis successfully shows that after overfitting the data by the first gradient step, the network gradually maximizes the margin resulting in the Grokking phenomenon. Although a criticism about the high dimensionality raised by a reviewer is reasonable, the novelty of the analysis can be yet acknowledged. Then, I recommend acceptance for this paper as an AC.

**Justification For Why Not Higher Score:**

Although his paper gives an interesting analysis, the assumptions in the paper is rather restrictive. For example, the dimension is very high ($d >> n^{5}$) and the training input points are nearly orthogonal. Thus I think it is below spotlight.

**Justification For Why Not Lower Score:**

The dynamics shown in this paper provides essentially important viewpoint to understand the Grokking phenomenon. Hence, the paper deserves publication. Although one reviewer raised a criticism on the high dimensionality assumption, there were not critical issues in the paper.

---

### Decision · Program_Chairs · 2024-01-16

Accept (poster)